# Stress landscape of folding brain serves as a map for axonal pathfinding

Akbar Solhtalab [1], Ali H. Foroughi [1], Lana Pierotich[2] & Mir Jalil Razavi [1] ✉

Understanding the mechanics linking cortical folding and brain connectivity is crucial for both healthy and abnormal brain development. Despite the importance of this relationship, existing models fail to explain how growing axon bundles navigate the stress field within a folding brain or how this bidirectional and dynamic interaction shapes the resulting surface morphologies and connectivity patterns. Here, we propose the concept of "axon reorientation" and formulate a mechanical model to uncover the dynamic multiscale mechanics of the linkages between cortical folding and connectivity development. Simulations incorporating axon bundle reorientation and stress-induced growth reveal potential mechanical mechanisms that lead to higher axon bundle density in gyri (ridges) compared to sulci (valleys). In particular, the connectivity patterning resulting from cortical folding exhibits a strong dependence on the growth rate and mechanical properties of the navigating axon bundles. Model predictions are supported by in vivo diffusion tensor imaging of the human brain.

Brain development entails a series of complex processes, commencing with the growth of neural tubes and progressing through neuronal proliferation, glial cell proliferation, neuronal migration and differentiation, convolution, axonal wiring, synaptogenesis, and myelination[1]. Cortical folding of the brain takes place during the early stages of brain development, wherein the cerebral cortex, the outer layer of the brain, undergoes convolutions, forming gyri (ridges) and sulci (valleys)[2,3]. This dynamic folding of the cortex significantly increases its surface area, which facilitates enhanced neural connectivity and information processing. Cortical folding occurs primarily during the third trimester of the gestational period, which reaches completion after birth, and contributes to the intricate structure and functionality of the mature brain.

Different hypotheses and theories have been proposed to explain the underlying mechanisms of cortical folding (see the review by Arellano and Rakic)[4]. Multiple internal and external causes, e.g., radial constraint[5], cranial constraint[6], differential tangential growth (DTG) caused by cellular factors[7–9], and axon maturation[10,11], have been suggested to have various degrees of influence on cortical folding. DTG has been shown to stimulate secondary and tertiary folding[7,12], although other factors, such as axonal forces and heterogeneous growth in the cortical plate, play a role in the regulation of the brain morphology[13–15]. DTG theory states that growth rate mismatch between more rapidly growing outer layers of the brain and slowly growing inner layers acts as the driving mechanism for brain gyrification. This hypothesis has been supported by past studies and recent experimental and computational investigations[2,7,12,16–24].

The emergence and development of brain connectivity fiber tracts mainly occur simultaneously with cortical folding[25]. During the early development of the brain, axonal fibers extend in length to form connections between different regions of the brain. Brain fiber tracts grow bidirectionally, extending from the cortex to the white matter and from the white matter to the cortex. As the cortex begins to fold, these fibers navigate through the evolving landscape of gyri and sulci. Imaging studies suggest a notably higher concentration of axonal fibers in gyri than in sulci[13,26]. This aligns with the observation that gyri consist of a greater number of neurons compared to sulci[27]. Consequently, the proportionally larger number of axons that descend into the white matter in gyri contributes to the noted thickening of the white matter at folds[28]. As axonal fibers depart from or arrive at the cortex, their paths tend to closely align with the convolutions of the cortical surface[29]. As such, in the later stages of development of the

[1]Department of Mechanical Engineering, State University of New York at Binghamton, Binghamton, NY, USA. [2]Division of Newborn Medicine, Boston Children's Hospital, Harvard Medical School, Boston, MA, USA. ✉e-mail: mrazavi@binghamton.edu

macaque brain, there is an observable increase in fractional anisotropy within the developing gyral white matter when contrasted with sulcal regions[30,31]. The axonal fiber density in the human brain also indicates that axonal fibers concentrate on gyri rather than the wall and sulci of the folds[13,26]. Figure 1 illustrates the progression of growth and folding in an individual fetal brain, transitioning from a smooth state to a folded state between two developmental time points. Concurrently, brain connectivity also evolves during this period, which is closely intertwined with the folding of the cortical plate.

In addition to the DTG, axon maturation has also been suggested to play a role in the gyrification of the brain[10, 15,32–34]. Recent studies indicate that axonal pulling is not the inducer of the cortical folding even though axons are under considerable tension[15,33]. Previous analysis on the relationship between axonal elongation and cortical growth[15] concludes that rather than axons pulling on the brain to induce cortical folding, the folding cortex pulls on the axons to trigger axonal elongation and white matter growth[35]. A recent revision of the tension-based morphogenesis model has been introduced to address the limitations of the original model[11]. Despite primary efforts to incorporate the effect of axonal fibers in the folding process for idealized geometries using continuum stretch-driven models[15,23,30], the mechanics of the dynamic interplay between cortical folding and connectivity development is not well understood to date. Understanding this interplay is of great importance, because correlations between abnormal folding patterns and underlying connectivity have been observed in autism spectrum disorder (ASD)[36], schizophrenia[37], and bipolar disorder[38]. It is postulated that linked folding-connectivity alterations have an early onset, possibly during the fetal stage of neurodevelopment. While there is imaging evidence of the correlation between brain folding and connectivity maps, it is unknown how disruption in connectivity alters folding patterns, or how abnormal cortical folding alters the growth of axons and connectivity development.

The development of axons and brain connectivity is a multifaceted process influenced by various factors that include genetic predispositions and environmental stimuli. The established role of chemical cues in axon pathfinding is well-documented, with numerous essential chemoattractants and repellents having been identified and used in many theoretical models for axon pathfinding[39–41]. Unlike chemotaxis, the role of mechanical factors or gradients in the guidance of axons remains mainly unclear and has only very recently received attention[42–46]. The growing axons or axon bundles are responsive to the geometry and stiffness of the substrate[47–51]. It has been shown that the growth rate and orientation of extending axons are influenced by specific combinations of the stiffness value and stiffness gradient of the substrate[42,52]. Therefore, beyond chemical cues, mechanical forces and signals as pivotal modulators influence the guidance of axonal pathways[35,53,54]. Accordingly, numerous mathematical and computational growth models have been put forth to anticipate the trajectory of axonal growth across diverse physical scenarios, which encompass factors such as varying loadings and stretches[44,55–57]. These models aim to provide a comprehensive understanding of how axons evolve in response to distinct environmental stimuli, which contribute valuable insights into the intricacies of neural development under diverse conditions. However, there is currently no model to explain how growing axon bundles navigate the stress field within the folding brain, or how the bidirectional, dynamic interaction between gyrification and connectivity shapes the resulting folding morphologies and connectivity patterns. Therefore, a nuanced understanding is crucial to unravel the complexities of how axons navigate within the evolving cortical landscape, which provides valuable insights into the unique patterns of fiber growth and distribution in different brain regions. This will result in a better understanding of the underlying mechanisms of physical interplay between abnormal folding and altered connectivity in brain disorders.

In this study, we propose the concept of "axon reorientation" in the folding brain and formulate a mechanical model to uncover the multiscale mechanics of the linkages between cortical folding and connectivity development. In contrast to previous studies that focused

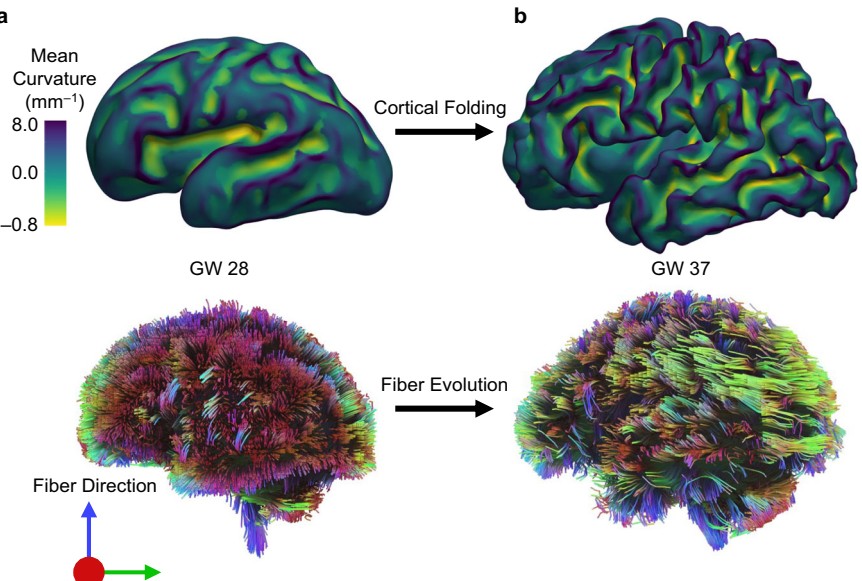

**Fig. 1 | Growth and folding of the human fetal brain. a** Top: Reconstructed surface of the brain from the magnetic resonance image (MRI) of a human fetal brain before the formation of folds at 28 gestational week (GW). Bottom: Fiber tractography of the same brain reconstructed from the diffusion tensor imaging (DTI) data. The white matter is composed mainly of axonal fiber bundles that reside in numerous fiber tracts of the brain. **b** Top: MRI of the same brain as in (**a**) at 37 GW, after growth and folding. Numerous gyri and sulci have formed on the surface of the brain. Bottom: Fiber tractography of the brain reconstructed from the DTI data at 37 GW. Fiber tracts follow gyri and sulci patterns. Various colors in the fiber tractography images represent the orientation of axonal fiber bundles. For better clarity, readers are referred to the color version of the figure in the online version of this article. The images were generated using the open-access tool FEDI and shared longitudinal fetal brain scans provided by Boston Children's Hospital under a data transfer agreement [Motion-robust super-resolution diffusion weighted MRI of early brain development] with Binghamton University.

on passive fibers[15,24,34,58] or active continuum-based fibers[30], our framework uniquely captures the intricate behavior of axon bundles, including stress-induced and stochastic growth, as well as the proposed reorientation response. The concept of "axon reorientation" in the folding brain originates from basic experiments demonstrating that neurites preferentially orient toward the direction of stretching[59,60]. We hypothesize that, in addition to stress-induced growth, axons reorient toward the direction of maximum tensile principal stress to establish stereotyped subcortical fiber patterns. Our simulations, incorporating this complex axon bundle growth and pathfinding, offer a more accurate representation than previous models, particularly in explaining why gyri have a higher axon density than sulci. These predictions are well-supported by in vivo diffusion tensor imaging of the human brain.

This modeling study focuses on elucidating the growth and pathfinding mechanisms of axons destined to extend from white matter toward the cortical plate. Those axons can be thalamocortical fiber tracts, monosynaptic projections to the cortex from subplate neurons, and ascending, monosynaptic projections from the basal forebrain, brainstem, and hypothalamus. The growth of thalamocortical fibers initiates in the embryonic period and concludes by the third trimester of gestation in humans, which coincides with cortical folding (Table S1)[61].

## Results

We investigated how axon bundles navigate and find their paths in a system that undergoes simultaneous folding, aiming to understand the dynamic interaction between cortical folding and connectivity development. We examined various factors that influence the interplay of cortical folding and connectivity development, including the biophysical properties of axons, their stochastic and stress-induced growth nature, and the mechanical properties of the cortex and underlying layers. It is important to note that all results and findings are derived from predictive models. These models suggest potential mechanisms that may apply to the developing brain, rather than describing the exact process in the human brain. The terminology used for model constituents and parameters is aligned with that of the human brain solely to establish an analogy.

### Axonal growth in a stress field

The main goal of this study is to elucidate the mechanics underlying axonal growth within the context of a folding brain, while also exploring the complex interplay between folding patterns and neuronal connectivity. Figure 2 illustrates the dynamic growth of axon bundles, hereafter referred to as fibers for simplicity, within a deformation field across the course of cortical expansion and folding, with time scaled to a dimensionless simulation time $T = G^{ctx}t$, that ranges from 0 to 1. At $T = 0$, cortical expansion initiates (corresponding to ~16–20 GWs in Table S1), when thalamocortical fiber tracts begin to spread in the deep subplate. At $T = 1$, the simulation ends (aligning with around 27–36 GWs in Table S1), when axons have settled in the cortical plate. We assumed a fixed growth rate of $G^{axn} = 0.8$ mm d$^{-1}$ and stress-induced elongation rate of $a = 0.015$ mm Pa$^{-1}$ d$^{-1}$ (Table S2). Prior to the onset of cortical folding, the expansion of the cortex induces relatively low levels of tension throughout the white matter region (bottom layer). As a result, the growth of fibers follows relatively random trajectories, as shown in Fig. 2a, b. Initial angles are randomly assigned within a range of $\pi/2 \pm 10\%$, and noise in the angle during the simulation is modeled using a Gaussian distribution with a mean of zero and a standard deviation of 0.025 rad[44]. Arrows in Fig. 2a, b indicate the direction of the growth and should not be confused with the maximum tensile principal stresses (MTPS) directions of the ECM before cortical folding, because they remain low during this initial phase. However, as folding initiates, the MTPS increases, creating primarily tangential tension beneath prospective sulci and primarily radial tension beneath prospective gyri, as shown in the normalized maximum principal stress contours. This observation aligns with previous studies on stresses within the developing brain[30]. Figures 2c–f show that as MTPS increases, a reorientation process occurs, with the preferred directions (indicated by green arrows) gradually realigning toward their corresponding MTPS directions (indicated by pink arrows). In the developed code, the growth of fibers is terminated once they reach the interface of the cortex and white matter. By the completion of cortical folding at $T = 1$, the reorientation process also concludes, with the preferred directions nearly aligned with their respective MTPS directions (Fig. 2f). The stress patterns induced by folding lead to alterations in the fibers alignment, which visually result in heightened fiber density beneath gyri. This stereotypical arrangement beneath folds mirrors the organizational patterns observed in the adult human brain[26]. The dynamic visualization of Fig. 2 is available in Supplementary Movie 1.

A sensitivity analysis was conducted to assess the robustness and reliability of the simulation framework in capturing the intricate dynamics of fiber growth and reorientation during brain cortical folding. Specifically, we focused on evaluating the influence of varying the number of simulations and their random distributions on the calculated fiber density within gyral and sulcal regions. Remarkably, our analysis revealed that beyond a threshold of 10 simulations, the fiber density stabilizes, which indicates convergence of results (Fig. S1a). Therefore, all subsequent simulations were conducted using a fixed number of 10 simulations, and the results were averaged.

In addition to assessing the sensitivity of our simulation framework to the number of simulations, we investigated the sensitivity of the fiber density in gyri and sulci to the number of fiber bundles within the white matter for a given set of parameters. Our analysis revealed that increasing the number of fibers beyond ~100, has a negligible effect on the fiber density in gyri and sulci after folding. Figure S1b illustrates the insensitivity of the fiber density in gyri and sulci beyond 100 fibers. Therefore, all subsequent simulations were conducted using a fixed number of 100 fibers to reduce the simulation time.

### Effect of axon growth rate on fiber organization

In this section, we explore the influence of the axon growth rate on the organization of fibers within the developing white matter. Simulations were conducted employing four distinct axon growth rates: $G^{axn} = 0.6$, 0.8, 1.0, and 1.2 mm d$^{-1}$ according to Table S2, while maintaining a fixed stress-induced elongation rate of $a = 0.015$ mm Pa$^{-1}$ d$^{-1}$ across all scenarios. Additionally, all other parameters within the simulation framework remained constant. Figure 3a, b compares the growth of fibers under two different growth rates (0.6 and 1.2 mm d$^{-1}$) in a system that simultaneously undergoes folding. As evident from the simulations, increasing the axon growth rate reduces the chaotic behavior of fiber trajectories. Notably, a higher axon growth rate facilitates a greater number of fibers that reach the cortical interface prior to the onset of cortical folding. As shown in Fig. 3a, most fibers at a growth rate of 1.2 mm d$^{-1}$ have reached the cortical surface by $T = 0.5$, before the initiation of folding. Conversely, for a growth rate of 0.6 mm d$^{-1}$, no fibers have reached the cortical interface by this time point (Fig. 3b).

In Fig. 3c, quantitative analysis of fiber density within gyri across different growth rate scenarios revealed intriguing patterns. Regardless of the growth rate, the fiber density in gyri is greater than in sulci. However, as the fiber growth rate increases, there is a noticeable decrease in fiber density within gyri, accompanied by a corresponding increase in density within sulci. The models predict that doubling the growth rate roughly decreases the fiber density in gyri by 28%. The underlying rationale for this observation can be elucidated through an understanding of the association time between folding of the system and growth of fibers. At lower growth rates, such as 0.6 mm d$^{-1}$, many fibers fail to reach the cortical interface before folding. Consequently,

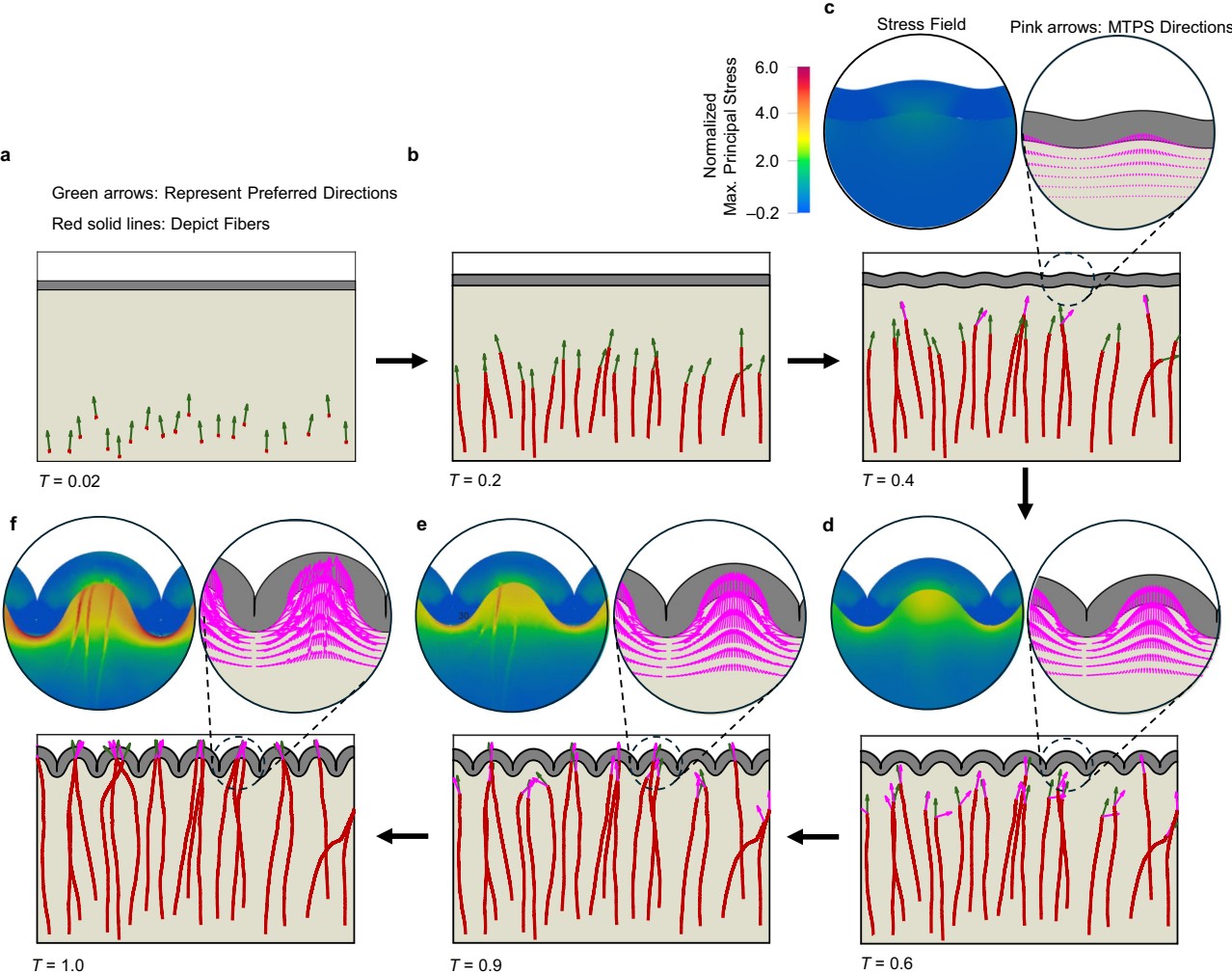

**Fig. 2 | Dynamic growth and pathfinding of fibers within a growing and folding bilayer system.** In this specific model, only 20 fibers have been initiated and grown for a better representation of the results. Pink arrows represent the maximum tensile principal stress (MTPS) directions and green arrows show the preferred directions. $T = G^{ctx}t$ is dimensionless simulation time that ranges from 0 to 1, where $G^{ctx}$ and $t$ are cortex growth rate and time, respectively. Normalized Maximum Principal Stress (defined as the ratio of the maximum principal stress to the shear modulus of the ECM ($\mu_s$)) is shown for each step. For better clarity, readers are referred to the color version of the figure in the online version of this article. **a** Fibers begin growing from the base of the white matter, extending upward toward the cortical plate. **b** Growth continues with fibers following random pathways. **c** Fibers keep growing, and bifurcation, marking the onset of significant morphological changes associated with fold formation, occurs at this stage. **d** During cortical folding, the Maximum Tensile Principal Stress (MTPS) directions (pink arrows) become predominantly horizontal beneath sulci and relatively vertical beneath gyri. Fibers reorient toward high-stress regions near the cortex, reorienting in alignment with the MTPS. **e**, **f** Fibers further reorient toward the MTPS and eventually reach the cortex. Notably, the majority of fibers terminate within the gyri region.

during the folding process, these fibers encounter regions of elevated stresses, which trigger a reorientation phenomenon and increased growth rate. As the MTPS beneath gyri is predominantly radial, and beneath sulci is tangential, the reorientation process gradually steers fibers toward the gyral regions. Conversely, with an increase in fiber growth rate, a larger proportion of fibers reach the cortical surface before folding commences, diminishing the opportunity for significant reorientation. This explains the decrease in fiber density within gyri and the corresponding increase within sulci.

In addition to the evaluation of fiber density in gyri and sulci, a statistical analysis was conducted to identify the significant levels based on multiple comparisons. The statistical analysis shown in Fig. 3c revealed significant differences in fiber density within gyri across growth rates. Specifically, comparisons between the growth rate of $0.6\ \mathrm{mm\ d^{-1}}$ and those of $1.0\ \mathrm{mm\ d^{-1}}$ and $1.2\ \mathrm{mm\ d^{-1}}$ yielded statistically significant differences in both gyri and sulci. Similarly, comparisons between the growth rate of $0.8\ \mathrm{mm\ d^{-1}}$ and those of $1.0\ \mathrm{mm\ d^{-1}}$

and $1.2\ \mathrm{mm\ d^{-1}}$ also yielded significant differences. The significant levels are indicated with $p$-values on Fig. 3c. These findings indicate that as growth rate increases, fiber density significantly declines in gyri and increases in sulci. Theoretically, this statistical finding indicates that higher growth rates correspond to decreased fiber density within gyral regions.

Figure 3c also indicates that beyond a certain growth rate of axons ($\sim 1\ \mathrm{mm\ d^{-1}}$), there is no significant change in the fiber density in gyri and sulci. This can also be attributed to the fact that at higher growth rates, fibers reach the cortical interface before the onset of folding. As a result, the folding-induced stress field of the ECM lacks sufficient time to redirect the fibers. Supplementary Movies 2 and 3 illustrate the dynamic growth and pathfinding of fibers at two distinct axon growth rates.

We chose a model size large enough relative to the wavelength of the folds to minimize the impact of boundary conditions, with a fixed boundary at the bottom and symmetric boundaries at the sides. To

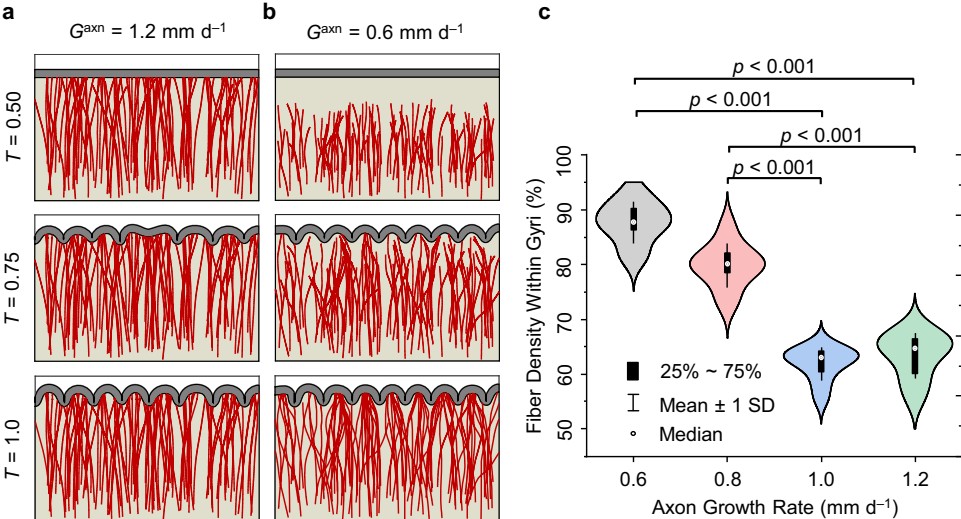

**Fig. 3 | Effect of the axon growth rate on the spatial distribution of fibers within the white matter region.** The plot showcases simulations conducted at axon growth rates ($G^{axn}$) of **a** 1.2 mm d$^{-1}$, and **b** 0.6 mm d$^{-1}$, which represent contrasting scenarios of fast and slow fiber growth, respectively. The dimensionless simulation time, $T = G^{ctx}t$, ranges from 0 to 1, where $G^{ctx}$ is the cortical growth rate, and $t$ is time. All other model parameters were held constant across all scenarios: $\mu_c/\mu_s = 2$, $\mu_f/\mu_s = 2$, $a = 0.015$ mm Pa$^{-1}$ d$^{-1}$, where $\mu_c$, $\mu_s$, and $a$ denote the shear modulus of the cortex, the shear modulus of the ECM and the stress-dependent elongation rate, respectively. **c** Effect of axon growth rate on fiber density within gyri after folding. The white dot indicates the median, while the black box represents the interquartile range (IQR), covering the 25th to 75th percentiles. The whiskers display the mean +/- SD (Standard Deviation). The analysis was conducted across four different axon growth rates, each repeated 10 times ($n = 40$). $p$-values are displayed on the figure, with a significance level of 0.05. All statistical tests were two-sided, with Tukey's post hoc method used for multiple comparisons. Source data are provided as a Source Data file.

ensure that the observed results are not dependent on the selected model geometry, we also developed circular models with radial fibers using the same parameters as in the rectangular models. A sample result is presented in Fig. S2, where two circular models with different axon growth rates are compared. As shown, the growth, pathfinding, and settlement of fibers in gyri rather than sulci follow a similar trend as in the rectangular model, confirming independence from boundary effects.

## Effect of axon mechanical properties on fold morphology and fiber organization

We explored the effect of the stiffness ratio of axon to ECM ($\mu_f/\mu_s$) across different scenarios to examine the influence of axon mechanical properties on cortical folding and fiber organization. Our analysis of fiber density within gyri and sulci, detailed in Fig. 4a, revealed significant changes across different stiffness ratios. Alterations in the stiffness ratio notably impact the overall density of fibers within gyri and sulci. As the stiffness ratio increases, fiber density in gyri significantly increases, while it decreases in sulci. Statistical analysis further disclosed significant differences in fiber density between stiffness ratios (Fig. 4b). Specifically, comparisons between a stiffness ratio of 1 and those of 2, 4, and 5 yielded statistically significant outcomes for both gyri and sulci. This phenomenon may be explained in terms of the stiffening effect induced by higher stiffness ratios within the white matter region, which prevents the formation of sulci in regions with dense fibers. In Fig. 4a, with a stiffness ratio of 1, which creates an isotropic substrate for folding of the cortex, fibers have no effect on the morphology of cortical folds. Conversely, as the stiffness ratio increases, variations in stiffness emerge within the subcortical domain, which alter the morphology of folds. Notably, regions with lower fiber density are prone to sulci formation, as evidenced by the comparison between stiffness ratios of 1 and 5 in Fig. 4a. Consequently, an increase in the stiffness ratio correlates with higher fiber density in gyri. The models predict that beyond a certain stiffness ratio of axons to ECM (~2), there is no significant change in the fiber density in gyri and sulci.

Furthermore, we investigated the effect of stiffness ratio on the final folding morphologies by using the gyrification index (GI) as a metric for comparison, as shown in Fig. 4c. The GI is defined as the ratio of the total length of the cortical contour (solid line in Fig. 4c) to the length of the convex hull (dashed line in Fig. 4c); refer to the Methods section for further details. This ratio indicates the extent of gyrification and surface complexity. Remarkably, with an increase in the stiffness ratio, the GI exhibited a statistically significant decrease, while its absolute value changed only slightly. This phenomenon can be attributed to the stiffening effect caused by higher stiffness ratios within the white matter region. This results in a non-compliant, stiffer substrate that limits the extent of deformation in areas with dense fibers, which therefore restricts the buckling of the cortex. Similar to the trend observed in fiber density within gyri and sulci, an increase in the stiffness of the fibers beyond the threshold (~2) does not have a significant effect on the GI (Fig. 4c).

Our results, as previously mentioned, indicate that the density of fibers in gyri and sulci remains consistent once the number of fibers exceeds 100. However, the number of fibers and their dynamic growth alter the material properties of the white matter (bottom layer) at each simulation step. This dynamic stiffening, which is achieved by converting ECM elements to fiber elements, can alter the final GI of the folding patterns. To understand the development of dynamic and heterogeneous stiffness map of the white matter region, we constructed various models with different numbers of fibers to represent different fiber volume fractions. In the 2D scenario, the fiber volume fraction is determined by the ratio of the surface area occupied by fiber elements within a specified region to the total surface area of that region. Local stiffness ratio is defined as the fiber volume fraction within a 1 mm radius circle centered around each material integration point, plus one, as given by $V_f \left( \frac{\mu_f}{\mu_s} \right) + V_{ECM} \left( \frac{\mu_s}{\mu_s} \right) = V_f + 1$ for models with $\frac{\mu_f}{\mu_s} = 2$. Equivalent stiffness ratio is defined as $V_f \left( \frac{\mu_f}{\mu_s} \right) + V_{ECM} \left( \frac{\mu_s}{\mu_s} \right)$ across the entire white mater region, updated at each simulation timestep. Figure 5a depicts models with varying fiber counts, ranging from 100

to 400, while Fig. 5b illustrates the local stiffness ratio map of the model's white matter region, resulting from the initiation, growth, and establishment of these fibers. As the volume fraction of stiff fibers increases, the local and overall stiffness of the bottom layer increases, resulting in a heterogeneous stiffness map. This behavior mirrors the stiffness map characteristics of white matter tissue as predicted by magnetic resonance elastography (MRE) studies[62]. Figure 5c, d illustrate the dynamic changes in fiber volume fraction and, consequently, the equivalent stiffness of the white matter region. Over time, both the fiber volume fraction and the equivalent stiffness increase and stabilize once the fibers reach the cortical plate and new fiber generation ceases. Increased stiffness in the white matter region leads to a reduction in cortical gyrification and a lower GI (Fig. 5e), although this effect becomes statistically significant only with a substantial change in the fiber volume fraction.

## Effect of cortex stiffness on fiber organization

In this section, we study the effect of the stiffness ratio between the cortex and ECM ($\mu_c/\mu_s$) on the navigation of fibers in the white matter and the formation of cortical folds. In the models, we used four distinct stiffness ratios (1, 2, 3, and 4), while keeping other parameters constant. We selected a stiffness ratio range of 1 to 4 for our parametric study, as the stiffness ratio between the cortex and subplate (later white matter) in a folding brain may vary widely due to the changing material properties of brain tissue during folding[63]. However, most experimental evidence supports a ratio of 1 to 2 between cortical and white matter stiffness in the mature brain[64, 65].

Figure 6a compares the fiber development and folding of two models with different stiffness ratios of the cortex to the ECM (1 and 4). Visually, as the stiffness ratio increases, the gyral-sulcal thickness ratio decreases, which is a trend consistent with previous findings[66]. Moreover, cortical folding initiates at earlier times with increasing stiffness ratio. For example, as illustrated in Fig. 6a, for the stiffness ratio of 1, cortical folding commenced approximately at $T = 0.75$, while it began around $T = 0.5$ for a stiffness ratio of 4. The timing of cortical folding has a significant effect on the reorientation process of fibers. The model predicts that folding begins earlier for higher stiffness ratios (e.g., 3 or 4), which prompts the reorientation process to occur earlier. This results in a predominant realignment of fibers toward the MTPS direction, particularly toward the gyri before they reach the cortex (Fig. 6a, right column). In contrast, at lower stiffness ratios (e.g., 1), buckling occurs later, which allows most fibers to reach the cortex before buckling occurs (Fig. 6a, left column). From a modeling perspective, this temporal difference in cortical folding significantly impacts the final organization of fibers in the white matter.

Quantitative analysis of fiber density within gyri, as shown in Fig. 6b, underscores the visual observations. With an increase in the stiffness ratio, a noticeable increase in fiber density within gyri occurs, accompanied by a corresponding decrease in density within sulci. This shift in fiber density distribution highlights the pivotal role of the cortex to ECM stiffness ratio in shaping cortical folding dynamics and fiber organization. The statistical analysis revealed significant differences in fiber density between stiffness ratios. Specifically, comparisons between the stiffness ratio of $\mu_c/\mu_s = 1$ and those of 2, 3, and 4 showed statistically significant differences for both gyri and sulci. Supplementary Movies 4 and 5 showcase the dynamic growth and pathfinding of fibers under two different stiffness ratios.

In the models, the initial thickness of the cortex is kept constant, resulting in similar folding patterns across all of them. However, varying the initial thickness can significantly alter the resulting fold shapes. Figure S3 illustrates that increasing the initial thickness of the cortex layer leads to larger folds and deeper penetration of the stress field into the white matter. Despite these variations, the growth and reorientation of fibers remain consistent with the base models. Additionally, variations in cortex thickness or heterogeneous growth within

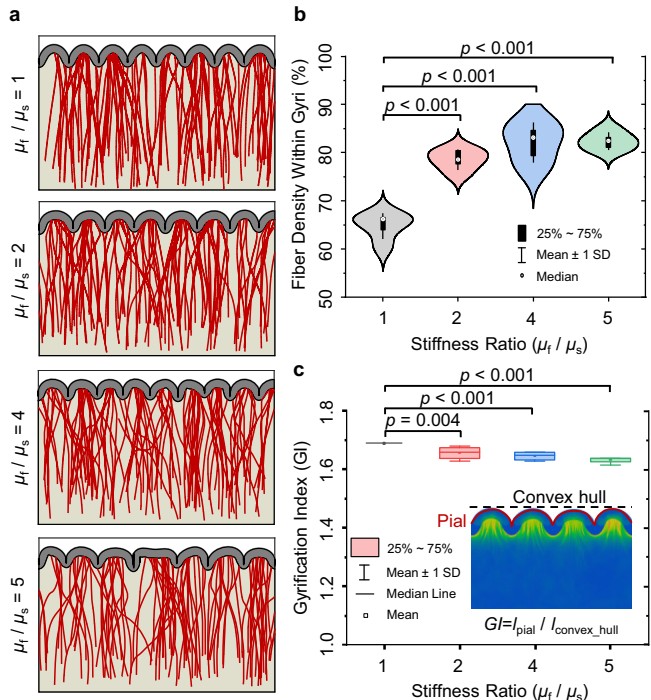

**Fig. 4 | Effect of axon-to-ECM stiffness ratio on fiber organization, density, and the gyrification index. a** Effect of stiffness ratio of axon to white matter ECM ($\mu_f/\mu_s$) on fold morphology and fiber organization. Ratio 1 indicates that there is no difference between the material properties of axons and ECM, which creates an isotropic substrate for the folding of the cortex. All other model parameters were kept constant across all scenarios: $G^{axn} = 0.8\,\mathrm{mm\,d^{-1}}$, $\mu_c/\mu_s = 2$, and $a = 0.015\,\mathrm{mm\,Pa^{-1}\,d^{-1}}$, where $G^{axn}$ is the axon growth rate, $\mu_c$ and $\mu_s$ denote the shear moduli of the cortex and ECM, respectively, and $a$ is the stress-dependent elongation rate. **b** Fiber density within gyri with respect to the various stiffness ratios ($\mu_f/\mu_s$). Changing the stiffness ratio significantly affects the fiber density in gyri and sulci. The white dot represents the median, the black box indicates the interquartile range (IQR), spanning the 25th to 75th percentiles, and the whiskers represent the mean +/- SD (Standard Deviation). The analysis includes a sample size of $n = 40$, with each stiffness ratio repeated ten times. $p$-values indicating significance levels are displayed in the graph. **c** The effect of the axon's material properties on the gyrification index (GI) of the formed folds is illustrated. The white dot represents the median, the colored box denotes the interquartile range (IQR), spanning the 25th to 75th percentiles, and the whiskers show the mean +/- SD (Standard Deviation). The analysis was conducted with a sample size of $n = 40$. The $p$-values are presented on the figure, with a significance threshold set at 0.05. Statistical analyses were two-sided, utilizing Tukey's post hoc method for multiple comparisons. Source data are provided as a Source Data file.

the cortex could produce different folding morphologies. This underscores the complexity of cortical fold variations in the human brain.

## Effect of stochastic nature of axon growth on fiber organization

In the previous sections, both tip growth and stress-induced growth were applied to the navigating fibers, with tip growth exhibiting controlled stochastic behavior to effectively guide fibers from the white matter region toward the cortex. In this section, we explore how increased stochastic behavior, potentially caused by disorders, affects fiber organization. Previous experimental studies have used Taxol (an inhibitor of microtubule dynamics) and Blebbistatin (a disruptor of actin filaments) to investigate the role of cytoskeletal dynamics in the axon steering mechanism[48]. The disruption of cytoskeletal dynamics in chemically treated neurons results in a significant decrease in axonal alignment due to alterations in cell-substrate interactions. To mimic the stochastic nature of axonal development and the randomness of

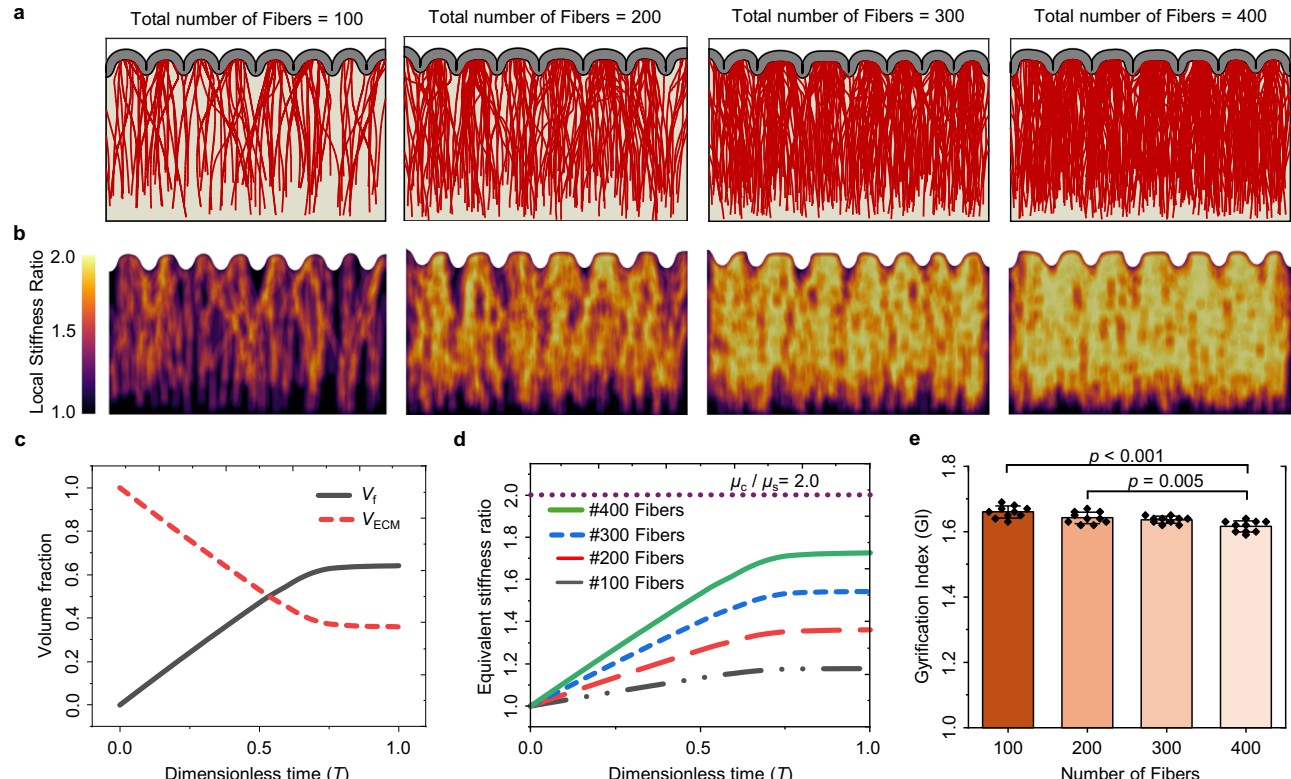

**Fig. 5 | Effect of fiber volume fraction on fold morphology and mechanical properties of the white matter. a** Effect of number of fibers on fold morphology and fiber organization. The parameters used are as follows: $G^{axn} = 0.8$ mm d$^{-1}$, $\mu_f/\mu_s = 2$, $\mu_c/\mu_s = 2$, and $a = 0.015$ mm Pa$^{-1}$ d$^{-1}$, where $G^{axn}$ is the axon growth rate, $\mu_f, \mu_s$, and $\mu_c$ are the shear moduli of the fiber, ECM, and cortex, respectively, and $a$ is the stress-dependent elongation rate. **b** Effect of fiber volume fraction on local stiffness ratio. **c** Change in overall volume fraction of fibers ($V_f$) and ECM ($V_{ECM}$) as a function of time for the case with 400 fibers. Fiber volume fraction starts from zero and gradually increases, while the volume fraction of ECM decreases. The total

volume fraction of fibers and ECM always sums to 1. **d** Change in the equivalent stiffness ratio of the white matter as a function of time. **e** Dependency of the GI to the number of fibers in the white matter region. Error bars represent the mean +/- SD (Standard Deviation), while black dots indicate individual data points. The analysis was conducted with a sample size of $n = 40$. *p*-values are displayed on the figure, with a significance level of 0.05 considered. Statistical analyses were two-sided, utilizing Tukey's post hoc method for multiple comparisons. Source data are provided as a Source Data file.

growth orientation observed in in vitro cell culturing experiments, we varied the standard deviation of the orientation angle of the preferred growth directions in the models. As shown in Fig. 7a–d, an increase in the standard deviation results in fibers that exhibit more chaotic growth trajectories. In some instances, these fibers fail to reach the cortex within the allocated time frame, which has minimal effect on the surface morphology of the folds.

We analyzed fiber growth patterns by determining the angular distribution of each fiber's tip relative to the positive "*x*" direction. We confined our analysis to angles within the range of $[0, \pi]$. The normalized angular distributions in Fig. 7a–d clearly show how randomness in fiber growth direction affects alignment and final organization of fibers. The data reveal a drastic decline in fiber alignment as the standard deviation increases. For instance, in Fig. 7a where the standard deviation is 0.025 rad, the majority of fibers grow predominantly in the $\pi/2$ direction. However, with a progressive increase in the standard deviation, the directional coherence of fibers diminishes significantly. This trend is consistent with in vitro findings when the axons are either intact or treated with Taxol and Blebbistatin[48].

Increasing the randomness effect in the growth of fibers (e.g., from 0.025 rad to 0.125 rad) results in fewer fibers that successfully reach their destination in gyri, as shown in Fig. 7e. This is because fibers with a significant degree of randomness in growth struggle to navigate and reach the cortex within a predefined time frame. This randomness predominantly affects the number of settled fibers in gyri more than in sulci. In cases of extreme randomness (e.g., 0.125 rad), even the folding

of the cortex is unable to redirect or increase the growth rate of the navigating fibers. Figure 7f indicates that the randomness in the growth of fibers alters the fiber density in gyri and sulci, although gyri consistently have more fibers than sulci in any scenario. The fiber density was calculated based on the fibers that reached gyri and sulci.

### Findings from imaging data for model evaluation
In this section, we aim to evaluate our simulations by comparing them with imaging data obtained from high-resolution sMRI and dMRI scans (see Methods section). We specifically used scans from 10 healthy adult brains aged 22−35 years to analyze the organization of thalamocortical fiber tracts. These fibers extend from the thalamus to cortical regions, similar to the simulations in directing fibers from the core to the cortex. The development and settlement of thalamocortical fiber tracts occur simultaneously with brain folding, which makes them suitable candidates for model evaluation (Table S1).

To showcase the anticipated anisotropy within the subplate, high-resolution DTI was used for an adult brain. Figure 8a illustrates fiber trajectories in the brain's coronal plane, with fiber orientations highlighted by the yellow solid line. Notably, beneath the sulci, fibers predominantly exhibit tangential alignment, while beneath the gyri, they align radially. These findings, consistent across all 10 cases, confirm a widespread pattern of tangential alignment near sulci and radial alignment near gyri. This aligns with our modeling predictions, suggesting a shift in fiber trajectory near sulci and subsequent radial alignment upon reaching gyri. Additionally, fiber trajectories

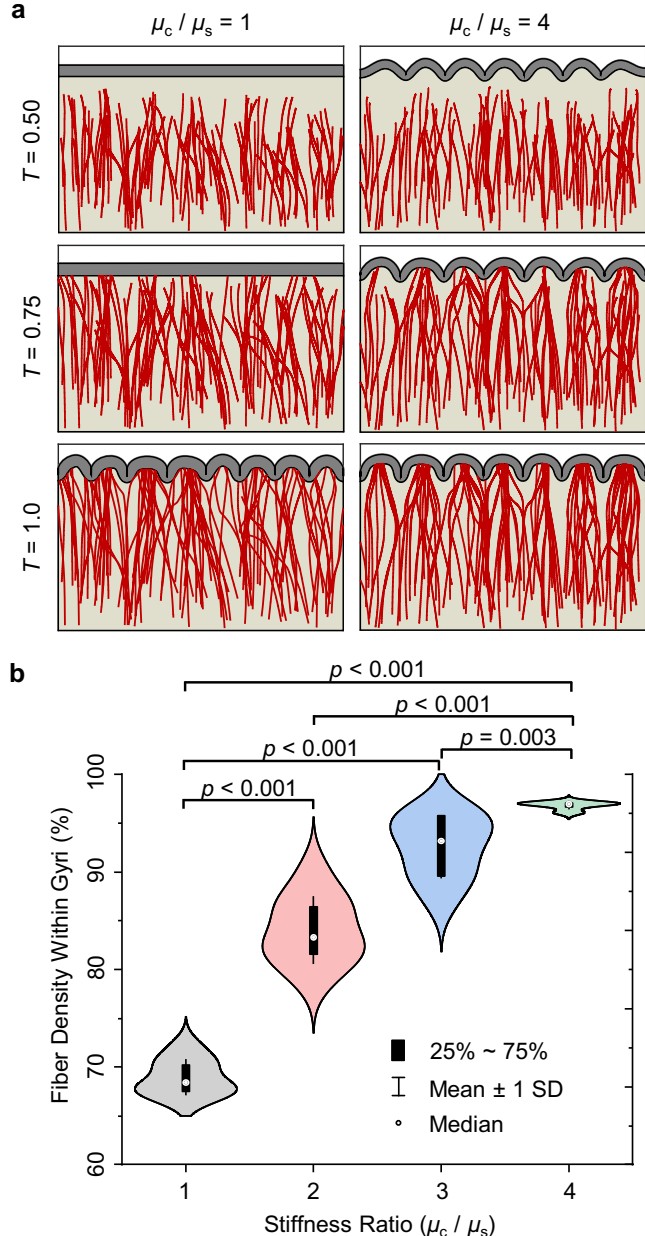

**b**

**Fig. 6 | Effect of the stiffness ratio of the cortex to the white matter ECM ($\mu_c/\mu_s$) on the navigation of fibers within the white matter and the formation of cortical folds.** The material properties for all scenarios are $\mu_f/\mu_s = 2$, $G^{axn} = 0.8\,mm\,d^{-1}$, $a = 0.015\,mm\,Pa^{-1}\,d^{-1}$, where $\mu_f$ and $\mu_s$ are the shear moduli of the fiber and ECM, respectively, $G^{axn}$ is the axon growth rate, and $a$ is the stress-dependent elongation rate. **a** Cortical folding and fiber organization for stiffness ratios of 1 (left) and 4 (right). **b** Fiber density within gyri for different stiffness ratios of the cortex to the white matter ECM ($\mu_c/\mu_s$). The white dot represents the median, the black box denotes the interquartile range (IQR), spanning the 25th to 75th percentiles, and the whiskers indicate the mean +/- SD. The analysis was conducted with a sample size of $n = 40$. The figure displays $p$-values, with a significance threshold of 0.05 being applied. Statistical analyses were two-sided, utilizing Tukey's post hoc method for multiple comparisons. Source data are provided as a Source Data file.

terminate in gyral areas, aligning with our simulations. Figure 8b presents similar results from axial perspectives, further confirming these observations.

Following the extraction of thalamocortical fiber tracts for each case, we calculated the fiber density within gyri and sulci (Table S3). The average fiber density of 10 cases is illustrated in Fig. 8c, and

notably, our analysis revealed a predominant settlement of fibers within gyri ($\sim 80\%$). Therefore, our models closely align with imaging findings that indicate a significantly greater fiber density in gyri compared to sulci. In addition, observations of subplate orientation suggest that fibers change their trajectories as they approach sulcal regions, favoring settlement within gyri. According to our simulation results, the proposed "axon reorientation" mechanism could potentially explain this behavior. However, it is worth noting that the quantified fiber density in gyri and sulci in the imaging cases encompasses all factors involved in human brain development, whereas in the models, only the parameters discussed in previous sections have been evaluated. Therefore, it is important to be cautious when making direct comparisons. Projection and distribution of thalamocortical fiber tracts on the white matter surface for 10 brains are shown in Fig. S4. For all cases, most of the fibers have terminated within gyri.

To further analyze the contribution of the reorientation process in fiber development and patterning, we conducted simulations with and without a stress-induced reorientation process for different axon growth rates, using an axon-to-ECM stiffness ratio of 2. Figure 8d illustrates the dynamic growth of fibers for a growth rate of $0.6\,mm\,d^{-1}$, which provides a comparison between simulations conducted with and without the stress-induced reorientation process. In the simulation conducted in the absence of the stress-induced reorientation, fibers navigate relatively randomly in the white matter region, showing no clear preference for alignment or directionality. In contrast, in the model with the stress-induced reorientation, fibers demonstrate more organized and directional growth trajectories, aligning toward the MTPS direction. This alignment is particularly evident as fibers approach the cortical surface, where they tend to orient toward the gyri. The dynamic growth of fibers for a growth rate of $1.2\,mm\,d^{-1}$ is shown in Fig. S5. The comparison with the imaging data highlights the significant potential influence of the reorientation process on shaping the spatial organization and alignment of fibers within the developing brain.

Figure 8e for growth rates of 0.6 and $0.8\,mm\,d^{-1}$ illustrates that the fiber density within gyri is significantly higher when the stress-induced reorientation process is incorporated in the models compared to models where this process is not included. However, for higher growth rates, such as 1.0 and $1.2\,mm\,d^{-1}$, the fiber density remains roughly the same regardless of the presence or absence of a stress-induced reorientation process. This can be attributed to the fact that when the growth rate is sufficiently high, most fibers reach the cortex before folding, which limits the opportunity for reorientation toward the MTPS direction. Additionally, even without the stress-induced reorientation process, the fiber density in gyri remains slightly higher than in sulci. This observation may stem from the difference in stiffness between fibers and the ECM, which provides a dynamically heterogeneous substrate for cortical folding. In this condition, regions packed with stiff fibers beneath the cortex prevent its downward deformation, leading to invagination in areas with fewer fibers. As a result, gyri form in regions where fibers are densely packed, while sulci form in areas with fewer fibers. This phenomenon is aggravated at higher growth rates of axons (1 and $1.2\,mm\,d^{-1}$), where stiff fibers reach beneath the cortex earlier than slowly growing fibers (0.6 and $0.8\,mm\,d^{-1}$) and exert more control over the determination of gyri and sulci locations. As a result, fiber density in gyri increases in models with higher growth rates compared to those with lower growth rates.

The dependency of fiber density in gyri on the axon growth rate differs between models with stress-induced reorientation and those without it. At low growth rates (0.6 and $0.8\,mm\,d^{-1}$), the stress field of the deforming substrate has sufficient time to redirect fibers more toward gyri, while at higher growth rates (1 and $1.2\,mm\,d^{-1}$), fibers reach the cortex before the onset of folding, and therefore, the stress-induced reorientation process does not have sufficient time to guide fibers. Based on the quantified imaging and simulation results shown in

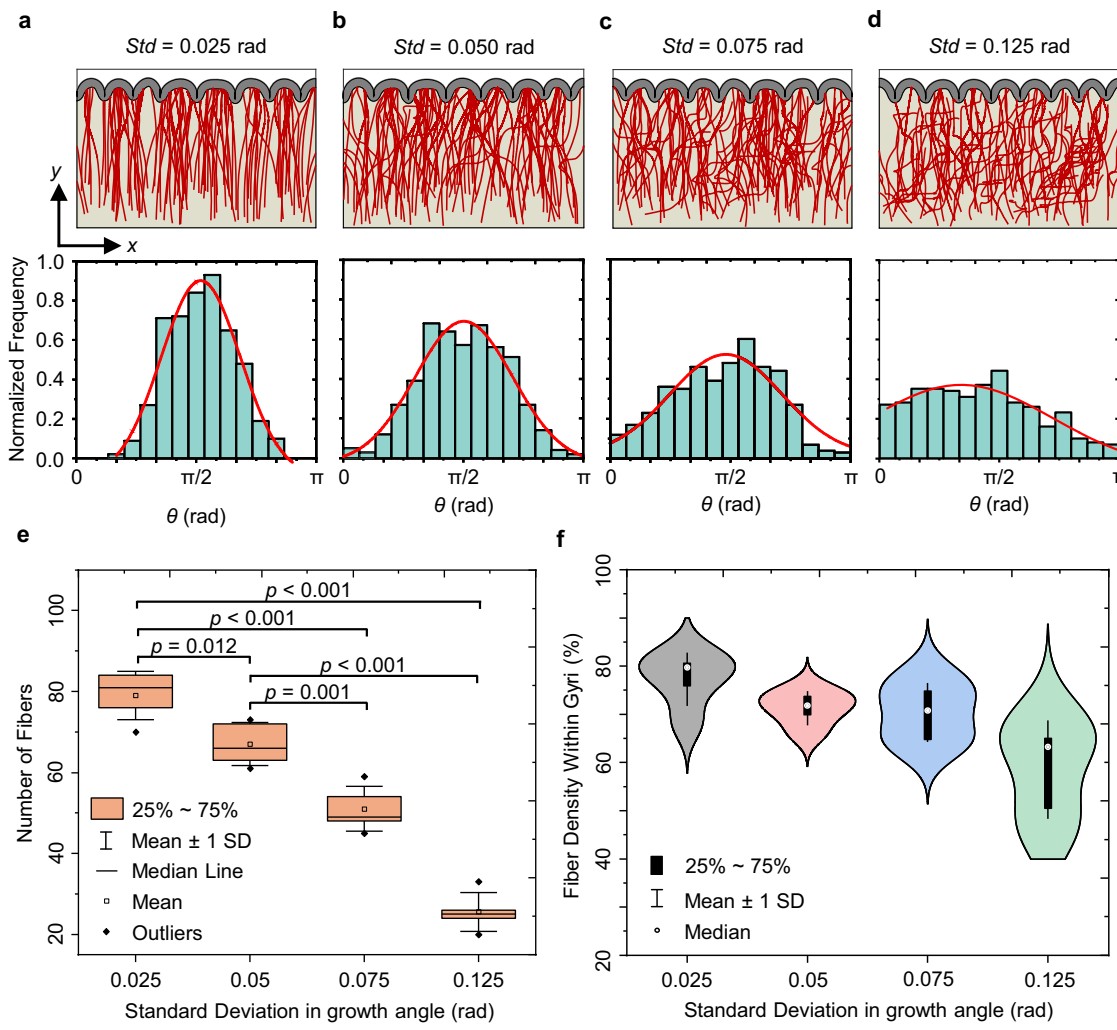

**Fig. 7 | Dependence of fiber organization in the white matter on fiber growth behavior.** Material properties for all scenarios are $\mu_f/\mu_s = 2$, $\mu_c/\mu_s = 2$, $G^{axn} = 0.8$ mm d$^{-1}$, $a = 0.015$ mm Pa$^{-1}$ d$^{-1}$, where $\mu_f$, $\mu_s$, and $\mu_c$ are the shear moduli of the fiber, ECM, and cortex, respectively, $G^{axn}$ is the axon growth rate, and $a$ is the stress-dependent elongation rate. **a–d** The effect of the stochastic nature of fiber growth on the final fiber organization by altering the standard deviation (Std) of the growth angle. Normalized angular distribution for fiber growth has been calculated for different standard deviations. The vertical axis (labeled Normalized Frequency) represents the ratio between the number of fibers that grow in a given direction and the total number of fibers in the model ($N = 100$). All distributions show data calculated at $T = 1.0$. **e** Number of fibers that reach the cortex at $T = 1$ across various standard deviations, which reveals a decrease in successful fiber arrivals to the

cortex with increased randomness in fiber growth. The figure displays p-values, with a significance level of 0.05. Statistical analyses were two-sided, utilizing Tukey's post hoc method for multiple comparisons. The boxes show the interquartile range (IQR), spanning the 25th to 75th percentiles, with the mean shown as a white dot and the median indicated by a black line within the box. The whiskers denote the mean +/- SD, while the black dots show outliers. The analysis was conducted with a sample size of $n = 40$. **f** Effect of randomness on fiber density within gyri. The median is represented by a white dot, and the black box illustrates the interquartile range (IQR), extending the 25th to 75th percentiles. The whiskers indicate the mean +/- SD, with the analysis based on a sample size of $n = 40$. Source data are provided as a Source Data file.

Fig. 8, we hypothesize that the stress-induced growth and reorientation process must synchronize with the intrinsic growth rate of axons to achieve the observed fiber density in gyri in the natural brain. Without incorporating stress-induced growth and reorientation mechanisms, the fiber density in gyri cannot match the observed data from imaging, regardless of the axon growth rate.

The convergence between simulation predictions and imaging data underscores the utility of the model in the elucidation of the underlying mechanisms that govern fiber organization within the developing brain. Therefore, the proposed modeling framework holds potential for interpreting brain development under both healthy and disordered conditions. Identifying and evaluating key mechanical parameters represent a significant step toward elucidating the role of mechanical forces in the formation of cortical folds and their interaction with the underlying neuronal landscape. In the next section, we

explore potential applications of these models to enhance our understanding of the complex processes involved in brain development and folding.

## Discussion

The mechanics of brain folding have recently garnered significant attention from researchers, although brain folding itself has been a fascinating topic for over a century[67]. This surge in interest is attributed to recent rapid advancements in acquiring high-quality brain images, conducting sophisticated experiments, and developing advanced computational modeling techniques. Collectively, recent studies have supported the crucial role of mechanics and physical forces, alongside other genetic and environmental factors, in brain folding processes[21,68–70]. Some of these studies have attempted to incorporate multiple identified key factors in brain folding to elucidate

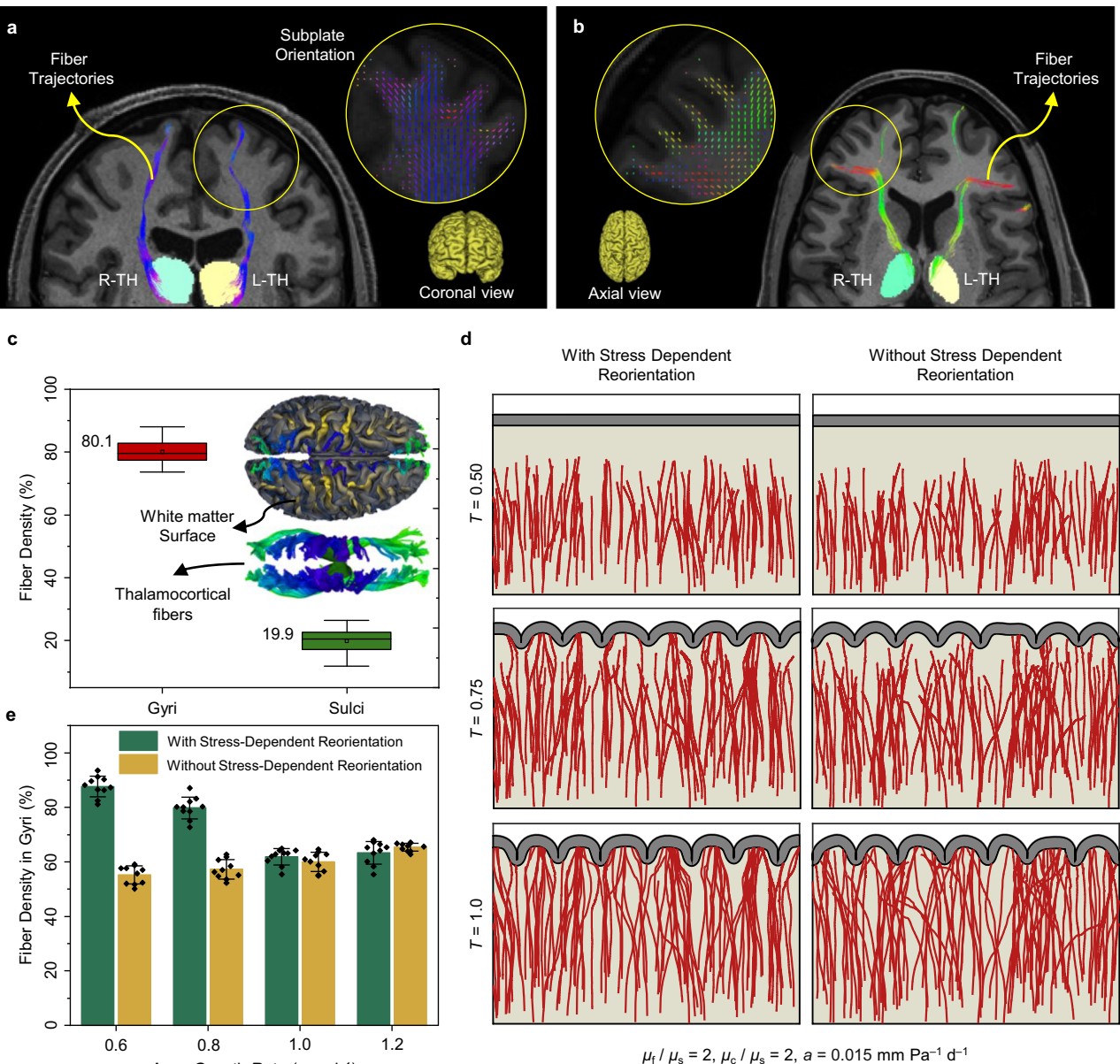

**Fig. 8 | Anisotropic fiber organization in the adult brain and the effect of stress-dependent fiber reorientation on connectivity patterning. a, b** Fiber trajectories and orientation map (enclosed within yellow circles) based on high-resolution DTI imaging for (**a**) coronal view, and (**b**) axial view. As shown in the fiber trajectories, thalamocortical fibers mainly terminate in gyral regions. Within the subplate, fiber orientation beneath the sulci is predominantly tangential, while those within gyri are radial. Observations from subplate orientation suggest that fibers alter their trajectories upon nearing sulcal regions, preferring settlement within gyri. **c** Average fiber density in gyri and sulci of 10 brains for thalamocortical fiber tracts. The boxes represent the interquartile range (IQR), spanning the 25th to 75th percentiles, with the mean shown as a dot and the median indicated by a black line within the box. The whiskers depict the mean +/- SD. The analysis was conducted with a sample size of $n = 10$. Blue areas on the surfaces of the white matter indicate the tips of thalamocortical fibers. For better

clarity, readers are referred to the color version of the figure in the online version of this article. The MRI and diffusion MRI images featured in this figure were sourced from the publicly available Q1 release of the WU-Minn Human Connectome Project (HCP) database (https://humanconnectome.org/). **d** Dynamic growth of fibers for the axon growth rate of $G^{axn} = 0.6\ \text{mm}\ \text{d}^{-1}$ in models with and without stress-induced reorientation process. $\mu_f$, $\mu_s$, and $\mu_c$ are the shear moduli of the fiber, ECM, and cortex, respectively, and $a$ is the stress-dependent elongation rate. **e** Comparison of fiber density within gyri for simulations conducted with and without the stress-dependent reorientation process, across different fiber growth rates. Error bars represent the mean +/- SD, while black dots show individual data points. The analysis was performed with a total sample size of $n = 80$, comprising 40 data points per group. Source data are provided as a Source Data file.

the underlying mechanisms of this complex process, which aims to better align with experimental and imaging data of the human brain. Notably, there has been a particular focus on understanding the mechanistic interplay between the differential tangential growth of the cortical plate and axonal forces in shaping and regulating cortical folds[15,23,24,30,34,71]. Recent reviews have underscored the significance of mechanical forces in axon growth and their role in nervous system

development, homeostasis, and disease[45,72,73]. However, despite the significant contributions of the latest findings, the dynamic physical interplay between the surface morphology of the brain and underlying neuronal map remains not fully understood.

In this work, we proposed and evaluated a mechanical model incorporating the "axon reorientation" mechanism to understand the bidirectional and dynamic interactions between cortical plate growth

and folding and axonal development. We introduced the concept of axon reorientation, inspired by the mechanics of soft fibrous tissues, to elucidate the gradual response of growing axon bundles to the stress field generated by the deforming subplate during cortex folding. The innovative framework proposed here facilitates the integration of complex stress-induced growth behaviors of axons with the dynamics of the folding system. This allows step-by-step simulation of axon bundle growth and navigation within the stress and deformation fields, capturing their mutual interactions as a mechanically coupled system. Therefore, the proposed model offers advantages, by explicitly simulating dynamically growing axon bundles whose lengths and orientations are not predefined, over recent continuum stretch-driven models[15,23,30]. In the proposed model, other factors, such as patterned growth in the cortical plate and heterogeneous material properties of white matter ECM, are included, but they are not discussed in this study to reduce the parameters under investigation. However, this comprehensive model can be used to assess the impact of multiple identified key parameters on brain folding and axonal development concurrently.

## Axonal fibers concentrate more in gyri than sulci

The imaging results in this study, which focuses on fiber tracts that project from the brain core to the cortical plate (e.g., thalamocortical tracts), reveal a notable concentration of fibers in gyri compared to sulci. Remarkably, this observation holds true for fiber tracts that spread from the cortical plate toward the brain core and adjacent regions as well. Our previous studies across all fiber tracts consistently indicate a higher density of fibers in gyri than in sulci, a trend observed in both macaque and chimpanzee brains[13,34,74]. This finding aligns with the notion that complex functional regions are predominantly located in gyral areas rather than in sulcal ones, given the strong association between structural and functional connectivity[75,76]. In Fig. 8c, the average fiber density in gyri across 10 randomly selected cases of healthy development is four times that of the fiber density in sulci. There are only small variations in fiber density observed in both gyri and sulci among cases, which suggests that this trend is expected to persist in a typically-developed individual brain. However, it should be noted that while fiber tractography is a powerful tool for visualizing white matter pathways[77,78], it has well-known limitations, particularly in accurately capturing the complexity of fiber orientations. It tends to bias toward detecting radial fibers in gyri, which can be problematic in regions of the cerebral cortex with intricate folding and convolutions[79]. While our study used tractography-based data for model comparison, we recognize that this method may not fully represent the complex organization of axonal pathways. However, despite these limitations, recent integrated tractography and histological tract-tracing studies support its practical performance in estimating structural connectivity[80–82].

## Axonal growth in a stress field

In general, the genome-wide epigenetic landscape coordinates precise spatiotemporal axon elongation, membrane expansion, navigation toward specific targets, branching, and synapse formation. We modeled this directed and controlled growth (tip growth) by constraining the growth angle in the preferred direction of axon growth. This aspect of growth guides axons in the absence of external forces from the core of the model (or brain) toward the cortex, which mimics the targeted growth of thalamocortical fiber tracts. Another growth mechanism, known as stress-induced growth (towed growth), was applied to growing axons to account for the effect of stress from the deforming white matter in accelerating axon growth. We focused solely on mechanical cues to redirect axons and control their directionality. However, the directionality of axons is influenced by a complex interplay of both chemical[83] and mechanical cues[42,44]. We assumed that

the contributions of non-mechanical cues are encompassed in the tip growth of the axons.

Our findings indicate that cortical folding arises from differential tangential growth, which creates a stress and deformation landscape that guides fibers and alters their growth rates (Fig. 2). Cortical folding generates tensile and radially oriented principal stresses beneath prospective gyri, while tangential and tensile principal stresses occur beneath prospective sulci. Essentially, fibers beneath gyri encounter tensile forces favorable to their continued growth and reorientation. Conversely, fibers beneath sulci experience radial compressive forces, which are unfavorable for their growth and prompt them to alter their growth direction. This behavior has been observed in in vitro experiments, wherein growing axons on a soft substrate, upon reaching the vicinity of the tip of a stiff probe, change their direction and reorient to circumvent the stiff obstruction[42]. In addition, models incorporating axon reorientation offer a potential mechanism to explain the observation in Fig. 8, where thalamocortical fibers shift their trajectories toward gyri as they approach sulcal regions, a phenomenon that has not been explored before.

Stretching toward gyri creates pulling forces in axons, which agrees with experimental measurements that show axons are under tension from early growth and even after connections are established[33,45,84,85]. This also aligns with the observed tension directions in the developing ferret brain[33], and the alignment of primary eigenvector directions of fibers in DTI data of rhesus macaques[30]. Our findings are closely aligned with those of Garcia et al.[30], demonstrating that an expansion-driven model of cortical folding generates sub-cortical stresses capable of explaining the anisotropy patterns observed in the developing white matter during brain folding. However, in their model, the subplate is treated as a continuous medium, with its reorganization and anisotropy driven only by stress from cortical folding. This approach overlooks the impact of subplate development on the formation and positioning of cortical folds. Our findings suggest that the heterogeneity of white matter, resulting from the gradual growth of individual stiff fibers, influences the folding patterns and dictates the location of gyri and sulci, even though the fibers do not directly induce cortical folding. This leads to the formation of gyri in areas with dense fibers and sulci in regions with lower fiber density. The bidirectional and dynamic relationship between cortical gyrification and connectivity development, highlighted by our results, has not been fully explored in previous studies.

The findings in this study are also consistent with those of Bayly et al.[23] which predicted that slower fiber elongation (or faster cortical growth) increases gyrification (Fig. 3b), while faster subplate response relative to cortical growth leads to decreased folding (Fig. 3a). These observations can be linked to polymicrogyria disorder, characterized by many small folds, and lissencephaly disorder, characterized by fewer but larger folds, both of which also exhibit alterations in short- and long-range connections. In addition, the results agree that a reduction in fiber density increases gyrification. We speculate that this observation results from the dynamic increase in the stiffness of the white matter, which alters the material properties of the substrate for cortex folding (Fig. 5). As established in the mechanics of film-substrate buckling, changes in substrate stiffness affect the critical strain, wavelength, and wave amplitude of buckling[86,87]. This may be a potential reason for the observation that early disruption of underlying connectivity, such as reduced density or reduced growth rate, results in an overproduction of cortical folds[88,89]. However, it is important to acknowledge that the development of fiber tracts in the brain varies significantly in terms of timing, location, length, and their interaction with the cortical plate. Therefore, there is no universally applicable relationship between cortical folding and connectivity. For instance, studies have demonstrated that reduced interhemispheric connectivity is associated with decreased gyrification in ASD[36], which

contrasts with our previous observation. Therefore, extrapolating unique observations regarding the relationship between cortical folding and connectivity from specific observations in one region of the cortical plate and fiber tract, or from one disorder to another, may result in misleading conclusions.

## Effect of axon biophysical properties on fold placement and fiber organization

The precise timing and spatial distribution of axonal growth are essential for shaping the intricate folding patterns of the cerebral cortex and establishing functional neuronal networks. Our model of axonal growth in cortical folding involves the interplay of deterministic and stochastic components in cortical development. Deterministic factors arise from the physical interactions between axons and the folding cortex, while stochastic components are related to biochemical and other contributing factors (tip growth). The nature of biochemical cues is unpredictable; therefore, employing a probabilistic method becomes essential to predict axonal growth patterns across a population[90]. Randomness plays a crucial role in analyzing stochastic systems. Increased randomness in growth angle, illustrated in Fig. 7, impedes axon pathfinding in the folding system, preventing axons from reaching the cortex on time, even when mechanical stimuli are present. The precise guidance of axonal growth cones toward target regions is regulated by a complex interplay of cytoskeletal mechanisms, receptor arrays, cell adhesive molecules, and transsynaptic signals. Any disruptions to these factors can potentially reduce the pathfinding ability of axons[91].

The degree of randomness in axonal growth can be experimentally altered through chemical treatments[48]. Notably, substances such as Taxol are known to suppress microtubule dynamics, and Blebbistatin, which disrupts actin polymerization, affect the cytoskeletal structure of neurons. These alterations in the cytoskeleton, in turn, modify the traction forces between neurons and their growth substrate, thereby increasing the randomness in axonal growth. This finding underscores the sensitivity of axonal pathfinding to chemical modulation and the potential for such interventions to influence the development of neural connectivity. There is theoretical evidence from previous studies[92] that shows that disturbances in connectivity and neural dynamics might contribute to brain disorders, supporting the notion that randomness in axonal reorientation could be linked to pathological states.

In this study, the growth rates of axons were derived from multiple studies that encompass various species and experimental conditions. We conducted a parametric investigation into the impact of axon growth rate on cortical folding and fiber organization. Our findings highlight the critical role of axon growth rate, both in the presence and absence of stress, in determining the distribution of fibers within gyri and sulci. Slowly extending fibers reach the cortex after the initiation of cortical folding and therefore experience the stress field of folding earlier than rapidly growing fibers, which arrive at the cortex before the initiation of cortical folding. Consequently, the stress field has sufficient time to redirect a greater portion of slow-growing fibers toward gyri compared to fast-growing fibers. Nevertheless, regardless of the growth rate of axons, the fiber density in gyri remains greater than in sulci (Fig. 8e). The observed results align with the findings of Bayly et al.[23], which suggest that the formation and characteristics of cortical folds depend on the rate of cortical growth relative to the stress-induced growth of the underlying layer. Accordingly, the temporal coordination between cortical folding and axonal development, which depends on both cortical and axonal growth rates, emerges as a crucial parameter in the normal development of the human brain.

Most of the studies measured or predicted larger stiffness of axons compared to their surrounding substance as ECM[93–95]. Differences in stiffness ratios vary significantly depending on the test methods or modeling assumptions. However, a recent study predicts that there

should be no significant difference between axon and ECM material properties[96]. In our recent study[93], we inversely predicted the independent mechanical properties of axonal fibers and ECM based on six previously reported experimental mechanical tests for bulk white matter tissue from the corpus callosum[64]. Our results predicted that axons are approximately five times stiffer than ECM and the non-linearity degree of axons is higher than ECM. The findings shown in Fig. 4 indicate that an increase in the stiffness ratio between fibers and ECM leads to a higher fiber density within gyral regions and a concurrent decrease in the gyrification index. Even under a condition where there is no difference in stiffness between fibers and ECM (i.e., isotropic substrate), fibers still exhibit a greater tendency to settle in gyral areas compared to sulcal regions, which suggests that fiber stiffness alone is not the sole determining factor. However, in this condition, the fiber density within gyri is lower compared to observations in Fig. 8c obtained from imaging data. Elevating the stiffness ratio ($\mu_f/\mu_s$) twofold results in a fiber density within gyri comparable to that observed in Fig. 8c. Therefore, it seems that the fiber stiffness regulates fiber density within both gyral and sulcal regions, and to replicate the imaging observations, the stiffness of fibers should theoretically exceed that of ECM by at least twofold. Beyond a twofold increase in the stiffness ratio of fibers to ECM, no significant effects on fiber density within gyral and sulcal regions are observed. These results align with those reported by Holland et al.[15] and Wang et al.[58] which confirm the important role of axonal fibers as both symmetry breaker and regulator of folding patterns. They showed that the inclusion of passive axon bundles can influence fold placement, and that altering the material properties of multiple axon tracts can significantly change folding patterns[58]. We have previously demonstrated that, in the context of strain energy minimization, a cortex on a substrate with predefined passive radial fibers tends to buckle, which results in a concentration of fibers in gyri rather than sulci[34]. However, in models with passive stiff fibers, even under optimal conditions, the fiber density in gyri and sulci remains only about 60% and 40%, respectively, assuming the removal of fold walls and the redistribution of fibers between gyri and sulci. Therefore, recent models with passive fibers[15,24,34,58] or dynamic continuum-based fibers[30] fail to realistically model the growth, navigation, and establishment of axonal connectivity.

Analysis of cortical development in various species with gyrified brains (e.g., ferret, cat, human) has demonstrated that regional variations in progenitor cell proliferation is a potential factor in determination of the spatial locations of primary folds and fissures. Areas exhibiting increased rates of proliferation undergo substantial expansion, maturing into primary folds, particularly in the outer subventricular zone (oSVZ). In contrast, regions characterized by lower proliferation rates experience comparatively limited expansion, resulting in the development of fissures[97,98]. Our findings from Figs. 4 and 5 indicate that the distribution and concentration of stiff fibers beneath the cortex play analogous roles in the formation of gyri and sulci, in which gyri form in regions of higher axon density and sulci in regions of lower density. Regions with lower fiber density provide a favorable substrate for cortical buckling and sulci formation, while those with higher density resist downward deformation and instead buckle outward. This potential mechanism may contribute to the localization of secondary and tertiary folds during brain folding following the formation of primary folds and fissures, which warrants further investigation. However, to gain a deeper understanding of how fibers contribute to the mechanical properties of the brain, it is essential to conduct more comprehensive mechanical property characterizations at both the microscale (e.g., radial glial cell and axon) and tissue scale (e.g., white matter).

## Effect of cortex stiffness on fiber organization

Increasing the stiffness ratio of the cortex to the ECM of white matter region holds significant implications for the redirection of fibers

toward gyri, as shown in Fig. 6. It is intriguing to speculate that achieving the observed fiber density in gyri necessitates a cortex stiffness surpassing that of the underlying layer, which is in agreement with an in vivo experimental test on brain tissue[64]. Moreover, it is plausible that the stiffness ratio may be even more pronounced in the fetal brain compared to the mature brain, given the known increase in white matter stiffness with myelination[99]. While the cortex, with its densely packed neuronal cell bodies and synapses, intuitively suggests greater mechanical stiffness than the white matter, the actual difference is less than an order of magnitude, as evidenced by various studies[64,65,100]. However, the extent to which ex vivo experiments can accurately estimate the material properties of the living brain in vivo remains uncertain. Undoubtedly, further investigation into the dynamic and longitudinal stiffness ratio between the cortex and the subcortex is warranted.

The fibers that grow with a higher shear modulus than the ECM dynamically alter the stiffness of white matter region, gradually increasing it as they transform ECM elements into fiber elements (Fig. 5b and d). This process also serves to reduce the difference between the stiffness of the cortex and white matter region. This phenomenon may explain why the contrast in stiffness between gray and white matter is more pronounced during early brain development compared to the mature state[1]. In a bilayer system such as the brain, where the stiffness of the gray matter layer surpasses that of the white matter substrate, the system is more prone to buckling rather than forming creases[101]. This observation aligns with experimental and imaging studies that indicate the formation of folds, which include secondary folds such as period-doubling, during brain development. It can be inferred from Fig. 6 that increasing the stiffness ratio between the cortex and the ECM of white matter region decreases the critical strain necessary for the onset of buckling within the system. Consequently, folds form before fibers reach the cortical plate, allowing the stress field induced by cortical buckling to redirect these fibers. This finding underscores how alterations in the material properties of the folding cortex, such as those that result from disorders, can profoundly impact the underlying neuronal map and organization.

## Limitations and potential improvements

This study, like other computational simulations, entails inherent simplifications and limitations that can be addressed and improved to better understand the correlation between axonal development and cortical folding. The following limitations provide valuable avenues for future research.

The main limitation of this study might be the utilization of parameter magnitudes that have been extracted from various sources, which may not be directly applicable to the context of cortical folding, because many were obtained from basic experiments. However, we attempted to address this limitation by conducting parametric studies aimed at quantifying the sensitivity of the observations to the parameters used. It is essential to conduct in vivo or in vitro studies to determine the reorientation behavior and growth rate of axons within the stress or deformation fields of their substrate. This requirement is also relevant for understanding the material properties of growing axons in the brain, because there exists a notable lack of comprehensive data in this area.

In this study, we considered isotropic material properties for the white matter ECM. However, the white matter exhibits spatially heterogeneous stiffness due to changes in material composition across different regions, resulting in a varied stiffness map[102]. This heterogeneity in substrate rigidity can influence the growth direction of axon bundles[44], as they tend to avoid stiff regions[42] despite exhibiting higher growth rates within those regions[103]. In the proposed FE model, because the ECM has been discretized into small elements, any element can be dynamically assigned a different material property. This approach provides the opportunity to conveniently map the stiffness landscape of the brain, as characterized in vivo or in vitro, onto the ECM.

The 2D models in this study served as proof of concept, which offers simplicity in explaining the underlying mechanisms involved in the interplay of cortical folding and connectivity. However, given that cortical folding and connectivity development occur in the 3D space of brain development, 3D models can offer more accurate representations of this complex process. Therefore, future studies should consider utilizing 3D models, because they provide better opportunities to compare results, especially regarding the 3D convolution of folds with imaging data. This aspect will be investigated further in our future studies.

In addition to radial fibers, tangential fibers (short- and long-association fibers), also play a role in cortical organization and connectivity[58]. These fibers differ in orientation and function from radial fibers, contributing to the complex architecture of the brain. The introduction of tangential fibers would necessitate a more intricate organization of fibers within the model, which could obscure the specific mechanisms underlying radial fiber growth and its interaction with the folding cortex. Therefore, tangential fibers were excluded from our models for simplicity, allowing us to focus on extracting meaningful statistical correlations between the parameters used. However, incorporating tangential fibers would provide a more comprehensive representation of axonal growth and connectivity. Extending our model to a true-scale representation that includes these additional fiber types will be the focus of our future research.

In our model, we treated the growing fibers as axon bundles that extend and navigate through the ECM from the brain's core toward the cortical plate, including thalamocortical fiber tracts. However, it is important to acknowledge that the composition and mechanical properties of the layer underlying cortical plate undergo a dynamic and complex evolution during the process of cortical folding. In the early stages of brain development, the subplate evolves from a loosely structured layer of cell bodies, radial glial scaffolds, and neuronal processes into a more organized, axon-dense white matter tissue[104]. It is also important to note that very little data is available regarding the mechanical response of radial glial cells, which are eventually eliminated as they differentiate into other cell types during cortical folding, warranting further investigation.

For future investigations, we speculate that achieving typical brain folding relies on the orchestrated interplay of mechanical cues alongside other factors. Any perturbations in mechanical factors might disrupt the intricate connections between folding patterns and developmental connectivity. We hypothesize that the absence of physical forces generated during cortical plate folding could hinder the attainment of the observed fiber density in gyri, particularly in higher-order functional areas. Such disruptions could subsequently impede normal connectivity processes.

In summary, this study proposed and evaluated a mechanical model incorporating the concept of "axon reorientation" to understand how growing axons navigate in a folding system, which simulates the interaction between cortical folding and axonal development (brain connectivity) in a developing brain. The results confirm the vital role of mechanical factors in determining cortical folding, connectivity development, and their bidirectional relationship. We studied key parameters such as the growth rate, stochastic behavior, density, and mechanical properties of axon bundles, as well as the stiffness ratio of cortex to white matter, to understand how axon bundles respond to the physical forces of the folding system and navigate within a complex stress field. The results indicated that axon bundles destined to grow from the core of the brain toward the cortical layer potentially experience stress-induced growth and reorientation, which settle more in gyri than sulci. This observation arises from the stress field of the white matter induced by cortex folding. Additionally, the biophysical properties of axons can alter the gyrification index and surface

morphology of the folds. Developing axon bundles dynamically change the material properties of the white matter, which provide a heterogeneous substrate for cortex folding. This heterogeneous stiffness map may also help identify the locations of prospective secondary and tertiary gyri and sulci. The bidirectional linkage underscores the significance of mechanics in the healthy development of the human brain during the early stages. Future in vivo or in vitro studies are essential to provide feedback regarding the proposed model and its assumptions, limitations, and predictions. This will open new avenues to understand the origins of linked abnormal cortical folding and connectivity disruptions in brain disorders.

## Methods

### Constitutive framework

To model the growth and folding of the human brain, we consider the well-supported DTG mechanism as the driving force of brain folding. In DTG, the differential expansion rates between the rapidly growing cortical plate and the comparatively slower subcortical region induce compressive forces, which lead to mechanical instability and eventual buckling of the cortical plate[7]. The cortex is formed by folded gray matter, which comprises primarily neuronal cell bodies, while the inner subcortical core of white matter is primarily composed of neuronal axons. It should be noted that the properties of the layer underlying the cortical plate evolve over time, particularly during the cortical folding process, which spans several months. During this period, the subplate, initially composed of radial glial cells, transforms into white matter, which includes growing and establishing axons. Our model focuses on the growth of axon bundles from thalamocortical fiber tracts, which predominantly develop during the second and third trimesters, aligning with the onset of cortical folding (Table S1). Therefore, in our model, we represent the growing fibers as axon bundles rather than radial glial cells. Nonetheless, from a modeling perspective, both axons and radial glial cells, can be treated as fibers, each possessing distinct material properties relative to the extracellular matrix (ECM)[30]. Considering this distinction, unique constitutive equations are essential for each constituent, given the differences in composition and growth mechanisms of the cortex, axons, and white matter ECM.

**Multiplicative decomposition of the deformation gradient.** We consider a continuum body, denoted as $\mathcal{B}_R$, which is associated with the region of space it occupies in a fixed reference configuration. Within this body, there are arbitrary material points represented as $\mathbf{X}$. As time progresses, the referential body $\mathcal{B}_R$ experiences a motion described by $\mathbf{x} = \varphi(\mathbf{X}, t)$, which leads to the formation of the current deformed body, denoted as $\mathcal{B}_t$. The deformation gradient

$$\mathbf{F} = \nabla_{\mathbf{X}} \varphi(\mathbf{X}, t) \tag{1}$$

captures the transformation undergone by the body during this process. The symbol $\nabla$ represents the gradient operator applied with respect to the material point $\mathbf{X}$ in the reference configuration.

In the field of continuum mechanics, the noticeable deformation of biological tissue can be interpreted as a combination of two mechanisms: growth, the deformation due to increase in size or number of cells and cell processes, and elastic deformation, which is caused by mechanical forces[105]. Then, the total deformation gradient can be expressed as follows

$$\mathbf{F} = \mathbf{F}^{e} \cdot \mathbf{F}^{g} \tag{2}$$

where $\mathbf{F}^{e}$ and $\mathbf{F}^{g}$ are elastic and growth deformation gradient tensors, respectively. This decomposition enables us to account for both elastic $\mathbf{F}^{e}$ and irreversible growth-related $\mathbf{F}^{g}$ components, which provide a comprehensive understanding of the overall deformation experienced by the body.

**Constitutive model for cortex.** The imaging data indicate that cortical thickness undergoes changes during the early developmental stage[106]. However, in the later stages, it is primarily the alteration in surface area that initiates cortical folding. These findings suggest that the growth of the cortex primarily occurs tangentially, as supported by several studies[20,21]. Therefore, we adopted a model for cortical growth that considers in-plane area expansion, while it assumes purely elastic behavior in the direction normal to the cortical layer, following the approach proposed by Holland et al.[15].

$$\mathbf{F}_{ctx}^{g} = \sqrt{\vartheta_{ctx}^{g}} \mathbf{I} + \left(1 - \sqrt{\vartheta_{ctx}^{g}}\right) \mathbf{n}_0 \otimes \mathbf{n}_0 \tag{3}$$

Equation (3) incorporates a growth multiplier $\vartheta_{ctx}^{g}$, the referential unit normal $\mathbf{n}_0$ of the pial surface, second order unit tensor $\mathbf{I}$, and accounts for the increase in the cortical area. The growth multiplier $\vartheta_{ctx}^{g}$ can be expressed as follows

$$\vartheta_{ctx}^{g} = \det(\mathbf{F}_{ctx}^{g}) \tag{4}$$

A linear kinetic model for the growth of the cortex (gray matter) was used[15]

$$\dot{\vartheta}_{ctx}^{g} = G^{ctx} \tag{5}$$

where $G^{ctx}$ is the cortical growth rate. Then the elastic deformation gradient can be explicitly calculated by the following equation

$$\mathbf{F}_{ctx}^{e} = \mathbf{F}_{ctx} \cdot \mathbf{F}_{ctx}^{g-1} = \frac{1}{\sqrt{\vartheta_{ctx}^{g}}} \mathbf{F}_{ctx} + \frac{\sqrt{\vartheta_{ctx}^{g}} - 1}{\sqrt{\vartheta_{ctx}^{g}}} \left(\mathbf{F}_{ctx} \cdot \mathbf{n}_0\right) \otimes \mathbf{n}_0 \tag{6}$$

Lastly, the elastic left ($\mathbf{b}^{e}$) and right ($\mathbf{C}^{e}$) Cauchy-Green tensors can be derived from $\mathbf{F}_{ctx}^{e}$, respectively

$$\mathbf{b}^{e} = \mathbf{F}_{ctx}^{e} \cdot \mathbf{F}_{ctx}^{et} \tag{7}$$

$$\mathbf{C}^{e} = \mathbf{F}_{ctx}^{et} \cdot \mathbf{F}_{ctx}^{e} \tag{8}$$

**Constitutive model for white matter.** In contrast to the previous studies to model the white matter as a stretch-induced axonal growth media[15,71], we separated the white matter into two distinct parts as ECM and axon. This separation between ECM and axon provides the opportunity to model the growth and reorientation of individual axon bundles explicitly inside a growing and deforming ECM. During the developmental process, there is an observed increase in the volume of white matter[15]. Therefore, we consider that the growth of the ECM is isotropic. The growth tensor is

$$\mathbf{F}_{sub}^{g} = \vartheta_{sub}^{g}{}^{1/3} \mathbf{I} \tag{9}$$

where $\mathbf{I}$ denotes the second order unit tensor and $\vartheta_{sub}^{g}$ is the growth parameter represents the increase in the volume of the ECM

$$\vartheta_{sub}^{g} = \det\left(\mathbf{F}_{sub}^{g}\right) = J^{g} \tag{10}$$

The growth rate of the ECM is defined as follows:

$$\dot{\vartheta}_{sub}^{g} = G^{sub} \tag{11}$$

Similar to the previous brain folding models[19,30,34,64], we used a standard neo-Hookean hyperelastic material model with the defined

free energy function in Eq. (12) for the ECM, as well as the cortex and, later, for axons, though with different shear moduli

$$W = \frac{\mu}{2}\left(J^{e-\frac{2}{3}}\mathrm{tr}\left(\mathbf{F}_{\mathrm{sub}}^{\mathrm{eT}} \cdot \mathbf{F}_{\mathrm{sub}}^{\mathrm{e}}\right) - 3\right) + \frac{k}{2}\left(J^{e} - 1\right)^2 \quad (12)$$

where $\mu$ is the shear modulus, $k$ is the bulk modulus, and $J^{e}:$ $= \det(\mathbf{F}_{\mathrm{sub}}^{\mathrm{e}}) > 0$ is the Jacobian. We introduce the Cauchy stress tensor of the ECM generated by the elastic deformation tensor in the material configuration as follows

$$\mathbf{T} = J^{e-1}\mathbf{F}_{\mathrm{sub}}^{\mathrm{e}} \cdot \frac{\partial W}{\partial \mathbf{F}_{\mathrm{sub}}^{\mathrm{eT}}} = \sum_{i=1}^{3}\sigma_i \mathbf{n}_i \otimes \mathbf{n}_i \quad (13)$$

where $J^{e}$ is the elastic volume change ratio, $\mathbf{n}_i$ that denotes the principal stress direction, and $\sigma_i$ is the principal stress value. In this contribution, we arrange the principal stresses in descending order of magnitude, such that the index $i$ increases as the principal stress decreases, specifically

$$\mathbf{n}_i^{T} \cdot \mathbf{n}_j^{T} = \delta_{ij}, \quad \sigma_1 > \sigma_2 > \sigma_3 \quad (14)$$

**Real-time axon reorientation and growth.** During the brain developmental period, axons initiate and group themselves into bundles and grow along precise pathways in search of their functional target to form connections between various regions[44]. Soft fibrous tissues such as the brain white matter exhibit a distinct alignment of fibers along specific directions[107]. This phenomenon typically develops upon mechanical loading[44]. For soft fibrous tissues, it has been shown that fibers tend to reorient along the principal stress directions[108,109]. Borrowing this idea, we propose a method that enables axon bundles to reorient and grow in alignment with the maximum tensile principal stress of the material.

We conceptualize the axon bundle as an integral part of the white matter, alongside the ECM. Therefore, we consider a material point at the position $\mathbf{X}$ in the reference configuration of white matter at time $t_k$, that signifies the initiation location of bundle that has a preferred direction, $\mathbf{n}_k^{A}$, in the material configuration. The fundamental concept of the current model is that the unit vector of the preferred direction, $\mathbf{n}_k^{A}$, is allowed to gradually align with the maximum tensile principal stress direction, $\mathbf{n}_{\mathrm{max},k}^{T}$, of the Cauchy stress tensor, $\mathbf{T}$, of the ECM, which is related to the maximum principal stress $\sigma_1$ in Eq. (14)

$$\mathbf{n}_k^{A} \rightarrow \mathbf{n}_{\mathrm{max},k}^{T} \quad \text{for } \sigma_1 \geq \sigma_2 \geq \sigma_3 \text{ and } \sigma_1 \geq 0 \quad (15)$$

We assume that the axonal reorientation is influenced solely by positive principal stresses. Following the methodology proposed by Menzel et al.[109], we introduce the rotation vector $\boldsymbol{\omega}_k$

$$\boldsymbol{\omega}_k = \frac{\pi}{2t^*}\mathbf{n}_k^{A} \times \mathbf{n}_{\mathrm{max},k}^{T} \quad (16)$$

whereby the parameter $t^* > 0$ acts like a relaxation parameter. The magnitude of rotation is determined by both the relaxation parameter $t^*$ and the angle between $\mathbf{n}_k^{A}$ and $\mathbf{n}_{\mathrm{max},k}^{T}$. For equal principal stresses, specifically when $\sigma_1 = \sigma_2 > \sigma_3$ or $\sigma_1 = \sigma_2 = \sigma_3$, any stress-induced reorientation of the bundle direction is disregarded. The rotation vector $\boldsymbol{\omega}_k$ can be decomposed into the unit normal vector $\mathbf{n}_k^{\omega}$ and the magnitude $\omega_k$

$$\begin{aligned} \boldsymbol{\omega}_k &= \omega_k \mathbf{n}_k^{\omega} \\ \omega_k &= \frac{\pi}{2t^*}\left\| \mathbf{n}_k^{A} \times \mathbf{n}_{\mathrm{max},k}^{T} \right\| \\ \mathbf{n}_k^{\omega} &= \frac{\mathbf{n}_k^{A} \times \mathbf{n}_{\mathrm{max},k}^{T}}{\left\| \mathbf{n}_k^{A} \times \mathbf{n}_{\mathrm{max},k}^{T} \right\|} \end{aligned} \quad (17)$$

As illustrated in Fig. 9a, the evolution of the preferred direction $\frac{\partial \mathbf{n}_k^{A}}{\partial t}$ can be defined as a rotation about the axis $\boldsymbol{\omega}_k$. Therefore, the variation of the preferred direction over time is expressed as follows

$$\frac{\partial \mathbf{n}_k^{A}}{\partial t} = \boldsymbol{\omega}_k \times \mathbf{n}_k^{A} \quad (18)$$

Upon substituting Eq. (17) into Eq. (18), we obtain

$$\frac{\partial \mathbf{n}_k^{A}}{\partial t} = \frac{\pi}{2t^*}\left[\mathbf{I} - \mathbf{n}_k^{A} \otimes \mathbf{n}_k^{A}\right] \cdot \mathbf{n}_{\mathrm{max},k}^{T} \quad (19)$$

The rate of change of the preferred direction is determined by the part of $\mathbf{n}_{\mathrm{max},k}^{T}$ that is normal to $\mathbf{n}_k^{A}$, and this is weighted by the constant scalar $\frac{\pi}{2t^*}$. Consequently, the reorientation is directly proportional to the sinus of the angle between $\mathbf{n}_k^{A}$ and $\mathbf{n}_{\mathrm{max},k}^{T}$. In this study, we assume that the magnitude of the maximum principal stress does not have any effect on the reorientation process. Upon the reorientation of the preferred direction toward the maximum tensile principal stress direction, the axon bundle initiates growth in the newly adopted direction (Fig. 9b).

**Discrete model of axon reorientation.** To integrate the reorientation of the preferred direction vector into a finite element (FE) model, we developed a discrete model to solve the evolution of Eq. (19). Additionally, we introduced centered Gaussian noise into $\mathbf{n}_k^{A}$ to account for the stochastic nature of the axon growth direction, which will be discussed further later.

$$\mathbf{n}_k^{A} = \cos(\theta + \epsilon)\mathbf{e}_x + \sin(\theta + \epsilon)\mathbf{e}_y \quad (20)$$

where $\theta$ is the angle between $\mathbf{n}_k^{A}$ and the $x$-axis, $\epsilon$ represents the Gaussian noise, and $\mathbf{e}_x$ and $\mathbf{e}_y$ are the unit vectors in the $x$- and $y$-axes, respectively. We consider a finite time interval $[t_k, t_{k+1}]$ within a total time $T$, and refer to the time increment as $\Delta t = t_{k+1} - t_k$. We assume that the preferred direction's vector is known at time $t_k$. The integration for $\mathbf{n}_k^{A}$ is based on an exponential scheme, namely

$$\mathbf{n}_{k+1}^{A} = \exp\left(-\Delta t \boldsymbol{\varepsilon} \cdot \boldsymbol{\omega}_k\right) \cdot \mathbf{n}_k^{A} = \mathbf{R}_k \cdot \mathbf{n}_k^{A} \quad (21)$$

where $\mathbf{n}_k^{A}$ and $\mathbf{n}_{k+1}^{A}$ are the preferred directions at $t_k$ and $t_{k+1}$, respectively, and $\boldsymbol{\varepsilon}$ is the third-order permutation symbol. The proper orthogonal tensor $\mathbf{R}_k$ is defined by the following closed-form expression

$$\mathbf{R}_k = \cos(\Delta t \omega_k)\mathbf{I} - \sin(\Delta t \omega_k)\boldsymbol{\varepsilon} \cdot \mathbf{n}_k^{\omega} + \left[1 - \cos(\Delta t \omega_k)\right]\mathbf{n}_k^{\omega} \otimes \mathbf{n}_k^{\omega} \quad (22)$$

The combination of Eqs. (21) and (22) yields the following formula for the current direction $\mathbf{n}_{k+1}^{A}$

$$\mathbf{n}_{k+1}^{A} = \cos(\Delta t \omega_k)\mathbf{n}_k^{A} + \sin(\Delta t \omega_k)\mathbf{n}_k^{\omega} \times \mathbf{n}_k^{A} + \left[1 - \cos(\Delta t \omega_k)\right]\left[\mathbf{n}_k^{\omega} \cdot \mathbf{n}_k^{A}\right]\mathbf{n}_k^{\omega} \quad (23)$$

Then, the rotation vector $\boldsymbol{\omega}_k$ can be updated by assuming that $\mathbf{n}_k^{A} \approx \mathbf{n}_{k+1}^{A}$

$$\boldsymbol{\omega}_k = \frac{\pi}{2t^*}\mathbf{n}_k^{A} \times \mathbf{n}_{\mathrm{max},k}^{T} \quad (24)$$

To account for the inherent randomness of axon growth, we introduced Gaussian noise (Eq. (20)) into the preferred direction, enabling stochastic variations. This randomness in the growth direction is applied to the tip growth of the axon bundle, which is further discussed in the next section. Consequently, in the absence of a stress

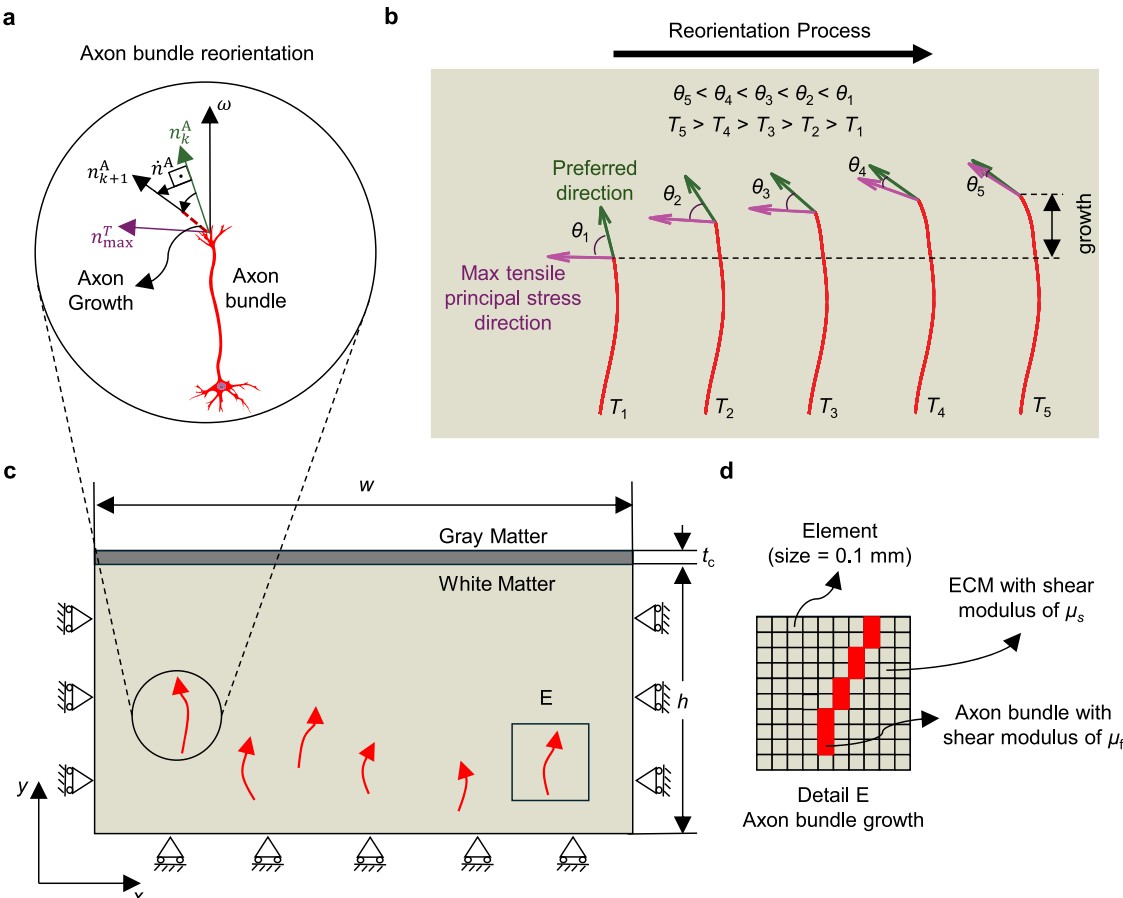

**Fig. 9 | Proposed model to link the growth of axon bundles with the stress and deformation fields of the white matter ECM. a** Evolution of preferred direction $\mathbf{n}_k^A$ of a growing axon bundle with the angular velocity $\boldsymbol{\omega}$. **b** Reorientation process of an axon bundle over time from $T_1$ to $T_5$. As time increases, the preferred direction gradually aligns more closely with the maximum tensile principal stress direction. **c** The geometry and parameters of the finite element model ($w = 60$ mm, $h = 30$ mm, $t_c = 1.5$ mm). **d** Growth and navigation of an axon bundle through the elements of the ECM, and their transformation into the material properties of the axon (red elements).

field, or when the maximum principal stress is compressive, the axon bundle grows randomly. However, in the presence of a tensile stress field, the axon bundle gradually aligns with the direction of the maximum tensile principal stresses. A summary of the entire algorithm for the reorientation of axon bundle is presented in Table S4.

**Tip and stress-induced growth of reoriented axon.** In our models, we consider two types of axon elongation, each of which arises from distinct mechanisms: tip growth, characterized by the exertion of forces by the growth cone at the axon's tip, which facilitates its forward movement. This type of growth can occur in the presence of sufficient biochemical signals and nutrients without external forces[56]. The other growth mechanism is stress-induced growth, which is interchangeably referred to as towed growth or stretch-dependent growth, wherein the shaft of the axon stretches due to deformation of the ECM or external forces. Experimental investigations[110–112] have demonstrated that axons subjected to external forces exhibit a chronic lengthening or shortening to maintain a desired level of axonal tension. Moreover, certain experimental studies have indicated that axons, when subjected to extremely low forces, exhibit the capability to grow at exceptionally high growth rates[113, 114]. Therefore, in our models, we assumed a fixed growth rate alongside a stress-induced fiber elongation rate. Therefore, we consider that, at each time increment, the overall axon growth in the oriented direction is dependent on not only the applied tension[35], but also the fixed axonal growth rate ($G^{axn}$). Accordingly, the overall growth rate of the axon can be defined by the stress-induced growth rate, as previously used by Garcia et al.[30] and Bayly et al.[23], along with the addition of a fixed axonal growth rate

$$\frac{\partial G}{\partial t} = a(\sigma - \sigma_0) + G^{axn} \tag{25}$$

Here, $\sigma$ represents the normal mechanical stress experienced by the axon in the direction of $\mathbf{n}^A$, $\sigma_0$ is the desired target stress, and $a > 0$ denotes the rate of stress-induced axon elongation rate. In the simulations conducted in this study, the target stress, which signifies the threshold triggering stress-induced growth, was established at zero. This choice aligns with experimental findings that indicate an exceedingly low threshold for tension-induced neurite elongation in embryonic forebrain neurons[113]. The growth rate is subsequently used to calculate the change in axon length $\Delta L$ during a discrete time increment $\Delta t$ through the Euler method

$$\Delta L = \frac{\partial G}{\partial t} \cdot \Delta t \tag{26}$$

The updated coordinates $(x_{k+1}, y_{k+1})$ of the axon bundle are determined based on the axon bundle's current location $(x_k, y_k)$ and the polar coordinates $\Delta L$ and $\theta$ (is the angle between $\mathbf{n}^A$ and horizontal axis) that represents the reoriented preferred direction

$$\begin{aligned} x_{k+1} &= x_k + \Delta L \cdot \cos(\theta) \\ y_{k+1} &= y_k + \Delta L \cdot \sin(\theta) \end{aligned} \tag{27}$$

This model provides a dynamic framework for the simulation of axon bundle growth, which captures the interplay between mechanical stress and directional preferences. The incorporation of the Euler method allows for the numerical exploration of axon bundle elongation over discrete time intervals, which offers insights into the spatiotemporal aspects of neural development.

## Finite element model

In our computational simulations as shown in Fig. 9c, we represent the developing brain as a simplified bilayer system, which comprises a tangentially growing cortex (the top layer) and a developing subcortical, or white matter, which includes growing ECM and axon bundles (the bottom layer). We incorporated the developed constitutive models in the previous subsection into the Abaqus/Explicit FE package using a user defined material subroutine (VUMAT). The cortex thickness of a matured brain typically ranges from 2 mm to 4 mm, depending on the specific region of interest[3,14,68]. Figure 9c represents the geometry and parameters of the developed FE model. A 2-D rectangle brain slice with the initial thicknesses of $t_c = 1.5$ mm and $h = 30$ mm was considered for the cortex and white matter, respectively. The model width was chosen to be $w = 60$ mm, which is sufficiently large relative to the wavelength of the folds[34]. This selection minimizes the influence of symmetric boundary effects on the folding patterns and the stress field induced by the folding process on regions far from the boundaries. The cortex and white matter (consisting only of ECM elements at the beginning of simulation) were meshed using the 4-node plane strain quadrilateral elements (CPE4R). The mesh size was chosen as 0.1 mm following an analysis of mesh convergence. To induce folding, cortical growth in tangential direction (along the $x$-axis in Fig. 9c) was specified to grow at a rate of $G^{ctx} = 0.02$ d$^{-1}$ [115] and the growth ratio between the cortex and white matter ECM was set as $G^{ctx}/G^{sub} = 6$. This equates to a sixfold increase in cortical surface area over a complete simulation period of 12 GWs, which aligns with studies showing an approximate 5- to 7- fold increase in cortical surface area during 20 to 37 gestational weeks[116–118]. Symmetric boundary conditions were applied along the "$x$" direction for both the left and right sides of the model, while the bottom side of the model was restrained along the "$y$" direction. The difference in growth ratios between the cortical layer and white matter serves as the driving force for the expansion and gyrification process[7,20]. The initiation and establishment of axon elements, with material properties distinct from the ECM, introduce imperfections into the system, facilitating the triggering of buckling. However, to ensure quasi-static equilibrium in the dynamic solver, the system's kinetic energy should remain below 5% of the internal energy[119].

In the FE models, every element of the bottom layer can be conceptualized as a part of an axon bundle with a distinct material property other than the ECM. To dynamically simulate axon bundle growth and real-time pathfinding, each bundle initiates at a random location, which spans 0 to 10 mm in the $y$ direction and 0 to 60 mm in the $x$ direction. Initial angles are selected randomly within the range of $\theta = \pi/2 \pm 10\%$ to direct axon bundles from the core of the model toward the cortex where the preferred direction is given by $\mathbf{n}^A = \cos(\theta)\mathbf{e}_x + \sin(\theta)\mathbf{e}_y$, where $\mathbf{e}_x$ and $\mathbf{e}_y$ are the unit vectors in the $x$- and $y$-axes, respectively. The preferred bundle direction, $\mathbf{n}^A$, is denoted as an internal variable and is locally stored at the integration point level. Following the adjustment of the preferred direction, $\mathbf{n}^A$, along the maximum tensile principal stress direction using Eq. (23) during the time increment $\Delta t$, the axon bundle initiates growth in the new direction using Eq. (25). The new tip point is generated at each iteration according to Eq. (27), and the ECM element containing this tip point is transformed into an axon element. This process is repeated until the final time step. Therefore, in the model and at any given time, each element can either be an axon element or an ECM element, but not both (Fig. 9d). Additionally, it should be noted that at each iteration,

the same isotropic growth is applied to both the ECM elements and the established axon elements. This isotropic growth helps to mitigate shear effects that might arise between the established, non-growing axon elements and the expanding ECM. The relaxation parameter $t^*$ was supposed to be 300 to obtain a gradual reorientation process without a broken angle in the path. This value was obtained based on trial and error[120]. We applied splines to create smooth visual representations of fiber paths, rather than showing them as connected chains of elements. The spline fitting was used primarily to enhance the visualization in the figures and did not affect the extraction of quantitative data. All values presented in the study were derived directly from the FE models without any alterations. The original and dynamic representations of the FE models are available in the supplementary movies.

We used the gyrification index (GI) as a metric to quantify the degree of cortical folding and assess the influence of model parameters on folding patterns. The GI is defined as the ratio of the cortical surface area (Pial) to the convex hull surface area. In our 2D models, this simplifies to the ratio of the total length of the pial line to the length of the convex hull line. The convex hull represents the smallest convex boundary that fully encloses the cortical boundary. For the models, the convex hull of the cortical boundary was calculated using MATLAB, which applies a standard algorithm to generate the minimal convex shape that encloses all points.

The model is limited to vertically growing axon bundles (radial fibers) to replicate the growth patterns of axon bundles, such as thalamocortical fiber tracts, that extend from the core of the brain toward the cortical plate. However, the organization of white matter in the brain is far more complex, encompassing not only radial fibers but also tangential fibers (short- and long-association fibers). In this study since the primary objective is to replicate the growth of thalamocortical fiber tracts, which are predominantly radial in nature and navigate from the thalamus toward the cortical plate, we did not consider tangential fibers in the models to focus on the fibers that navigate toward the cortical plate and gradually experience the induced stress field as the result of cortical folding.

Table 1 presents the values and ranges of previously reported stretch- or force-dependent elongation rates, along with the calculated stress-induced fiber elongation rate, denoted as $a$. The axonal growth rate ($G^{axn}$) varies widely, ranging from 0.2 to 1.8 mm d$^{-1}$ [121,122], and the stress-induced elongation rate spans 0.0004 to 0.31 mm Pa$^{-1}$ d$^{-1}$. Here, we assume fixed axonal growth rates of $G^{axn} = 0.6, 0.8, 1.0$, and 1.2 mm d$^{-1}$ and stress-induced elongation rate of $a = 0.015$ mm Pa$^{-1}$ d$^{-1}$. To provide further detail, an axonal bundle initiates growth at a consistent rate of $G^{axn}$ mm d$^{-1}$ in regions far from the cortex, which is characterized by extremely low or non-existent stress. The resulting trajectory of the bundle exhibits a more chaotic pattern, which aligns with observations from prior studies[48,56]. Conversely, in regions that experience stress as the result of folding, the preferred direction of the bundle undergoes a rotation to align with the maximum tensile principal stress direction, which leads to accelerated growth at a rate of $(G^{axn} + 0.015\sigma)$ mm d$^{-1}$, where $\sigma$ represents the stress. Given that cortical folding completes over an extended time scale, a nonlinear hyperelastic material model has been employed as a first-order approximation to examine the behavior of axonal fibers. This approach aligns with other micromechanical modeling studies that involve axonal fibers[93,96,123]. In the models, axonal fibers can be stretched and elongated or compressed under the deformation of the brain tissue[15].

The shear modulus ratio of the cortex to the white matter ECM was set as 2[65]. However, this parameter is subject to variation. The ratio of shear moduli between axons and ECM has been documented in a broad range from 1 to 13[93,96,123–125]. However, these values pertain to axons in the mature brain with a high degree of myelination. For the premature brain, where myelination is minimal, it is anticipated that

**Table 1 | Parameter range for neurite elongation based on available literature**

| Cell type | Reported axon caliber $D$ ($\mu$m) | Stretch-dependent elongation rate (mm d$^{-1}$) | Force-dependent elongation rate b ($\mu$m pN$^{-1}$ h$^{-1}$) | Calculated stress-dependent elongation rate a (mm Pa$^{-1}$ d$^{-1}$) |
|---|---|---|---|---|
| Embryonic rat dorsal root ganglia[133] | 0.9 | 2.0 | – | 0.0004 – 0.02* |
| P0-P1 murine hippocampal neurons[113] | 5.0 | – | 0.66 | 0.31** |
| Embryonic chick forebrain neurons[114] | 1.0 | – | 0.05 – 0.5 | 0.001 – 0.01** |
| Embryonic chick dorsal root ganglia[134] | 2.0 | – | 0.02 – 0.55 | 0.001 – 0.04** |

* In studies that measured stretch-dependent elongation rates, the stress-dependent elongation rate, $a$, was estimated using the relation $a = G^{axn}/E$, following the approach of Garcia et al.[30]. The Young's modulus of individual neurites, which varies between 100 and 4600 Pa[135,136], provides a corresponding range for the stress-dependent elongation rate.

** In studies that reported force-dependent elongation rate ($b$), the parameter $a$ was determined using the formula $a = bA$, where $A$ is the cross-sectional area of the neurite.

the shear modulus ratio between axons and ECM is lower compared to the mature brain[126]. The stiffness of white matter escalates with the increase in myelin content[99]. Here, we opted for a baseline ratio of 2 and will explore the impact of various ratios on axonal growth and cortical folding later. The parameters and their values used in the simulations are listed in Table S2.

### Image processing and defining folding morphologies and fiber density

To evaluate our models, we quantified the density of fiber tracts in gyri and sulci for 10 human brains using available imaging data. Our specific focus was on thalamocortical fiber tracts, which serve as representatives of ascending fiber tracts. These fiber tracts originate from the core of the brain and extend toward the cortex, similar to the FE models. According to Table S1, thalamocortical functional connectivity strength experiences a notable peak increase between the 29th and 31st GW, which occurs just prior to the establishment of axonal synapses in the cortex[61].

Structural MRI (sMRI) and diffusion MRI (dMRI) scans were obtained from 10 healthy adult brains aged 22–35 years, sourced from the Q1 release of the WU-Minn Human Connectome Project (HCP) (https://humanconnectome.org/). Each subject underwent acquisition of a pair of T1-weighted (T1w) scans and a pair of T2-weighted (T2w) scans, all at a spatial resolution of 0.7 mm isotropic voxels. The sMRI parameters were as follows: TR = 2400 ms, TE = 2.14 ms, flip angle = 8 degrees, image matrix = 260 × 311 × 260, with a resolution of 0.7 × 0.7 × 0.7 mm³. Cortical surfaces of both white matter and gray matter have been reconstructed from T2-weighted MRI data and provided in the HCP dataset, which follow the steps of skull removal, tissue segmentation, and surface reconstruction. More details for the above-mentioned steps have been discussed in Makropoulos et al.[127].

On dMRI data, skull-strip and eddy current correction were applied using FSL[128]. Then, in DSI Studio, a model-free generalized $Q$-sampling imaging method was adopted to estimate the density of diffusing water at different orientations. A multi-shell diffusion scheme was employed with b-values of 1000, 2000, and 3000 s mm$^{-2}$, and 90 diffusion sampling directions for each. The in-plane resolution and slice thickness were both set to 1.25 mm. Restricted diffusion was quantified using restricted diffusion imaging[129]. Data reconstruction utilized generalized q-sampling imaging with a diffusion sampling length ratio of 1.25. Fiber tracking employed a deterministic streamline tracking algorithm[130] with augmented strategies for improved reproducibility. Anisotropy thresholding ranged randomly between 0.5 and 0.7 Otsu threshold. Tracks outside the length range of 30.0 to 200.0 mm were discarded, in which with a total of 1,000,000 seeds were placed. After generating the fibers for the whole brain, we extracted the thalamocortical fiber tracts for all 10 brains. Figure S6a represents the thalamocortical fiber tracts of a brain sample.

On triangular surfaces of gray matter and white matter, sulcal depths were computed by using Freesurfer, examples of which can be seen on an adult brain in Fig. S6b. Sulcal depth, $h_{sulc}$, is defined as the distance between points on the cortical surface along the sulcus and

the corresponding points on a reference surface. Gyral regions were defined as those with $h_{sulc} \geq 0.0$ mm and sulcal regions with $h_{sulc} < 0.0$ mm. In this study, we divided the surface morphology into gyri and sulci, which eliminates the walls to ensure consistency with our models and enable a direct and unbiased comparison. To calculate the fiber density in gyri and sulci, we divided the number of fibers within each region by their respective areas. Additionally, we normalized the fiber density by dividing it by the total fiber density, defined as the sum of the fiber densities in gyri and sulci, and then multiplied it by 100. This percentage roughly represents the proportion of fibers present in gyri or sulci relative to the total number of fibers, based on the comparable surface areas of gyri and sulci on the white matter's surface. Figure S6c illustrates an example of the mapping of thalamocortical fiber tracts within the gyri and sulci of the white matter surface.

For the FE models, we generated a mid-surface, which represents an intermediate surface located between the outer (pial) surface and the inner (white matter) surface of the cortex. This mid-surface is essentially a geometric approximation of the average position of the cortical surface. Once the mid-surface is generated, regions above the mid-surface were classified as gyri, while regions below the mid-surface were classified as sulci.

### Statistical analysis

A comprehensive statistical analysis of the study findings was conducted using SPSS to examine the influence of the biophysical properties of axons, cortex, and white matter ECM on cortical folding and fiber organization. We employed Tukey's Honestly Significant Difference (HSD) test[131] with a significance level of 0.05 and 95% confidence intervals. This method was used to compare means across multiple groups and is particularly valuable for identification of statistically significant differences between pairs of groups.

### Reporting summary

Further information on research design is available in the Nature Portfolio Reporting Summary linked to this article.

## Data availability

The data supporting these findings, including model files and processed MRI data from this study, have been deposited in Zenodo[132] (https://doi.org/10.5281/zenodo.14553393). Source data are provided with this paper.

## Code availability

The codes and data supporting the findings of this study are available as part of the Zenodo dataset[132] (https://doi.org/10.5281/zenodo.14553393).

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

## Acknowledgements

This work was supported by the National Science Foundation [CMMI: 2123061] and partially by the SUNY Research Seed Grant [#231111] awarded to M. J. Razavi. The authors would like to thank Ali Gholipour from Boston Children's Hospital for sharing the longitudinal fetal brain scans. Data were provided in part by the Human Connectome Project, WU-Minn Consortium (Principal Investigators: David Van Essen and Kamil Ugurbil; 1U54MH091657) funded by the 16 NIH Institutes and Centers that support the NIH Blueprint for Neuroscience Research; and by the McDonnell Center for Systems Neuroscience at Washington University.

## Author contributions

A. S.: Methodology, Formal analysis, Software, Writing—original draft. A. H. F.: Methodology, Formal analysis, Software, Writing—original draft. L. P.: Investigation, Writing—review and editing, M. J. R: Conceptualization, Investigation, Supervision, Writing—review and editing, Funding acquisition. All authors reviewed the manuscript.

## Competing interests

The authors declare no competing interests.
