## [Transparent Peer Review file · Nature Communications]

Stress Landscape of Folding Brain Serves as a Map for Axonal Pathfinding

Corresponding Author: Dr Mir Jalil Razavi

Version 1:

Reviewer comments:

Reviewer #1

(Remarks to the Author)

The manuscript describes the development and application of a mathematical model of axon growth and alignment in the developing brain during the period of cortical folding. In this model axons are assumed to elongate at a rate that depends on stress, and align with the direction of maximum tensile principal stress (MTPS). The results of the model are presented in terms of axon fiber density below gyral peaks and sulcal fundi.

Strengths: The manuscript has numerous strengths. The authors propose plausible mechanisms for axon growth and reorientation. Their mathematical model is rigorous and clear, consistent with both physical laws and reasonable constitutive models. The authors systematically explore the effects of the assumed mechanisms and their governing parameters on local axon density. Notably, they compare the model predictions to experimental measurements of axon density from imaging data.

The model exhibits a nice balance between simplicity and complexity. The spatial domain is only 2D, and the properties are consistent throughout cortical and subcortical regions. This simplicity clarifies the roles of the factors that are included: stress-dependent growth, axon re-orientation, and the separation of subplate into axons and “extracellular matrix” (ECM).

Weaknesses: There are a few weaknesses that could be addressed in a revision.

1. The manuscript in its current form does not include any figures showing the stress state in the tissue. The maps of axon distribution would be complemented by maps of the stress field.
2. The folding patterns are generally similar. It would be interesting to explore more clearly different folding/instability patterns and see the resulting stress and fiber fields.
3. The evidence in the paper is fundamentally correlative. The study does not include a specific experimental perturbation of stress field or folding patterns, and accompanying effect on fiber patterns. So conclusions should be tempered.

Other comments

1. The term “validate” (p.15, p.30) is not really accurate for a simplified model such as this one (or really most models), due to inevitable simplifying assumptions and approximations. (“All models are wrong; some are useful”). This is not saying that a model is bad, just a guide to how to interpret it. The terms “evaluate” or “assess” are better, and the authors already use those on occasion.

2. The authors cite reference [42] to claim that outer cortical layer is stiffer than subplate, and explore factors up to 4. I think that many other studies have observed smaller differences in stiffness. I suggest that the authors acknowledge that the preponderance of evidence supports a factor of 1-2 between cortical and subcortical stiffness.

3. A recent paper describes the temporal evolution of tissue properties during the period of folding. Walter C, Balouchzadeh R, Garcia KE, Kroenke CD, Pathak A, Bayly PV. Multi-scale measurement of stiffness in the developing ferret brain. Sci Rep. 2023 Nov 23;13(1):20583. doi: 10.1038/s41598-023-47900-4. PMID: 37996465; PMCID: PMC10667369.

(Remarks on code availability)

Reviewer #2

(Remarks to the Author)

This computational study improves upon previous models of cortical folding and subcortical axon remodeling to illustrate how mechanical stresses resulting from cortical folding may serve a role in “axonal pathfinding”, defined via a combination of towed growth and axonal tip reorientation/migration. The latter is accomplished by implementation of novel modeling methods, which also provide compelling, intuitive illustrations of the process. However, authors fail to adequately cite past modeling work in places, as well as contextualize what model predictions are novel to the current study versus redundant with previous mechanics-based studies using alternative implementations of stress-dependent reorganization. For example, in the abstract authors state, “The results of the study, for the first time, demonstrate the potential mechanical mechanisms by which the density of axon bundles in gyri (ridges) is higher than in sulci (valleys), which is an observation derived from experiments and imaging data that has not been fundamentally explained before.” However, previous models implementing towed growth or passive axon have already demonstrated how either of these mechanisms can result in higher density of axon bundles in gyri than sulci.

In light of past work, the impact of the current work is less than currently stated by authors. However, the current model does possess unique elements that are truly novel and interesting. Thus, the work is not suitable for publication in its current form, but I encourage authors to revise and resubmit their work after addressing the points below.

Major concerns

1. Authors should clarify what aspects of their model and analysis are novel, as opposed to extension or replication of previous work. Much of the method related to towed growth appear similar to Garcia et al. 2021, but are not cited as such. Use of previously reported theoretical frameworks, equations, parameter choices, as well as comparisons to human / animal models, should be clearly contextualized as such.
2. An novel aspect of the proposed is implementation of axon rotation to the primary stress direction, which results in interesting and compelling behavior. Authors claim that this addition accelerated/accentuated the accumulation of radial fibers in gyri. However, the accumulation of radial fibers within gyri has also been demonstrated in models without this feature. This includes (1) models of simple folding deformations in a substrate containing discrete axons (e.g., Chavodhnejad et al. 2021) as well as (2) models of towed growth that considered radial fiber density as part of a composite tissue definition (Garcia et al 2021). In the Introduction and Discussion, it would be useful and important to compare and contrast the current model, in light of these previous studies.
3. An important limitation of the current model is the failure to implement or consider tangential fibers. Reasoning for omitting these fibers should be provided, and omission should be described in the discussion/limitations section.
4. Similarly, the current study relies on tractography-based comparisons to human data, which are notoriously biased to capture radial fibers in gyri. Authors should describe limitations related to tractography in the Discussion section.
5. All models of the current manuscript are limited to a 2D rectangular space that, while commonly applied in simple studies of brain folding, has important limitations. Most critically, fixed or symmetry constraints along the bottom edge and sides will lead to stresses that may not reflect – or will less realistically reflect – stresses in the developing brain. Authors are encouraged to explore the stress state in more realistic, even if still 2D, representations of the brain, such as circles/ellipses or even ellipsoids via axisymmetry. In either case, limitations of this framework should be discussed.
6. The current description of the white matter constitutive model is confusing. First, authors describe an inner subcortical core “primarily composed of neuronal axons,” then as a combination of axons and ECM. The terminology in Methods could be clarified. Notably, the developing subcortical tissue is not primarily composed of neuronal axons – rather, at early stages, the subcortical tissue is dominated by radial glial cells/scaffolds, extracellular matrix, and other cells. If the “white matter” tissue was indeed mostly axons, justify this choice and describe in Limitations.
7. Perhaps related to the above point, authors describe growth of the white matter as “elastic” and isotropic. The description of growth as “elastic” is confusing to this reviewer – perhaps they meant inelastic? This is an important distinction. Also, is the “isotropic” growth separate from the axon elongation described elsewhere? How do these fit together?
8. Table 2 has some important inaccuracies. I encourage authors to double check these fields.

Minor concerns:

1. In introduction, authors state, “This hypothesis has been supported by recent experimental and computational studies,” and do cite several more recent studies (11,15–21). However, this hypothesis has a much longer history, and is supported by studies dating back at least 50 years. It would be good to cite these original papers. A review of these can be found in the current papers referenced.
2. On page 2, a reference is needed for the statement, “This aligns with the observation that gyri consist of a greater number of neurons compared to sulci.”
3. On page 3, the statement, “This hypothesis has been recently updated 10” is unclear, as it implies that the hypothesis of axon elongation in response to folding has been updated.
4. On page 4, the statement, “The intricate interplay between growing axonal fiber bundles and stress-strain fields and the mechanisms that underlie the observed greater fiber density in gyri compared to sulci lacks clarity,” is redundant and perhaps overstated. I would recommend omitting this sentence.
5. On page 5, the statement, “Table S1 indicates that the major part of cortical folding and connectivity development of the thalamocortical fiber tracts occur concurrently,” could be omitted and instead Table S1 referenced in parentheses at the end of the previous sentence.

6. On page 6, the statement, "The imaging data indicate that cortical thickness undergoes changes during the early developmental stage," should include a reference.
7. Fig. 2 panel a seems to suggest that direction changes will happen at the neuronal cell body. However, biologically I would imagine the cell body as the anchor point, and the other end as the axon tip as the point being guided by stress. Perhaps authors can clarify their intention with this figure and consider an alternative illustration.
8. On page 13, isotopically should be isotropically
9. The statement, "The effect of boundary conditions was eliminated by increasing the width of the model to $w = 60 \text{ mm}$ " should be adjusted, as increasing the width of the model does not eliminate the effect of boundary conditions completely. I suspect authors mean that this would eliminate one or more specific effects related to model width when the domain is smaller.
10. On page 18, "This observation aligns with previous studies on stresses within the developing brain" appears to have the wrong reference. Please check all in text citations throughout the manuscript.
11. The statement, "This equates to a sixfold increase in cortical surface area over a complete simulation period of 50 gestational days 37." appears to have the wrong reference. Please check all in text citations throughout the manuscript.

(Remarks on code availability)

Reviewer #3

(Remarks to the Author)

This work introduces a multi-scale model for axonal growth that investigates its link to cortical folding. In a bilayer model with growing cortex and sub-cortex, the growing axons are hypothesized to reorient themselves along the direction of the maximum tensile principle stress in the white matter, which can be different from the initial direction of growth. Various parametric studies reproduce experimental observations (higher density of axonal fibers under gyri than sulci) and indicate potential inverse correlations between gyrification index and cortex:substrate stiffness ratio. Studies also indicate an inverse relationship between growth rate of axons to the gyrification index, further indicating that if the equivalent stiffness of the white matter (tissue + axon) increases, gyrification decreases. The paper makes important observations, but needs further improvement in the manuscript before publication.

Major concerns:

1. There is confusion on how the axon is modeled in the FE simulations. Can each element in the subcortex be both an axon and white matter? Fig 2 and Fig 7c suggest that an element can either be an axon OR white matter, not both. Later on, simulations account for equivalent stiffness. Further, the visualization of how an axon grows seems to indicate that axon is smoothly reorienting itself, which suggests that an element can be both. Can the authors clarify this?
2. The equations for reorientation (20-23) as well as the algorithm provided suggest that the process is deterministic. Yet, later on in the text, there is mention of gaussian noise (with zero mean) added. This should be clarified in both the text and the equations.
3. Related, in Fig. 11 and in the relevant section, 'Without Reorientation' really means that the reorientation is not stress-driven (rather due to the small gaussian noise at each step). If this is the case, the terminology should be clarified.
4. The method/equation for calculating gyrification index is missing; it is also important to carefully define how the convex hull is found.
5. The paper would be much clearer with a table of the values/ranges of all the parameters used in the simulations.
6. Code and data should be shared via Github (only data is mentioned, and has not yet been shared).

Minor concerns:

1. P22: " In Fig. 6(a), a stiffness ratio of 1, which creates an isotropic substrate for folding of the cortex, has no effect on the morphology of cortical folds." This should be reworded; the stiffness ratio does affect the morphology, but the axons don't.
2. There are several notation issues that the authors need to pay attention to:
 - 2.1 The notation for derivative w.r.t time (D_t vs $\dot{\theta}$ vs $\partial D / \partial t$)
 - 2.2 The normals defined in the algorithm and in equations 15-19 are going to be updated at each time step 'k', hence each of them require a subscript 'k'.
 - 2.3 Likewise with the variables ω and R (eq. 20).
 - 2.4 The basis vectors e_x and e_y on pg 13 (top) should be defined.
 - 2.5 Units should be consistently used, e.g. day or d but not both
 - 2.6 The caption of Fig 7 uses the 'x' symbol which in the text implies the cross product.
 - 2.7 If the authors mean that higher order tensors are in bold and scalars are in unbold, then the variables ϵ and ω

in eq 20 should be in bold.

3. There are several typos as well that need to be corrected:

3.1 Section number 2.1 is used twice. The proceeding sections should also be renumbered.

3.2 On pg 26, there is a line break that divides a sentence into two paragraphs.

3.3 The thickness of the cortex t_c is shown as 1.5 m, which should likely be millimeters

3.4 reference #9 should be Van Essen, not Essen

3.5 In the first sentence of the introduction, I believe you are referring to neural tubes, not neuronal tubes

4. Red-green color scales should be avoided due to issues for colorblind individuals

(Remarks on code availability)

No code was included. The manuscript mentions that the data will be available from their github page, but there's actually no mention of sharing the code even after publication.

Reviewer #4

(Remarks to the Author)

This manuscript investigated the interplay between cortical folding and brain connectivity development through a proposed mechanical model. The authors explored how growing axon bundles navigate within a dynamic stress field influenced by the folding cortex. There are several key findings, including the mechanical stress influence on axonal growth, the bidirectional relationship between the biophysical properties of axons and cortex patterns, and how axonal growth alters the material properties of white matter. The manuscript showed the critical role of mechanics in early brain development and suggests that understanding these interactions could provide insights into brain disorders characterized by abnormal folding and connectivity. However, the proposed model seems to be rough and confusing.

1. Although the introduction included an extensive literature review, it is still difficult to identify the novelty of this work.

2. In Eq. (24), the authors provided a growth rate of the axon which add a fixed growth rate G^{axn} to the traditional stress-dependent growth rate. However, this makes no sense. In fact, after adding the growth rate G^{axn} , the desired target (homeostatic) stress σ_0 becomes $\sigma_0 - G^{axn} / a$. The authors set $\sigma_0 = 0$ and $G^{axn} > 0$, which corresponds to a negative homeostatic stress. Therefore, the growth rate Eq. (24) is inconsistent with the experimental investigations [Refs. 66-68].

3. Many of the parameters used in the simulations in the study were not real data from the brains, but used some experimental data of other tissues. It is impossible to estimate whether these influenced the results.

4. The study simulated the interaction between cortical folding and axonal development in the developing brain, which displays clear 3D features. However, the proposed model was simplified to a 2D geometry.

5. Below Eq. (1), the authors stated that the gradient operator is applied with respect to the material point X in the reference configuration. In Eq. (1), however, the gradient operator is applied with respect to point x in the current configuration.

6. Eqs. (3)-(4) and Eqs. (9)-(10) both refer to F^g , but they represent different quantities. To avoid misunderstanding, different notations should be used.

7. There are some inconsistencies between Fig. 5 and its citations on Pages 20 and 21.

(Remarks on code availability)

Version 2:

Reviewer comments:

Reviewer #1

(Remarks to the Author)

The authors have addressed the concerns raised in my previous review.

(Remarks on code availability)

Reviewer #2

(Remarks to the Author)

Thank you for these clarifications to the manuscript. The new version is much improved!

(Remarks on code availability)

Reviewer #3

(Remarks to the Author)

The responses from the authors and the changes that they have made in the manuscript have addressed many of the issues raised by the reviewers, but I still have significant concerns about missing or unclear methods and overstated claims.

Major issues:

1. The abstract says "In particular, the degree of gyrification is shown to correlate strongly with the density, growth rate, and mechanical properties of underlying axon bundles." However, there is very little evidence - certainly not to support a claim of "strong correlation". For example, Fig. 5 looks at the effect of growth rate but changes in morphology are not quantified; Figs. 6 and 7 quantify the effect of axonal stiffness and number of fibers on GI and find statistically significant results (but realistically, likely a very small effect size). What seems to be somewhat more supported by the data is that the cortical folding morphology affects the growth of axons (in their simulations; not necessarily in vivo).

2. Relatedly, at the bottom of page 28 the section is titled "Effect of stochastic nature of axon growth on cortical folding and fiber organization" but the related figure, Fig. 9, does not quantify the changes in cortical folding (e.g. gyrification index, wavelength, or sulcal depth).

3. Several methodological details are still missing.

a. While the response to reviewers explained the use of splines to visualize axonal directions, the only mention of splines in the manuscript is to point out where they aren't used (p22).

b. The calculation of GI is said to require fitting of a "smoothed, curved line that minimally encloses the cortical surface". More details would be helpful. I also assume the dotted line in Fig. 6c is a hand drawing and not the actual convex hull, because it shows some small areas of concavity.

c. Many similar simulations include some kind of perturbation or imperfection to trigger buckling. Was one used here?

4. I believe Table 1 has a mistake; if the "Else if" statement in step 2 is not true, then the quantity n^T_{max} would be undefined.

5. In Fig. S1, why does the number of simulations affect the results? I expected to see some narrowing of a range of uncertainty, but in fact uncertainty/standard deviation is not even indicated on the plot.

6. The inclusion of the timeline in Table S1 is helpful. However, please indicate the approximate weeks that correspond to your simulations.

Minor issues:

7. The scales of the y axes in Fig. 7e and 6c are extremely misleading and should be revised to include 1.

8. In Fig. S3, there are interesting green shapes in the middle of the mostly-blue substrate. Can the authors explain these?

9. Fig. S4 still has the titles "with/no reorientation" that was fixed in the last revision.

10. On p29, specify what quantity you are discussing the standard deviation of.

11. Some nomenclature is inconsistent - both preferred and present direction are used on p14, and MPTS is used instead of MTPS on p22.

(Remarks on code availability)

I briefly looked through the code and did not see any significant reasons for concern.

Reviewer #4

(Remarks to the Author)

I have no further comments.

(Remarks on code availability)

Version 3:

Reviewer comments:

Reviewer #3

(Remarks to the Author)

Thank you to the authors for their revisions and response.

I continue to have concerns about the support of the authors' claims.

1. The revised abstract states that "Importantly, our results demonstrate that the density of fibers reaching the cortical plate prior to folding plays a significant role in determining the specific locations where gyri and sulci form." Where is this information shown? Figs. 5 and 8 show simulations where fibers reach the cortical plate before and after folding begins, but don't compare resulting morphologies or fold placement, only fiber density in gyri and sulci.

2. The revised abstract also mentions "the connectivity patterning in the folding-induced stress field". What does this mean? What stress field?

3. The mention of cortical folding morphology depending on fiber organization was removed from the section heading, but not the caption of Fig. 9.

(Remarks on code availability)

Review Reply Letter

We greatly appreciate the reviewers for their thorough review, supportive comments, and constructive feedback on the manuscript. We are pleased that the reviewers have acknowledged the proposed model, its novelty, and its findings. In response to the reviewers' insightful comments and suggestions, we have diligently revised our manuscript, making substantial changes to address all concerns and enhance the clarity of the proposed model and framework, differentiating them from previous models and studies. We have also added new figures that further strengthen the study and support its novel findings. In this letter, we have provided detailed responses to each specific question raised, highlighted in blue. Additionally, all modifications have been incorporated and are marked in green in the review reply letter and in blue in the revised manuscript. In the revised manuscript, we have avoided overusing terms like “novel”, “new”, and “first” throughout the text, in line with the guidelines for Nature formats. Instead, we have subtly emphasized the originality of our work and the significance of our findings. We hope the reviewers find that the changes we have made in response to their feedback have significantly improved the manuscript.

Reviewer #1 (Remarks to the Author):

The manuscript describes the development and application of a mathematical model of axon growth and alignment in the developing brain during the period of cortical folding. In this model axons are assumed to elongate at a rate that depends on stress, and align with the direction of maximum tensile principal stress (MTPS). The results of the model are presented in terms of axon fiber density below gyral peaks and sulcal fundi.

Strengths: The manuscript has numerous strengths. The authors propose plausible mechanisms for axon growth and reorientation. Their mathematical model is rigorous and clear, consistent with both physical laws and reasonable constitutive models. The authors systematically explore the effects of the assumed mechanisms and their governing parameters on local axon density. Notably, they compare the model predictions to experimental measurements of axon density from imaging data.

The model exhibits a nice balance between simplicity and complexity. The spatial domain is only 2D, and the properties are consistent throughout cortical and subcortical regions. This simplicity clarifies the roles of the factors that are included: stress-dependent growth, axon re-orientation, and the separation of subplate into axons and “extracellular matrix” (ECM).

We sincerely thank the reviewer for their positive comments and for providing a concise and thoughtful summary of our work.

Weaknesses: There are a few weaknesses that could be addressed in a revision.

We sincerely thank the reviewer for their constructive comments, all of which we have carefully addressed in the revised manuscript. Below, we provide detailed responses and the actions taken in response to each comment.

1. The manuscript in its current form does not include any figures showing the stress state in the tissue. The maps of axon distribution would be complemented by maps of the stress field.

Thank you to the reviewer for this constructive comment. In the revised manuscript, we have included an image that illustrates how the distribution map of axon bundles aligns with the stress state of the tissue. This new image has been added to Figure 1 of the manuscript, and is also shown here. As depicted, the growing axon bundles closely follow the tissue's stress map, aligning with areas of tensile and compressive stress. We have made minor revisions to the manuscript to highlight the inclusion of stress contours. Additionally, we added a new figure in the Supplementary Information that displays the stress field across the entire model (Fig. S3). For further details, please refer to the next comment (comment #2).

Fig. 4 Dynamic growth and pathfinding of fibers within a growing and folding bilayer system. In this specific model, only 20 fibers have been initiated and grown for a better representation of the results. Pink arrows represent the maximum tensile principal stress (MTPS) directions and green arrows show the preferred directions. $T = G^{ctx}t$ is dimensionless simulation time that ranges from 0 to 1, where G^{ctx} and t are cortex growth rate and time, respectively. Normalized Maximum Principal Stress (defined as the ratio of the maximum principal stress to the shear modulus of the ECM (μ_s)) is shown for each step. For better clarity, readers are referred to the color version of the figure in the online version of this article.

2. The folding patterns are generally similar. It would be interesting to explore more clearly different folding/instability patterns and see the resulting stress and fiber fields.

Thank you to the reviewer for this insightful comment. The observed similarity in folding patterns is attributed to several factors:

1. **Constant initial thickness:** The initial thickness of the cortex layer is held constant in our model. Variations in this parameter could affect fold size; increasing or decreasing the initial thickness of the cortex would correspondingly increase or decrease the size of the folds.
2. **Uniform cortex layer:** Our model assumes a uniform cortex layer thickness. In reality, the cortex has varying thickness across anatomical sites, which influences the morphology of brain folds.
3. **Homogeneous growth:** To simplify the model, we did not include heterogeneous growth within the cortex layer. A more complex growth profile could alter fold morphology.

These simplifications were made to focus exclusively on the growth and pathfinding of axon bundles within the folding stress field. To address this, we have added a supplementary figure (Figure S3 in the Supplementary Information) demonstrating how variations in the initial cortex thickness impact fold size and stress distribution during and after folding process. Increased cortex thickness results in larger folds and greater stress penetration into the bottom layer, which in turn affects the reorientation of axon bundles. Additionally, we have included a statement in the revised manuscript to highlight the variation in cortical folds in the human brain and address the reviewer's comment. This will help readers understand the implications of cortical thickness on fold morphology and the observed growth patterns of axon bundles.

“It is important to note that in our models, the initial thickness of the cortex is kept constant, which results in similar folding patterns across all models. However, varying the initial thickness can significantly alter the resulting fold shapes. Figure S3 illustrates that increasing the initial thickness of the cortex layer leads to larger folds and greater penetration of the stress field into the white matter layer. Despite these variations, the observations for the growth and reorientation of fibers remain consistent with the base models. Additionally, variations in cortex thickness or a heterogeneous growth profile within the cortex could produce different folding morphologies. This highlights that cortical folds in the human brain exhibit more complex variations.”

Fig S3. Cortical folding and normalized maximum principal stress map for an initial cortex layer thickness of (a) 1.5 mm, and (b) 3 mm. In both scenarios, all other parameters remained constant ($G^{axn} = 0.8 \text{ mm/d}$, $\mu_c/\mu_s = 2$, $a = 0.015 \text{ mm Pa}^{-1} \text{ d}^{-1}$). As shown, increasing the cortex layer thickness leads to a longer folding wavelength and the emergence of larger folds. Additionally, the maximum principal stress increases slightly with the thicker cortex layer.

3. The evidence in the paper is fundamentally correlative. The study does not include a specific experimental perturbation of stress field or folding patterns, and accompanying effect on fiber patterns. So conclusions should be tempered.

Thank you to the reviewer for this insightful comment. We fully agree that we should be cautious in making strong or definitive claims based on our findings, as the evidence is primarily correlational and lacks experimental validation. We have partially addressed this in the limitations section of our study; however, it is essential to ensure that this caution is reflected throughout the manuscript. Therefore, we revised manuscript to be more tentative rather than definitive, providing appropriate caution and clearly acknowledging the limitations of the evidence. To hint the readers, we add the following statement at the beginning of the Results section:

“It is important to note that all results and findings are derived from predictive models. These models suggest potential mechanisms that may apply to the developing brain, rather than describing the exact process in the human brain. The terminology used for model constituents and parameters is aligned with that of the human brain solely to establish an analogy.”

Other comments

1. The term “validate” (p.15, p.30) is not really accurate for a simplified model such as this one (or really most models), due to inevitable simplifying assumptions and approximations. (“All models are wrong; some are useful”). This is not saying that a model is bad, just a guide to how to interpret it. The terms “evaluate” or “assess” are better, and the authors already use those on occasion.

Thank you to the reviewer for this critical comment. We fully agree that the term “validate” is not accurate in the absence of experimental validation. We apologize for the oversight and have corrected all instances of “validate” to “evaluate” in the revised manuscript.

2. The authors cite reference [42] to claim that outer cortical layer is stiffer than subplate, and explore factors up to 4. I think that many other studies have observed smaller differences in stiffness. I suggest that the authors acknowledge that the preponderance of evidence supports a factor of 1-2 between cortical and subcortical stiffness.

Thank you to the reviewer for this constructive comment. The stiffness ratio of the cortex to white matter exhibits considerable variability depending on the experimental setup, loading conditions, and other factors. We agree that most evidence supports a stiffness ratio in the range of 1-2. Budday et al¹. characterized the material properties of the cortex and white matter through simultaneous fitting into multiple loading cases, finding that the stiffness ratio of the cortex to the Corona Radiata and corpus callosum is approximately 1.8 and 3.18, respectively, assuming a neo-Hookean material behavior. Therefore, in our parametric study, we considered a stiffness ratio range from 1 to 4. In addition, we speculate that the stiffness ratio between the cortex and subplate/white matter in the fetal brain may be higher than in the mature brain, as the absence of myelination in the fetal brain could lead to an even softer subplate². Larger stiffness ratios, such as 2 to 12, have also been used in parametric studies by Budday et al. ³, and even larger ratios

by Budday et al⁴. However, we acknowledge that most experimental studies suggest a stiffness ratio between 1 and 2^{5,6}. To address this comment, we have revised the manuscript to emphasize that the preponderance of evidence supports a factor of 1-2 between cortical and subcortical stiffness.

“We selected a stiffness ratio range of one to four for our parametric study, given that the stiffness ratio between the cortex and subplate (later white matter) in a folding brain may vary widely due to the changing material properties of brain tissue during folding⁷. However, the majority of experimental evidence supports a ratio of one to two between cortical and white matter stiffness in the mature brain^{1,5,6}.”

3. A recent paper describes the temporal evolution of tissue properties during the period of folding. Walter C, Balouchzadeh R, Garcia KE, Kroenke CD, Pathak A, Bayly PV. Multi-scale measurement of stiffness in the developing ferret brain. Sci Rep. 2023 Nov 23;13(1):20583. doi: 10.1038/s41598-023-47900-4. PMID: 37996465; PMCID: PMC10667369.

Thank you to the reviewer for recommending the relevant paper. It provided valuable insights that helped us justify our choice of a stiffness ratio range of 1 to 4 (Reference number 7).

“We selected a stiffness ratio range of one to four for our parametric study, given that the stiffness ratio between the cortex and subplate in a folding brain may vary widely due to the changing material properties of brain tissue during folding⁷. However, the majority of experimental evidence supports a ratio of one to two between cortical and white matter stiffness in the mature brain^{1,5,6}.”

Reviewer #2 (Remarks to the Author):

This computational study improves upon previous models of cortical folding and subcortical axon remodeling to illustrate how mechanical stresses resulting from cortical folding may serve a role in “axonal pathfinding”, defined via a combination of towed growth and axonal tip reorientation/migration. The latter is accomplished by implementation of novel modeling methods, which also provide compelling, intuitive illustrations of the process. However, authors fail to adequately cite past modeling work in places, as well as contextualize what model predictions are novel to the current study versus redundant with previous mechanics-based studies using alternative implementations of stress-dependent reorganization. For example, in the abstract authors state, “The results of the study, for the first time, demonstrate the potential mechanical mechanisms by which the density of axon bundles in gyri (ridges) is higher than in sulci (valleys), which is an observation derived from experiments and imaging data that has not been fundamentally explained before.” However, previous models implementing towed growth or passive axon have already demonstrated how either of these mechanisms can result in higher density of axon bundles in gyri than sulci.

In light of past work, the impact of the current work is less than currently stated by authors. However, the current model does possess unique elements that are truly novel and interesting. Thus, the work is not suitable for publication in its current form, but I encourage authors to revise and resubmit their work after addressing the points below.

We sincerely thank the reviewer for their thorough review, positive comments, and encouragement to resubmit the study. We appreciate the thoughtful insights provided, as they have significantly contributed to enhancing the quality of our study, and we apologize for any

confusion or lack of clarity in conveying the uniqueness of our proposed method and its novel and significant findings.

In response to the reviewer concerns, we have clarified in the abstract and throughout the manuscript which aspects of our findings are novel and how they differ from existing models. Specifically, we have emphasized the unique elements of our modeling approach, particularly the novel implementation of axonal reorientation, and provided a more balanced assessment of the study's impact in light of past work. These revisions ensure that our contributions are accurately positioned within the broader context of existing research, and we hope they will address the reviewer's concerns.

Major concerns

1. Authors should clarify what aspects of their model and analysis are novel, as opposed to extension or replication of previous work. Much of the method related to towed growth appear similar to Garcia et al. 2021, but are not cited as such. Use of previously reported theoretical frameworks, equations, parameter choices, as well as comparisons to human / animal models, should be clearly contextualized as such.

Thank you to the reviewer for this constructive comment. The key novelty of our proposed model, as the reviewer later acknowledges, lies in the introduction and implementation of the concept of axonal reorientation, which is being explored for the first time. Additionally, this study uniquely models the dynamic growth of individual axon bundles, whose lengths are time-dependent and explicitly navigate within the extracellular matrix (ECM). Furthermore, the developed computational model presents a novel framework that enables the practical implementation of this concept. We acknowledge that stress-dependent growth of axons is a well-established concept in brain mechanics, as proposed and implemented in several studies⁸⁻¹⁰, predating Garcia et al.¹¹. However, previous studies have modeled white matter as a continuum medium exhibiting stress-dependent growth behavior. In contrast, our study introduces an innovative approach by explicitly separating the white matter into growing axon bundles and the extracellular matrix (ECM). Our model accounts for the growth of axon bundles influenced by other biological mechanisms, such as molecular axon guidance factors¹². This is represented as a stochastic tip growth mechanism, which has not been considered in previous studies. In the revised manuscript, we have thoroughly addressed this issue by clearly differentiating our proposed concept and its unique findings from previous studies. We have made specific revisions to highlight the novel aspects of our approach and to clarify how our study builds upon and extends existing research. Specifically, we have carefully contextualized our use of previously reported theoretical frameworks, equations, and parameter choices, ensuring that it is clear when we are building on established work and when we are introducing novel ideas. However, in the revised manuscript, we have avoided overusing terms like "novel," "new," and "first" throughout the text, in line with the guidelines for Nature formats. Instead, we have subtly emphasized the originality of our work and the significance of our findings.

We have cited Garcia et al.¹¹ multiple times throughout the manuscript, which is the most relevant study to our current work, discussed their results, and compared our findings with theirs. In the revised manuscript, we have addressed this issue by incorporating new discussions that specifically relate to previous studies, including Garcia et al.¹¹ and a recently published work by Wang et al.¹³ (2024).

- **Introduction:** “However, there is currently no model to explain how growing axon bundles navigate the stress field within the folding brain, or how the bidirectional, dynamic interaction between gyrification and connectivity shapes the resulting folding morphologies and connectivity patterns.”

“In this study, we propose the concept of “axon reorientation” in the folding brain and formulate a mechanical model to uncover the multiscale mechanics of the linkages between cortical folding and connectivity development. This model is the first to simulate the growth and pathfinding of individual axon bundles within the dynamic stress landscape of a folding brain. In contrast to previous studies that focused on passive fibers^{9,13–15} or active continuum-based fibers¹¹, our framework uniquely captures the intricate behavior of axon bundles, including stress-induced and stochastic growth, as well as the newly proposed reorientation response. The concept of “axon reorientation” in the folding brain originates from basic experiments demonstrating that neurites preferentially orient toward the direction of stretching^{16,17}. We hypothesize that, in addition to stress-induced growth, axons reorient towards the direction of maximum principal tensile stress to establish stereotyped subcortical fiber patterns. Our simulations, incorporating this complex axon bundle growth and pathfinding, offer a more accurate representation than previous models, particularly in explaining why gyri have a higher axon density than sulci. These predictions are well-supported by in vivo diffusion tensor imaging of the human brain.”

- **Discussion:** “Our findings are closely aligned with those of Garcia et al.¹¹, demonstrating that an expansion-driven model of cortical folding generates subcortical stresses capable of explaining the anisotropy patterns observed in the developing white matter during brain folding. However, in their model, the subplate is treated as a continuum medium, with its reorganization and anisotropy solely driven by stress induced by cortical folding. This approach overlooks the impact of subplate development on the formation and positioning of cortical folds. Our findings suggest that the heterogeneity of white matter, resulting from the gradual growth of individual stiff fibers, influences the folding patterns and dictates the location of gyri and sulci, even if the fibers do not induce cortical folding. This leads to the formation of gyri in areas with dense fibers and sulci in regions with lower fiber density. The bidirectional and dynamic relationship between cortical gyrification and connectivity development, highlighted by our results, has not been fully explored in previous studies.”

2. A novel aspect of the proposed is implementation of axon rotation to the primary stress direction, which results in interesting and compelling behavior. Authors claim that this addition accelerated/accentuated the accumulation of radial fibers in gyri. However, the accumulation of radial fibers within gyri has also been demonstrated in models without this feature. This includes (1) models of simple folding deformations in a substrate containing discrete axons (e.g., Chavodhnejad et al. 2021) as well as (2) models of towed growth that considered radial fiber density as part of a composite tissue definition (Garcia et al 2021). In the Introduction and Discussion, it would be useful and important to compare and contrast the current model, in light of these previous studies.

Thank you to the reviewer for acknowledging the novelty of the work and raising a very critical and constructive comment. We fully agree with the reviewer that “*previous models implementing towed growth or passive axons have already demonstrated how these mechanisms can result in higher density of axon bundles in gyri than in sulci.*” However, previous studies, including our own work with discrete fibers (Chavoshnejad et al.¹⁵) and the continuum-based towed growth study by Garcia et al.¹¹, have not explored the specific mechanisms that could explain the significant difference observed in the density of fibers between gyri and sulci in the real brain, such as in

thalamocortical fiber tracts. Our previous model demonstrated that with passive fibers, even under optimal conditions, the fiber density in gyri and sulci is only around 60% and 40%, respectively, assuming the elimination of fold walls and redistribution of fibers between gyri and sulci.

In fact, the motivation for this study stemmed from the absence of a mechanical model that explains the growth and navigation of explicit axon bundles in the folding brain. In the Introduction and Discussion sections, we have distinguished our model from previous studies, providing a detailed comparison (please see the response to the previous comment and associated revisions).

- **Discussion:** “These results align with those reported by Holland et al.⁹ and Wang et al.¹³ which confirm the important role of axonal fibers as both symmetry breaker and regulator of folding patterns. They showed that the inclusion of passive axon bundles can influence fold placement, and that altering the material properties of multiple axon tracts can significantly change folding patterns¹³. We have previously demonstrated that, in the context of strain energy minimization, a cortex on a substrate with predefined passive radial fibers tends to buckle, which results in a concentration of fibers in gyri rather than sulci¹⁵. However, in models with passive stiff fibers, even under optimal conditions, the fiber density in gyri and sulci remains only about 60% and 40%, respectively, assuming the removal of fold walls and the redistribution of fibers between gyri and sulci. Therefore, recent models with passive fibers^{9,13–15} or dynamic continuum-based fibers¹¹ lack to realistically model the growth, navigation, and establishment of axonal connectivity.”

3. An important limitation of the current model is the failure to implement or consider tangential fibers. Reasoning for omitting these fibers should be provided, and omission should be described in the discussion/limitations section.

We sincerely thank the reviewer for highlighting this critical point. As noted in the manuscript, our current study primarily focuses on modeling the growth and reorientation of axon bundles that move radially from the core of the brain towards the cortical plate, specifically targeting radial fibers. These radial fibers are crucial for understanding the fundamental aspects of axon pathfinding in relation to cortical folding. However, as the reviewer has rightly pointed out, tangential fibers (short- and long-association fibers), also play a role in cortical organization and connectivity. These fibers differ in orientation and function from radial fibers, contributing to the complex architecture of the brain.

Incorporating tangential fibers into our model is indeed feasible and would provide a more comprehensive representation of axonal growth and connectivity. However, the inclusion of these fibers would significantly increase the complexity of the model, potentially leading to challenges in extracting statistically meaningful conclusions. Our primary objective in this study is to replicate the growth of thalamocortical fiber tracts, which are predominantly radial in nature and navigate from the thalamus towards the cortical plate. The introduction of tangential fibers would necessitate a more intricate organization of fibers within the model, which could obscure the specific mechanisms underlying radial fiber growth and its interaction with the folding cortex.

In response to the reviewer’s comment, we have expanded the discussion on this limitation in the revised manuscript, addressing it specifically in the Methods and Limitations sections. We also acknowledge that future extensions of our model could incorporate these additional fiber types to provide a more holistic understanding of axonal growth and cortical folding. Such extensions

would benefit from the insights gained from our current focus on radial fibers, serving as a foundation for more complex models that include the full spectrum of axonal orientations.

- **Methods:** The model is limited to vertically growing axon bundles (radial fibers) to replicate the growth patterns of axon bundles, such as thalamocortical fiber tracts, that extend from the core of the brain toward the cortical plate. However, the organization of white matter in the brain is far more complex, encompassing not only radial fibers but also tangential fibers (short- and long-association fibers). In this study since the primary objective is to replicate the growth of thalamocortical fiber tracts, which are predominantly radial in nature and navigate from the thalamus towards the cortical plate, we did not consider tangential fibers in the models to focus on the fibers that navigate towards the cortical plate and gradually experience the induced stress field as the result of cortical folding.
- **Limitations:** In addition to radial fibers, tangential fibers (short- and long-association fibers), also play a role in cortical organization and connectivity¹³. These fibers differ in orientation and function from radial fibers, contributing to the complex architecture of the brain. The introduction of tangential fibers would necessitate a more intricate organization of fibers within the model, which could obscure the specific mechanisms underlying radial fiber growth and its interaction with the folding cortex. Therefore, tangential fibers were excluded from our models for simplicity, allowing us to focus on extracting meaningful statistical correlations between the parameters used. However, incorporating tangential fibers would provide a more comprehensive representation of axonal growth and connectivity. Extending our model to a true-scale representation that includes these additional fiber types will be the focus of our future research.

4. Similarly, the current study relies on tractography-based comparisons to human data, which are notoriously biased to capture radial fibers in gyri. Authors should describe limitations related to tractography in the Discussion section.

We thank the reviewer for raising this insightful comment. We acknowledge that tractography has inherent limitations, particularly its bias towards capturing radial fibers in gyri¹⁸. In the revised manuscript, we have added a discussion to address these limitations and clarify the potential impact on our findings.

- **Discussion:** “However, it should be noted that while fiber tractography is a powerful tool for visualizing white matter pathways, it has well-known limitations, particularly in accurately capturing the complexity of fiber orientations. It tends to bias towards detecting radial fibers in gyri, which can be problematic in regions of the cerebral cortex with intricate folding and convolutions¹⁸. While our study used tractography-based data for model comparison, we recognize that this method may not fully represent the complex organization of axonal pathways. However, despite these limitations, recent integrated tractography and histological tract-tracing studies support its practical performance in estimating structural connectivity^{19–21}.”

5. All models of the current manuscript are limited to a 2D rectangular space that, while commonly applied in simple studies of brain folding, has important limitations. Most critically, fixed or symmetry constraints along the bottom edge and sides will lead to stresses that may not reflect – or will less realistically reflect – stresses in the developing brain. Authors are encouraged to explore the stress state in more realistic, even if still 2D, representations of the brain, such as

circles/ellipses or even ellipsoids via axisymmetry. In either case, limitations of this framework should be discussed.

We sincerely thank the reviewer for this critical comment. We fully agree that 2D model simulations have inherent limitations in capturing the complex 3D morphologies of the folding brain. We have discussed this limitation in the manuscript: *“The 2D models in this study served as proof of concept, which offers simplicity in explaining the underlying mechanisms involved in the interplay of cortical folding and connectivity. However, given that cortical folding and connectivity development occur in the 3D space of brain development, 3D models can offer more accurate representations of this complex process. Therefore, future studies should consider utilizing 3D models, because they provide better opportunities to compare results, especially regarding the 3D convolution of folds with imaging data. This aspect will be investigated further in our future studies.”*

As mentioned in the previous paragraph, for our proof-of-concept study, we opted for a 2D model similar to other foundational studies such as those by Garcia et al.¹¹, Toro et al.²², or Bayly et al.⁸. This approach offers a better opportunity to test our proposed hypothesis. We acknowledge the reviewer’s concern about boundary conditions potentially altering the stress and deformation fields, which could, in turn, affect the distribution of axon bundles. However, it is important to note that when investigating properties far from the boundary, the influence of boundary conditions can be minimized. For instance, in our model with a fixed boundary at the bottom, the stress field from cortical folding dissipates before reaching the boundary, and the model’s bottom layer thickness is sufficient to minimize boundary effects as axon bundles enter the stress field far from the bottom boundary. A similar principle applies to the side symmetric boundaries; when focusing on the system’s response away from these boundaries, the growing and navigating axon bundles exhibit consistent behavior.

We sincerely appreciate the reviewer’s intriguing suggestion to explore extending the model to a circular geometry. A circular model not only mitigates side boundary effects but also more closely resembles the human brain’s structure, particularly with respect to radial fibers. We have taken this opportunity to develop a circular model using the same parameters as the rectangular model. The results, which we have included here and in the Supplementary Information of the revised manuscript, demonstrate similar observations to those in the rectangular model. We plan to extend this study to include 3D modeling in future research. We included the following explanation in the manuscript to reflect the observations we made using the circular models.

“It should be noted that the model size was chosen to be sufficiently large relative to the wavelength of the folds to minimize the impact of boundary conditions (fixed boundary at the bottom and symmetric boundaries at the sides). However, to ensure that the observed results are not dependent on the selected model geometry, we also developed circular models with radial fibers using the same parameters as in the rectangular models. A sample result is presented in Fig. S2, where two circular models with different axon growth rates are compared. As shown, the growth and pathfinding of fibers and their settlement in gyri rather than sulci follow a similar trend to those in the rectangular model, confirming their independence from boundary effects.”

Fig. S2. Dynamic growth and pathfinding of fibers within a growing and folding 2D circular system for two different growth rates (a) 0.6 mm/d , and (b) 1.0 mm/d . As shown, the fibers grow and orient more toward the gyri than the sulci, consistent with the behavior observed in the rectangular models.

6. The current description of the white matter constitutive model is confusing. First, authors describe an inner subcortical core “primarily composed of neuronal axons,” then as a combination of axons and ECM. The terminology in Methods could be clarified. Notably, the developing subcortical tissue is not primarily composed of neuronal axons – rather, at early stages, the subcortical tissue is dominated by radial glial cells/scaffolds, extracellular matrix, and other cells. If the “white matter” tissue was indeed mostly axons, justify this choice and describe in Limitations.

We thank the reviewer for this critical comment and apologize for any confusion regarding the material properties and structure of the subcortical, or more generally, the white matter region in our model. To address this, we have revised the terminology in the Methods section.

We fully agree with the reviewer that the composition of the early-stage developing subplate differs from that of the later-developing white matter. However, distinguishing between these two domains and selecting only one as the substrate for cortical folding is not straightforward. The primary reason is that cortical folding occurs over several months, during which the subplate, with its radial glial cells, evolves into white matter containing growing and establishing axons. Therefore, the properties of the underlying layer of the cortical plate change over time.

In our model, we primarily consider the growth of axon bundles from thalamocortical fiber tracts, as they begin to develop during the third trimester, coinciding with cortical folding. Please refer to Table S1 for the timeline of thalamocortical fiber tract growth and establishment. Consequently, in our model, we treat the growing fibers as axon bundles rather than radial glial cells. Despite this consideration, from a modeling perspective, we can treat axons and other structures, such as radial glial cells, all as fibers with distinct material properties compared to the ECM¹¹. Garcia

et al.¹¹ also have used the term “*axons and other processes, such as radial glial scaffolds*”, without differentiating between these elements. It is also important to note that very little data is available regarding the mechanical behavior of radial glial cells, which are eventually eliminated as they differentiate into other cell types during cortical folding. In the absence of quantitative data characterizing these processes, making assumptions would be misleading. We justified our choice in the Methods section and discussed it in the Limitations section of the revised manuscript.

- **Methods:** It should be noted that the properties of the layer underlying the cortical plate evolve over time, particularly during the cortical folding process, which spans several months. During this period, the subplate, initially composed of radial glial cells, transforms into white matter, which includes growing and establishing axons. Our model primarily focuses on the growth of axon bundles from thalamocortical fiber tracts, as they begin to develop during the third trimester, coinciding with the onset of cortical folding (Table S1). Therefore, in our model, we represent the growing fibers as axon bundles rather than radial glial cells. Nonetheless, from a modeling perspective, both axons and radial glial cells, can be treated as fibers, each possessing distinct material properties relative to the ECM¹¹.
- **Limitations:** In our model, we treated the growing fibers as axon bundles that extend and navigate through the ECM from the brain’s core toward the cortical plate, such as thalamocortical fiber tracts. However, it is important to acknowledge that the composition and mechanical properties of the layer underlying cortical plate undergo a dynamic and complex evolution during the process of cortical folding. In the early stages of brain development, the subplate evolves from a loosely structured layer of cell bodies, radial glial scaffolds, and neuronal processes into a more organized, axon-dense white matter tissue²³. It is also important to note that very little data is available regarding the mechanical response of radial glial cells, which are eventually eliminated as they differentiate into other cell types during cortical folding, warranting further investigation.

7. Perhaps related to the above point, authors describe growth of the white matter as “elastic” and isotropic. The description of growth as “elastic” is confusing to this reviewer – perhaps they meant inelastic? This is an important distinction. Also, is the “isotropic” growth separate from the axon elongation described elsewhere? How do these fit together?

Thank you to the reviewer for this important observation. We acknowledge that the term “elastic” growth is confusing in the manuscript. We have revised the text to clarify this point. We also revised manuscript to distinguish between white matter and ECM. The isotropic growth is only applied in the ECM part and tip and stress-induced growth are applied to axon bundles. However, at each iteration, the same isotropic growth is applied to both the ECM elements and the established axon elements. This isotropic growth helps to mitigate shear effects that might arise between the established, non-growing axon elements and the expanding ECM.

- **Methods:** “The new tip point is generated at each iteration according to Eq. (27), and the ECM element containing this tip point is transformed into an axon element. This process is repeated until the final time step. Therefore, in the model and at any given time, each element can either be an axon element or an ECM element, but not both (Fig. 2(d)). Additionally, it should be noted that at each iteration, the same isotropic growth is applied to both the ECM elements and the established axon elements. This isotropic growth helps

to mitigate shear effects that might arise between the established, non-growing axon elements and the expanding ECM.”

8. Table 2 has some important inaccuracies. I encourage authors to double check these fields.

Thank you to the reviewer for this valuable observation. We have addressed the concerns in Table 2 of the revised manuscript, and for your convenience, we have included the updated table below.

Table 2. Parameter range for neurite elongation based on available literature

Cell type	Reported axon caliber D (μm)	Stretch-dependent elongation rate (mm d^{-1})	Force-dependent elongation rate b ($\mu\text{m pN}^{-1} \text{h}^{-1}$)	Calculated stress-dependent elongation rate a ($\text{mm Pa}^{-1} \text{d}^{-1}$)
Embryonic rat dorsal root ganglia ²⁴	0.9	2.0	–	0.0004 – 0.02 *
P0-P1 murine hippocampal neurons ²⁵	5.0	–	0.66	0.31 **
Embryonic chick forebrain neurons ²⁶	1.0	–	0.05 – 0.5	0.001 – 0.01 **
Embryonic chick dorsal root ganglia ²⁷	2.0	–	0.02 – 0.55	0.001 – 0.04 **

* In studies that measured stretch-dependent elongation rates, the stress-dependent elongation rate, a , was estimated using the relation $a = G^{axn}/E$, following the approach of Garcia et al.¹¹. The Young's modulus of individual neurites, which varies between 100 and 4600 Pa^{28,29}, provides a corresponding range for the stress-dependent elongation rate.

** In studies that reported force-dependent elongation rate, the parameter a was determined using the formula $a = bA$, where A is the cross-sectional area of the neurite.

Minor concerns:

1. In introduction, authors state, “This hypothesis has been supported by recent experimental and computational studies,” and do cite several more recent studies (11,15–21). However, this hypothesis has a much longer history, and is supported by studies dating back at least 50 years. It would be good to cite these original papers. A review of these can be found in the current papers referenced.

Thank you to the reviewer for this observation. We have cited the pioneering study on the concept of tangential differential growth (DTG) at its first mention, as the reviewer noted, dating back to 1975. Subsequently, we have cited supporting recent studies. We apologize for any missing relevant citations and have revised the manuscript to include all pertinent references.

“Different hypotheses and theories have been proposed to explain the underlying mechanisms of cortical folding (see the review by Arellano and Rakic)³⁰.”

“This hypothesis has been supported by past studies and recent experimental and computational investigations^{4,8,14,31–40}.”

2. On page 2, a reference is needed for the statement, “This aligns with the observation that gyri consist of a greater number of neurons compared to sulci.”

We thank the reviewer for this comment. We added a reference for the statement.

“This aligns with the observation that gyri consist of a greater number of neurons compared to sulci⁴¹.”

3. On page 3, the statement, “This hypothesis has been recently updated 10” is unclear, as it implies that the hypothesis of axon elongation in response to folding has been updated.

We thank the reviewer for this comment. We have revised the manuscript to clarify any confusion and have ensured accuracy by adopting the terminology used by Van Essen in his paper⁴².

“A recent revision of the tension-based morphogenesis model has been introduced to address the limitations of the original model⁴².”

4. On page 4, the statement, “The intricate interplay between growing axonal fiber bundles and stress-strain fields and the mechanisms that underlie the observed greater fiber density in gyri compared to sulci lacks clarity,” is redundant and perhaps overstated. I would recommend omitting this sentence.

We thank the reviewer for this comment. We removed the sentence.

5. On page 5, the statement, “Table S1 indicates that the major part of cortical folding and connectivity development of the thalamocortical fiber tracts occur concurrently,” could be omitted and instead Table S1 referenced in parentheses at the end of the previous sentence.

We thank the reviewer for this comment. We addressed the comment in the revised manuscript.

6. On page 6, the statement, “The imaging data indicate that cortical thickness undergoes changes during the early developmental stage,” should include a reference.

We thank the reviewer for this comment. We added a reference to support the statement.

“The imaging data indicate that cortical thickness undergoes changes during the early developmental stage⁴³.”

7. Fig. 2 panel a seems to suggest that direction changes will happen at the neuronal cell body. However, biologically I would imagine the cell body as the anchor point, and the other end as the axon tip as the point being guided by stress. Perhaps authors can clarify their intention with this figure and consider an alternative illustration.

We appreciate the reviewer pointing out this issue. As noted, axons extend and navigate with directional guidance from their growth cones. We have corrected this in the revised Figure 2, which is provided here for reference.

8. On page 13, isotopically should be isotropically.

We thank the reviewer for catching the typo. We addressed the comment.

9. The statement, "The effect of boundary conditions was eliminated by increasing the width of the model to $w = 60 \text{ mm}$ " should be adjusted, as increasing the width of the model does not eliminate the effect of boundary conditions completely. I suspect authors mean that this would eliminate one or more specific effects related to model width when the domain is smaller.

We thank the reviewer for this comment. We revised the text to clarify that as the width of the model increases, the effects of the boundaries on regions far from them become negligible. We have elaborated on this issue further in response to comment #5. As shown in Fig. S2, both the rectangular model with symmetric boundaries and the circular model exhibit similar behavior.

"The model width was chosen to be $w = 60 \text{ mm}$, which is sufficiently large relative to the wavelength of the folds¹⁵. This selection minimizes the influence of symmetric boundary effects on the folding patterns and the stress field induced by the folding process on regions far from the boundaries."

10. On page 18, “This observation aligns with previous studies on stresses within the developing brain” appears to have the wrong reference. Please check all in text citations throughout the manuscript.

We appreciate the reviewer for identifying this important issue. It appears that our referencing software inadvertently made some unintended changes to the references. We apologize for any inconvenience caused. We have corrected the references and thoroughly reviewed them to ensure proper citation.

11. The statement, “This equates to a sixfold increase in cortical surface area over a complete simulation period of 50 gestational days 37.” appears to have the wrong reference. Please check all in text citations throughout the manuscript.

Again, we apologize for any inconvenience caused. We have corrected the references and thoroughly reviewed them to ensure proper citation.

Reviewer #3 (Remarks to the Author):

This work introduces a multi-scale model for axonal growth that investigates its link to cortical folding. In a bilayer model with growing cortex and sub-cortex, the growing axons are hypothesized to reorient themselves along the direction of the maximum tensile principle stress in the white matter, which can be different from the initial direction of growth. Various parametric studies reproduce experimental observations (higher density of axonal fibers under gyri than sulci) and indicate potential inverse correlations between gyrification index and cortex:substrate stiffness ratio. Studies also indicate an inverse relationship between growth rate of axons to the gyrification index, further indicating that if the equivalent stiffness of the white matter (tissue + axon) increases, gyrification decreases. The paper makes important observations, but needs further improvement in the manuscript before publication.

We sincerely thank the reviewer for their positive feedback, thoughtful summary of our work, and constructive comments, which helped us further improve the manuscript.

Major concerns:

1. There is confusion on how the axon is modeled in the FE simulations. Can each element in the subcortex be both an axon and white matter? Fig 2 and Fig 7c suggest that an element can either be an axon OR white matter, not both. Later on, simulations account for equivalent stiffness. Further, the visualization of how an axon grows seems to indicate that axon is smoothly reorienting itself, which suggests that an element can be both. Can the authors clarify this?

Thank you to the reviewer for this critical comment. We apologize for any confusion.

In our model, each element can either be a fiber element (with stiffer behavior) or an ECM element (with a softer response) not both. Initially (at $T=0$), all elements are ECM elements with lower stiffness. As the simulation progresses, fibers are initiated randomly within the ECM elements. These ECM elements then transform into fiber elements, which have higher stiffness. In subsequent iterations, stress-dependent growth and tip growth are applied to the axon elements to predict their paths. The predicted paths then cause further transformation of ECM elements into fiber elements, continuing until the final time step. Thus, at any given time, an element can only be one type, either fiber or ECM. We modeled the white matter as two separate components:

fibers and ECM. However, we used a unified finite element model to represent the interaction between these domains, mirroring real brain dynamics. The transformation process is illustrated in Movies 2 and 3, which show fibers navigating by converting ECM elements (blue) into fiber elements (red). Figure 7(c) also demonstrates that elements are either fibers or ECM, not both. Initially, all elements are ECM (volume fraction of 1), but as time progresses, some ECM elements transform into fibers, reducing the ECM volume fraction and increasing the fiber volume fraction. The total volume fraction of fibers and ECM always sums to 1. We revised the manuscript to clarify this.

“The new tip point is generated at each iteration according to Eq. (27), and the ECM element containing this tip point is transformed into an axon element. This process is repeated until the final time step. Therefore, in the model and at any given time, each element can either be an axon element or an ECM element, but not both (Fig. 2(d)). Additionally, it should be noted that at each iteration, the same isotropic growth is applied to both the ECM elements and the established axon elements. This isotropic growth helps to mitigate shear effects that might arise between the established, non-growing axon elements and the expanding ECM.”

We defined “equivalent stiffness” to indicate the proportion of the bottom layer of the model occupied by stiff fiber elements versus softer ECM elements. Therefore, the concept of “equivalent stiffness” applies to the entire bottom layer, encompassing both stiff fiber elements and soft ECM elements, rather than to individual elements. This concept has been clarified in the revised manuscript. Additionally, we have prepared an exciting new figure (Fig. 7(b)) that illustrate how the initiation and growth of fibers create a dynamic, heterogeneous stiffness map in the white matter region.

“In the 2D scenario, the fiber volume fraction is determined by the ratio of the surface area occupied by fiber elements within a specified region to the total surface area of that region. Local stiffness ratio is defined as the fiber volume fraction within a 1 mm radius circle centered around each material integration point, plus one, as given by $V_f \left(\frac{\mu_f}{\mu_s} \right) + V_{ECM} \left(\frac{\mu_s}{\mu_s} \right) = V_f + 1$ for models with $\frac{\mu_f}{\mu_s} = 2$. Equivalent stiffness ratio is defined as $V_f \left(\frac{\mu_f}{\mu_s} \right) + V_{ECM} \left(\frac{\mu_s}{\mu_s} \right)$ across the entire white matter region, updated at each simulation timestep. Fig. 7(a) depicts models with varying fiber counts, ranging from 100 to 400, while Fig. 7(b) illustrates the local stiffness ratio map of the model’s white matter region, resulting from the initiation, growth, and establishment of these fibers. As the volume fraction of stiff fibers increases, the local and overall stiffness of the bottom layer increases, resulting in a heterogeneous stiffness map. This behavior mirrors the stiffness map characteristics of white matter tissue as predicted by magnetic resonance elastography (MRE) studies⁴⁴. Figures 7(c) and (d) illustrate the dynamic changes in fiber volume fraction and, consequently, the equivalent stiffness of the white matter region. Over time, both the fiber volume fraction and the equivalent stiffness increase and stabilize once the fibers reach the cortical plate and new fiber generation ceases. The increased stiffness in the white matter region leads to a reduction in cortical gyrification and a lower GI (Fig. 7(e)).”

Fig. 7 (a) Effect of number of fibers on cortical folding and fiber organization. The parameters used are as follows: $G^{axn} = 0.8 \text{ mm/d}$, $\mu_f/\mu_s = 2$, $\mu_c/\mu_s = 2$, $a = 0.015 \text{ mm Pa}^{-1} \text{ d}^{-1}$. (b) Effect of fiber volume fraction on local stiffness ratio. (c) Change in overall volume fraction of fibers (V_f) and ECM (V_{ECM}) as a function of time for the case with 400 fibers. Fiber volume fraction starts from zero and gradually increases, while the volume fraction of ECM decreases. The total volume fraction of fibers and ECM always sums to 1. (d) Change in the equivalent stiffness ratio of the white matter as a function of time. (e) Dependency of the GI to the number of fibers in the white matter region. p-values are displayed on the figure, where a significance level of 0.05 is considered.

To enhance the clarity of the figures, we developed a MATLAB code that traces each fiber path, originally composed of discrete elements, and replaces it with a spline curve. This approach provides a clearer depiction of the fibers' navigation and reorientation paths. However, in the provided movies (e.g., Movies 2 and 3), which show the finite element models without modifications, the fiber paths are represented as growing chains of discrete segments. This clarification has also been included in the revised manuscript.

“Movies 2 and 3 illustrate the dynamic growth and pathfinding of fibers at two distinct axon growth rates. In the movies, which show the finite element models without modifications (no fitting splines), the fiber paths are represented as chains of discrete segments.”

2. The equations for reorientation (20-23) as well as the algorithm provided suggest that the process is deterministic. Yet, later on in the text, there is mention of gaussian noise (with zero mean) added. This should be clarified in both the text and the equations.

Thank you to the reviewer for this critical comment. In our model, the pathfinding of fibers involves two components: one deterministic, related to stress-dependent growth, and the other stochastic,

associated with tip growth. The deterministic part represents the axons' response to mechanical forces, while the stochastic part reflects their response to other stimuli, such as biochemical cues, which have a random nature⁴⁵⁻⁴⁷. We included this stochastic component to avoid the biased conclusion that axon navigation and pathfinding are driven solely by mechanical forces. Numerous studies have shown that axons, when provided with nutrients or exposed to biochemical cues, can grow and even exhibit directed growth. Our model aims to replicate real conditions where axonal growth results from both deterministic and stochastic mechanisms. We have clarified this in the revised manuscript.

“Additionally, we introduced centered Gaussian noise into \mathbf{n}_k^A to account for the stochastic nature of the axon growth direction, which will be discussed further later.

$$\mathbf{n}_k^A = \cos(\theta + \epsilon) \mathbf{i} + \sin(\theta + \epsilon) \mathbf{j} \quad (20)$$

where θ is the angle between \mathbf{n}_k^A and the x-axis, and ϵ represents the Gaussian noise.”

“To account for the inherent randomness of axon growth, we introduced Gaussian noise (Eq. (20)) into the preferred direction, enabling stochastic variations. This randomness in the growth direction is applied to the tip growth of the axon bundle, which is further discussed in the next section. Consequently, in the absence of a stress field, or when the maximum principal stress is compressive, the axon bundle grows randomly. However, in the presence of a stress field, the axon bundle gradually aligns with the direction of the maximum tensile principal stresses.”

3. Related, in Fig. 11 and in the relevant section, 'Without Reorientation' really means that the reorientation is not stress-driven (rather due to the small gaussian noise at each step). If this is the case, the terminology should be clarified.

Thank you to the reviewer for bringing this to our attention. We acknowledge that the term “Without Reorientation” may have caused confusion, as axons can reorient even in the absence of mechanical forces. In the revised manuscript, we have changed the terms “With Reorientation” and “Without Reorientation” to “With Stress-Dependent Reorientation” and “Without Stress-Dependent Reorientation” to clarify this distinction.”

4. The method/equation for calculating gyrification index is missing; it is also important to carefully define how the convex hull is found.

Thank you to the reviewer for this constructive comment. In the revised manuscript, we have included the definition and equation for calculating the “gyrification index”. Additionally, we added a schematic figure to visually illustrate the concept of the gyrification index, which we have copied here for your reference. Furthermore, we have also detailed the method used to calculate the convex hull.

“The GI is defined as the ratio of the total length of the cortical contour (represented by the solid line in Fig. 6(c)) to the length of the convex hull (shown by the dashed line in Fig. 6(c)). This ratio indicates the extent of gyrification and surface complexity. The convex hull is determined by fitting a smoothed, curved line that minimally encloses the outermost points of the cortical surface.”

Fig. 6 (a) Effect of stiffness ratio of axon to white matter ECM (μ_f/μ_s) on cortical folding and fiber organization. Ratio 1 indicates that there is no difference between the material properties of axons and ECM, which creates an isotropic substrate for the folding of the cortex. All other parameters remained constant for all scenarios ($G^{axn} = 0.8 \text{ mm/d}$, $\mu_c/\mu_s = 2$, $a = 0.015 \text{ mm Pa}^{-1} \text{ d}^{-1}$). (b) Fiber density within gyri with respect to the various stiffness ratios (μ_f/μ_s). Changing the stiffness ratio significantly affects the fiber density in gyri and sulci. p-values for significance levels are shown in the graph. (c) Effect of the axon's material properties on the gyration index (GI) of the formed folds.

5. The paper would be much clearer with a table of the values/ranges of all the parameters used in the simulations.

Thank you to the reviewer for this helpful comment. In the revised manuscript, we have added a table that lists all the parameters used in the simulations, along with their corresponding values and ranges. For your convenience, we have copied it here for your reference.

Table 3. Parameters and their values used in the simulations

Parameter	Cortical thickness (mm)	Axon growth rate (mm d ⁻¹)	Stiffness ratio of the cortex to the ECM (μ_c/μ_s)	Stiffness ratio of fibers to the ECM (μ_f/μ_s)	stress-dependent axon elongation rate a (mm Pa ⁻¹ d ⁻¹)	Standard deviation of the growth angle (Noise) (rad)
Values	1.5	0.6, 0.8, 1.0, 1.2	1, 2, 3, 4	1, 2, 4, 5	0.015	0.025 – 0.125

6. Code and data should be shared via Github (only data is mentioned, and has not yet been shared).

Thank you to the reviewer for this comment. We included the associated codes with the initial submission, but we apologize for any confusion regarding the availability of data and codes. We have revised the “Data Availability” section as follows: The codes and data supporting the findings of this study are available on the research group’s Github account.

“Data availability

The codes and data supporting the findings of this study are available on the research group’s Github account (<https://github.com/MechanicsofSoftBioMaterialsLab/Axonal-PathFinding>).

Minor concerns:

Thank you for your detailed review and thoughtful minor comments. We have addressed all of them in the revised manuscript, ensuring clarity and accuracy throughout. We appreciate your careful consideration and believe these revisions have strengthened the overall quality of the paper.

1. P22: " In Fig. 6(a), a stiffness ratio of 1, which creates an isotropic substrate for folding of the cortex, has no effect on the morphology of cortical folds." This should be reworded; the stiffness ratio does affect the morphology, but the axons don't.

Addressed: “In Fig. 6(a), a stiffness ratio of 1, which creates an isotropic substrate for folding of the cortex, fibers have no effect on the morphology of cortical folds.”

2. There are several notation issues that the authors need to pay attention to:
2.1 The notation for derivative w.r.t time (D_t vs $\dot{\theta}$ vs $\partial D/\partial t$). Addressed.

2.2 The normals defined in the algorithm and in equations 15-19 are going to be updated at each time step 'k', hence each of them require a subscript 'k'. Addressed.

2.3 Likewise with the variables ω and R (eq. 20). Addressed.

2.4 The basis vectors e_x and e_y on pg 13 (top) should be defined. Addressed.

2.5 Units should be consistently used, e.g. day or d but not both. Addressed.

2.6 The caption of Fig 7 uses the 'x' symbol which in the text implies the cross product. Addressed.

2.7 If the authors mean that higher order tensors are in bold and scalars are in unbold, then the variables epsilon and omega in eq 20 should be in bold. Addressed.

3. There are several typos as well that need to be corrected:

3.1 Section number 2.1 is used twice. The proceeding sections should also be renumbered. Addressed.

3.2 On pg 26, there is a line break that divides a sentence into two paragraphs. Addressed.

3.3 The thickness of the cortex t_c is shown as 1.5 m, which should likely be millimeters. Addressed.

3.4 reference #9 should be Van Essen, not Essen.

Thank you to the reviewer for bringing this to our attention. We used referencing software to automatically add the citations, but unfortunately, the issue was with the source of the citation. We have corrected this in the revised manuscript and thoroughly checked all the references to ensure accuracy.

3.5 In the first sentence of the introduction, I believe you are referring to neural tubes, not neuronal tubes.

Thank you to the reviewer for bringing this to our attention. We corrected the term in the revised manuscript.

4. Red-green color scales should be avoided due to issues for colorblind individuals.

Thank you to the reviewer for bringing this to our attention. We revised Figures 1, 3, and 11 to address the comments.

Figure 1, revised. The images were generated using the open-access tool FED1 (<https://github.com/IntelligentImaging/FED1>) and shared longitudinal fetal brain scans provided by Boston Children's Hospital under a data transfer agreement [Motion-robust super-resolution diffusion weighted MRI of early brain development] with Binghamton University.

Figure 3, revised. The MRI and diffusion MRI images shown in this figure were obtained from the publicly available Q1 release of the WUMinn Human Connectome Project (HCP) database (<https://humanconnectome.org/>)

Figure 11, revised. The MRI and diffusion MRI images shown in this figure were obtained from the publicly available Q1 release of the WUMinn Human Connectome Project (HCP) database (<https://humanconnectome.org/>)

Reviewer #3 (Remarks on code availability):

No code was included. The manuscript mentions that the data will be available from their github page, but there's actually no mention of sharing the code even after publication.

Thank you to the reviewer for this comment. In the initial submission, we included sample codes, but it seems they were not accessible to the reviewers. To ensure that the codes and data are available to other researchers, we have now uploaded them to the research group's GitHub account. We have also updated the "Data Availability" section to inform readers that the codes and supporting data are now accessible through the research group's GitHub account.

"The codes and data supporting the findings of this study are available on the research group's GitHub account. (<https://github.com/MechanicsofSoftBioMaterialsLab/Axonal-PathFinding>)"

Reviewer #4 (Remarks to the Author):

This manuscript investigated the interplay between cortical folding and brain connectivity development through a proposed mechanical model. The authors explored how growing axon bundles navigate within a dynamic stress field influenced by the folding cortex. There are several key findings, including the mechanical stress influence on axonal growth, the bidirectional relationship between the biophysical properties of axons and cortex patterns, and how axonal growth alters the material properties of white matter. The manuscript showed the critical role of mechanics in early brain development and suggests that understanding these interactions could provide insights into brain disorders characterized by abnormal folding and connectivity. However, the proposed model seems to be rough and confusing.

We sincerely thank the reviewer for their positive comments and for providing a concise and thoughtful summary of our work. We apologize for any confusion or lack of clarity in our presentation of the proposed model and its novel, significant findings. To enhance clarity, we have revised the Methods section and other relevant parts of the manuscript, with changes highlighted for your review. Additionally, we have thoroughly addressed feedback from all the reviewers, incorporating new figures and explanations to improve clarity. We hope these revisions have effectively resolved any remaining ambiguities and strengthened the overall presentation and impact of our study.

1. Although the introduction included an extensive literature review, it is still difficult to identify the novelty of this work.

Thank you to the reviewer for this critical comment. We apologize for the lack of clarity. In the revised manuscript, we have made a concerted effort to clearly distinguish the novelty and significance of this study from previous work. Below are paragraphs from the revised manuscript that highlight these distinctions for your reference.

- **Abstract:** "Understanding the mechanics underlying the connections between cortical folding and the development of brain connectivity is crucial, especially given that many brain disorders exhibit abnormal folding patterns and disrupted connectivity. Despite the importance of this relationship, existing models fail to explain how growing axon bundles navigate the stress field within a folding brain, or how this bidirectional and dynamic interaction shapes the resulting surface morphologies and connectivity patterns. Here, we propose the concept of "axon reorientation" in the folding brain and formulate a mechanical

model to uncover the dynamic multiscale mechanics of the linkages between cortical folding and connectivity development. Simulations incorporating axon bundle reorientation and stress-induced growth reveal the potential mechanical mechanisms that lead to higher axon bundle density in gyri (ridges) compared to sulci (valleys), offering a more accurate representation than previous models. In particular, the degree of gyrification is shown to correlate strongly with the density, growth rate, and mechanical properties of underlying axon bundles. Model predictions are supported by in vivo diffusion tensor imaging of the human brain. Our findings suggest that mechanics play a far more significant role in the healthy development of cortical folds and connectivity than previously recognized.”

- **Introduction:** “However, there is currently no model to explain how growing axon bundles navigate the stress field within the folding brain, or how the bidirectional, dynamic interaction between gyrification and connectivity shapes the resulting folding morphologies and connectivity patterns.”

“In this study, we propose the concept of “axon reorientation” in the folding brain and formulate a mechanical model to uncover the multiscale mechanics of the linkages between cortical folding and connectivity development. This model is the first to simulate the growth and pathfinding of individual axon bundles within the dynamic stress landscape of a folding brain. In contrast to previous studies that focused on passive fibers^{9,13-15} or active continuum-based fibers¹¹, our framework uniquely captures the intricate behavior of axon bundles, including stress-induced and stochastic growth, as well as the newly proposed reorientation response. The concept of “axon reorientation” in the folding brain originates from basic experiments demonstrating that neurites preferentially orient toward the direction of stretching^{16,17}. We hypothesize that, in addition to stress-induced growth, axons reorient towards the direction of maximum principal tensile stress to establish stereotyped subcortical fiber patterns. Our simulations, incorporating this complex axon bundle growth and pathfinding, offer a more accurate representation than previous models, particularly in explaining why gyri have a higher axon density than sulci.”

2. In Eq. (24), the authors provided a growth rate of the axon which add a fixed growth rate G^{axn} to the traditional stress-dependent growth rate. However, this makes no sense. In fact, after adding the growth rate G^{axn} , the desired target (homeostatic) stress σ_0 becomes $\sigma_0 - G^{axn} / a$. The authors set $\sigma_0 = 0$ and $G^{axn} > 0$, which corresponds to a negative homeostatic stress. Therefore, the growth rate Eq. (24) is inconsistent with the experimental investigations [Refs. 66-68].

Thank you to the reviewer for this insightful comment. The axonal growth rate proposed in our study [$\frac{\partial G}{\partial t} = a(\sigma - \sigma_0) + G^{axn}$] models axon extension as a combination of intrinsic and extrinsic factors. The term G^{axn} represents the intrinsic growth rate of the axon, reflecting its inherent ability to extend independent of external forces⁴⁸. This term captures baseline growth behavior, which varies widely according to different reports. The extrinsic component, $a(\sigma - \sigma_0)$, accounts for stress-dependent growth, where σ is the local stress experienced by the axon from the surrounding extracellular matrix, and σ_0 is the homeostatic stress. The constant a modulates the sensitivity of the axon to these external forces. When σ exceeds σ_0 , positive stress stimulates faster axonal extension, while in the absence of stress, the axon relies solely on its intrinsic growth. As illustrated in the figure below from Chang et al.¹⁶ neurites in a non-stretching field exhibit intrinsic growth but significantly increase their growth rate when subjected to tension, gradually orienting toward the applied force. Our model aims to replicate this behavior.

In our model, σ_0 was set to zero for simplicity, assuming the baseline growth rate is independent of initial stress, which is consistent with references^{11,25,49}. This does not imply that the actual biological homeostatic stress is negative. Rather, it reflects a baseline state where the axon grows at rate G^{axn} in the absence of external stress (i.e., $\sigma = 0$). Our focus is on how external tensile forces from the ECM can accelerate axon growth and affect the pathfinding process. Notably, only tensile stress is considered in our model to affect both the growth rate and direction of growing axons. As stated in the manuscript, negative stress does not influence the length or orientation of growing axon bundles.

We fully agree with the reviewer that, with the current formulation, if an axon grows with a constant rate G^{axn} , theoretically, negative stress would be needed to bring the total growth rate to zero. However, as explained, our growth model is designed to account for both intrinsic growth in the absence of external forces and extrinsic growth under tensile forces, consistent with experimental observations^{16,25,26}, rather than exploring the effect of compressive stress on growing axons.

The optical microscope images of NSCs grew on non-stretching and stretching groups I, II and III, respectively. Neurite outgrowth and axon elongation were observed in different conditions for 1, 3, and 7 days. This figure has been adopted from Change et al.¹⁶ and presented here, to show that neurites grow even in the absence of external forces, and increase their growth rate when subjected to tensile forces and change their direction towards the applied force direction.

3. Many of the parameters used in the simulations in the study were not real data from the brains, but used some experimental data of other tissues. It is impossible to estimate whether these influenced the results.

Thank you to the reviewer for this critical comment. We fully agree that the parameters used in our model, along with their values or ranges, have not been calibrated from *in vivo* observations, and therefore, they may differ from the actual values in the developing brain. However, it should be noted that characterizing these parameters in a growing brain with developing axonal fibers and folding tissue poses significant challenges, if not making it nearly impossible. Consequently,

most brain folding models, including ours, rely on parameters derived from *in vitro* tissue experiments, which themselves may differ from *in vivo* properties.

To address this limitation, researchers in brain folding models typically conduct parametric analyses to study the impact of specific parameters on folding morphology or the correlation between gyrification and connectivity. A relevant example is the study by Garcia et al.¹¹ published in Nature Communications, which, similar to our study, used parameter values and ranges from various sources that were not exclusively derived for the study. Predictive models like ours are abundant in the literature, using parametric modeling to explore parameters that can exhibit a broad range of variations in reality, such as cortical growth rates, axonal growth rates with and without axial forces, and the stiffness of axons, cortex, and white matter. The following studies have also employed this approach

1. Bayly, P. V., R. J. Okamoto, G. Xu, Y. Shi, and L. A. Taber. "A cortical folding model incorporating stress-dependent growth explains gyral wavelengths and stress patterns in the developing brain." *Physical biology* 10, no. 1 (2013): 016005.
2. Tallinen, Tuomas, Jun Young Chung, François Rousseau, Nadine Girard, Julien Lefèvre, and Lakshminarayanan Mahadevan. "On the growth and form of cortical convolutions." *Nature Physics* 12, no. 6 (2016): 588-593.
3. Balouchzadeh, Ramin, Philip V. Bayly, and Kara E. Garcia. "Effects of stress-dependent growth on evolution of sulcal direction and curvature in models of cortical folding." *Brain Multiphysics* 4 (2023): 100065.
4. Budday, Silvia, Paul Steinmann, and Ellen Kuhl. "The role of mechanics during brain development." *Journal of the Mechanics and Physics of Solids* 72 (2014): 75-92.
5. Budday, Silvia, Charles Raybaud, and Ellen Kuhl. "A mechanical model predicts morphological abnormalities in the developing human brain." *Scientific reports* 4, no. 1 (2014): 5644.
6. Budday, Silvia, Paul Steinmann, and Ellen Kuhl. "Secondary instabilities modulate cortical complexity in the mammalian brain." *Philosophical Magazine* 95, no. 28-30 (2015): 3244-3256.
7. Wang, Xincheng, Shuolun Wang, and Maria A. Holland. "Axonal tension contributes to consistent fold placement." *Soft Matter* 20, no. 14 (2024): 3053-3065.

Therefore, in the Limitations section, we have explicitly stated that: "The *main limitation* of this study might be the utilization of parameter magnitudes that have been extracted from various sources, which may not be directly applicable to the context of cortical folding, because many were obtained from basic experiments. However, we attempted to address this limitation by conducting parametric studies aimed at quantifying the sensitivity of the observations to the parameters used."

Taking the reviewer's insightful comment into consideration, we have revised the manuscript to better emphasize the predictive nature of the models and their results. We have carefully modified the writing to clarify that the primary aim of our models is to offer a predictive framework rather than to draw definitive conclusions. This approach aligns with the inherent uncertainties and complexities of modeling brain development and acknowledges the variability of parameters that are not fully characterized *in vivo*. We believe these revisions will better convey the purpose and limitations of our study.

"It is important to note that all results and findings are derived from predictive models. These models suggest potential mechanisms that may apply to the developing brain, rather than

describing the exact process in the human brain. The terminology used for model constituents and parameters is aligned with that of the human brain solely to establish an analogy.”

4. The study simulated the interaction between cortical folding and axonal development in the developing brain, which displays clear 3D features. However, the proposed model was simplified to a 2D geometry.

Thank you to the reviewer for this critical comment. We fully agree with the reviewer that 3D models offer a more accurate representation of brain growth and folding. However, for proof-of-concept studies, 2D models are often employed because they provide the necessary simplicity to focus on the key parameters within the models. This approach is widely used in the field of the mechanics of brain folding, where 2D models are favored for fundamental studies, while 3D models are reserved for capturing the complete complexity of brain folding. Below are a few examples of seminal fundamental studies that have effectively used 2D domains or axisymmetric 2D models:

Bayly et al.⁸ (2013)

Toro and Burnod²² (2005)

Budday et al.⁵² (2014)

Holland et al.⁹ (2015)

Garcia et al.¹¹ (2021)

In the current study, we selected a 2D domain to propose our new theory and demonstrate its proof of concept. However, we have previously developed 3D models to study brain growth and folding, including the integration of passive fibers. The following image represents an example of one such 3D model. While 3D models are valuable for detailed simulations, they are less suited for theory development due to their complexity and the numerous parameters involved, compared to the more streamlined 2D models.

Chavoshnejad et al.¹⁴ (2023)

To support our point, we include the following quote from Reviewer #1: “*The model exhibits a nice balance between simplicity and complexity. The spatial domain is only 2D, and the properties are consistent throughout cortical and subcortical regions. This simplicity clarifies the roles of the factors that are includes: stress-dependent growth, axon re-orientation, and the separation of subplate into axons and “extracellular matrix” (ECM).*”

To ensure the reader understands the rationale behind selecting a 2D model and the associated limitations, we have explicitly discussed this in the Limitations section of the study: “*The 2D models in this study served as proof of concept, which offers simplicity in explaining the underlying mechanisms involved in the interplay of cortical folding and connectivity. However, given that*

cortical folding and connectivity development occur in the 3D space of brain development, 3D models can offer more accurate representations of this complex process. Therefore, future studies should consider utilizing 3D models, because they provide better opportunities to compare results, especially regarding the 3D convolution of folds with imaging data. This aspect will be investigated further in our future studies.”

5. Below Eq. (1), the authors stated that the gradient operator is applied with respect to the material point X in the reference configuration. In Eq. (1), however, the gradient operator is applied with respect to point x in the current configuration.

Thank you for your insightful comment. We have revised the text to accurately reflect that the gradient operator in Eq. (1) is applied with respect to point X in the reference configuration. We appreciate your attention to this detail and have made the necessary corrections in the manuscript.

6. Eqs. (3)-(4) and Eqs. (9)-(10) both refer to F^g , but they represent different quantities. To avoid misunderstanding, different notations should be used.

Thank you for pointing this out. We have revised the manuscript to use distinct notations for F^g in Eqs. (3)-(4) and Eqs. (9)-(10) to clearly differentiate between the two quantities. This change should help prevent any potential misunderstanding. We appreciate your careful review and attention to detail.

7. There are some inconsistencies between Fig. 5 and its citations on Pages 20 and 21.

Thank you to the reviewer for bringing this to our attention. We revised the manuscript and addressed the issue.

References

1. Budday, S. *et al.* Mechanical characterization of human brain tissue. *Acta Biomater.* **48**, 319–340 (2017).
2. Weickenmeier, J. *et al.* Brain stiffness increases with myelin content. *Acta Biomater.* **42**, 265–272 (2016).
3. Budday, S., Steinmann, P. & Kuhl, E. Secondary instabilities modulate cortical complexity in the mammalian brain. *Philos. Mag.* **95**, 3244–3256 (2015).
4. Budday, S., Steinmann, P. & Kuhl, E. The role of mechanics during brain development. *J. Mech. Phys. Solids* **72**, 75–92 (2014).
5. Budday, S. *et al.* Mechanical properties of gray and white matter brain tissue by indentation. *J. Mech. Behav. Biomed. Mater.* **46**, 318–330 (2015).
6. Budday, S., Ovaert, T. C., Holzapfel, G. A., Steinmann, P. & Kuhl, E. Fifty Shades of Brain: A Review on the Mechanical Testing and Modeling of Brain Tissue. *Arch. Comput. Methods Eng.* (2019) doi:10.1007/s11831-019-09352-w.
7. Walter, C. *et al.* Multi-scale measurement of stiffness in the developing ferret brain. *Sci. Rep.* **13**, 20583 (2023).
8. Bayly, P. V., Okamoto, R. J., Xu, G., Shi, Y. & Taber, L. A. A cortical folding model incorporating stress-dependent growth explains gyral wavelengths and stress patterns in the developing brain. *Phys. Biol.* **10**, 016005 (2013).
9. Holland, M. A., Miller, K. E. & Kuhl, E. Emerging Brain Morphologies from Axonal Elongation. *Ann. Biomed. Eng.* **43**, 1640–1653 (2015).
10. Goriely, A., Budday, S. & Kuhl, E. Neuromechanics. in *Advances in Applied Mechanics* vol. 48 79–139 (Elsevier, 2015).
11. Garcia, K. E., Wang, X. & Kroenke, C. D. A model of tension-induced fiber growth predicts white matter organization during brain folding. *Nat. Commun.* **12**, 6681 (2021).
12. Stoekli, E. T. Understanding axon guidance: are we nearly there yet? *Development* **145**, dev151415 (2018).
13. Wang, X., Wang, S. & Holland, M. A. Axonal tension contributes to consistent fold placement. *Soft Matter* **20**, 3053–3065 (2024).
14. Chavoshnejad, P. *et al.* Mechanical hierarchy in the formation and modulation of cortical folding patterns. *Sci. Rep.* **13**, 13177 (2023).
15. Chavoshnejad, P. *et al.* Role of axonal fibers in the cortical folding patterns: A tale of variability and regularity. *Brain Multiphysics* **2**, 100029 (2021).
16. Chang, Y.-J., Tsai, C.-J., Tseng, F.-G., Chen, T.-J. & Wang, T.-W. Micropatterned stretching system for the investigation of mechanical tension on neural stem cells behavior. *Nanomedicine Nanotechnol. Biol. Med.* **9**, 345–355 (2013).
17. Franze, K. The mechanical control of nervous system development. *Development* **140**, 3069–3077 (2013).
18. Schilling, K. *et al.* Confirmation of a gyral bias in diffusion MRI fiber tractography. *Hum. Brain Mapp.* **39**, 1449–1466 (2018).
19. Delettre, C. *et al.* Comparison between diffusion MRI tractography and histological tract-tracing of cortico-cortical structural connectivity in the ferret brain. *Netw. Neurosci.* **3**, 1038–1050 (2019).

20. Yendiki, A. *et al.* Post mortem mapping of connectional anatomy for the validation of diffusion MRI. *NeuroImage* **256**, 119146 (2022).
21. Girard, G. *et al.* On the cortical connectivity in the macaque brain: A comparison of diffusion tractography and histological tracing data. *NeuroImage* **221**, 117201 (2020).
22. Toro, R. & Burnod, Y. A Morphogenetic Model for the Development of Cortical Convolution. *Cereb. Cortex* **15**, 1900–1913 (2005).
23. Kostovic, I. & Rakic, P. Developmental history of the transient subplate zone in the visual and somatosensory cortex of the macaque monkey and human brain. *J. Comp. Neurol.* **297**, 441–470 (1990).
24. Pfister, B. J., Iwata, A., Meaney, D. F. & Smith, D. H. Extreme Stretch Growth of Integrated Axons. *J. Neurosci.* **24**, 7978–7983 (2004).
25. De Vincentiis, S. *et al.* Extremely Low Forces Induce Extreme Axon Growth. *J. Neurosci.* **40**, 4997–5007 (2020).
26. Chada, S., Lamoureux, P., Buxbaum, R. E. & Heidemann, S. R. Cytomechanics of neurite outgrowth from chick brain neurons. *J. Cell Sci.* **110**, 1179–1186 (1997).
27. Zheng, J. *et al.* Tensile regulation of axonal elongation and initiation. *J. Neurosci.* **11**, 1117–1125 (1991).
28. Magdesian, M. H. *et al.* Atomic Force Microscopy Reveals Important Differences in Axonal Resistance to Injury. *Biophys. J.* **103**, 405–414 (2012).
29. Zhang, Y. *et al.* Modeling of the axon membrane skeleton structure and implications for its mechanical properties. *PLOS Comput. Biol.* **13**, e1005407 (2017).
30. Arellano, J. I. & Rakic, P. Old Models Know Wrinkles Best: A Critical Review on the Mechanisms of Cortical Gyrification. in *Neocortical Neurogenesis in Development and Evolution* (ed. Huttner, W.) 499–525 (Wiley, 2023). doi:10.1002/9781119860914.ch23.
31. Welker, W. Why Does Cerebral Cortex Fissure and Fold? in *Cerebral Cortex* (eds. Jones, E. G. & Peters, A.) vol. 8B 3–136 (Springer US, Boston, MA, 1990).
32. Caviness, V. Mechanical model of brain convolitional development. *Science* **189**, 18–21 (1975).
33. Tallinen, T. *et al.* On the growth and form of cortical convolutions. *Nat. Phys.* **12**, 588–593 (2016).
34. Armstrong, E., Schleicher, A., Omran, H., Curtis, M. & Zilles, K. The Ontogeny of Human Gyrification. *Cereb. Cortex* **5**, 56–63 (1995).
35. Hilgetag, C. C. & Barbas, H. Developmental mechanics of the primate cerebral cortex. *Anat. Embryol. (Berl.)* **210**, 411–417 (2005).
36. Raghavan, R., Lawton, W., Ranjan, S. R. & Viswanathan, R. R. A Continuum Mechanics-based Model for Cortical Growth. *J. Theor. Biol.* **187**, 285–296 (1997).
37. Tallinen, T., Chung, J. Y., Biggins, J. S. & Mahadevan, L. Gyrification from constrained cortical expansion. *Proc. Natl. Acad. Sci.* **111**, 12667–12672 (2014).
38. Ronan, L. *et al.* Differential Tangential Expansion as a Mechanism for Cortical Gyrification. *Cereb. Cortex* **24**, 2219–2228 (2014).
39. Razavi, M. J., Zhang, T., Li, X., Liu, T. & Wang, X. Role of mechanical factors in cortical folding development. *Phys. Rev. E* **92**, (2015).
40. Jalil Razavi, M., Zhang, T., Liu, T. & Wang, X. Cortical Folding Pattern and its Consistency Induced by Biological Growth. *Sci. Rep.* **5**, (2015).

41. Mortazavi, F., Romano, S. E., Rosene, D. L. & Rockland, K. S. A Survey of White Matter Neurons at the Gyral Crowns and Sulcal Depths in the Rhesus Monkey. *Front. Neuroanat.* **11**, 69 (2017).
42. Van Essen, D. C. A 2020 view of tension-based cortical morphogenesis. *Proc. Natl. Acad. Sci.* **117**, 32868–32879 (2020).
43. Liu, M. *et al.* Robust Cortical Thickness Morphometry of Neonatal Brain and Systematic Evaluation Using Multi-Site MRI Datasets. *Front. Neurosci.* **15**, 650082 (2021).
44. Smith, D. R. *et al.* Multi-Excitation Magnetic Resonance Elastography of the Brain: Wave Propagation in Anisotropic White Matter. *J. Biomech. Eng.* **142**, 071005 (2020).
45. Staii, C. *et al.* Distance dependence of neuronal growth on nanopatterned gold surfaces. *Langmuir ACS J. Surf. Colloids* **27**, 233–239 (2011).
46. Sunnerberg, J. P., Descoteaux, M., Kaplan, D. L. & Staii, C. Axonal growth on surfaces with periodic geometrical patterns. *PLOS ONE* **16**, e0257659 (2021).
47. Rizzo, D. J. *et al.* Neuronal growth as diffusion in an effective potential. *Phys. Rev. E* **88**, 042707 (2013).
48. Moore, D. L. & Goldberg, J. L. Multiple transcription factor families regulate axon growth and regeneration. *Dev. Neurobiol.* **71**, 1186–1211 (2011).
49. Raffa, V. *et al.* Piconewton Mechanical Forces Promote Neurite Growth. *Biophys. J.* **115**, 2026–2033 (2018).
50. Yurchenko, I., Vensi Basso, J. M., Syrotenko, V. S. & Staii, C. Anomalous diffusion for neuronal growth on surfaces with controlled geometries. *PLOS ONE* **14**, e0216181 (2019).
51. Katz, M. J., George, E. B. & Gilbert, L. J. Axonal elongation as a stochastic walk. *Cell Motil.* **4**, 351–370 (1984).
52. Budday, S., Raybaud, C. & Kuhl, E. A mechanical model predicts morphological abnormalities in the developing human brain. *Sci. Rep.* **4**, 5644 (2015).

Review Reply letter

We sincerely thank all the reviewers for their thorough review, positive feedback, and constructive comments. We especially appreciate Reviewer #3 for their detailed evaluation of the revised manuscript and thoughtful suggestions. Below, we provide a point-by-point response to the minor and major revisions made to the manuscript. In this letter, we have provided detailed responses to each specific question raised, highlighted in blue. Additionally, new changes to the main text, including those based on editorial comments/formatting requirements, are highlighted in red in the revised manuscript.

Reviewer #1 (Remarks to the Author):

The authors have addressed the concerns raised in my previous review.

We thank the reviewer for their positive feedback on the revised manuscript. We genuinely appreciate the reviewer's comments and insights, which have been invaluable in enhancing the quality of our study.

Reviewer #2 (Remarks to the Author):

Thank you for these clarifications to the manuscript. The new version is much improved.

We sincerely thank the reviewer for their positive feedback on the revised manuscript. We greatly appreciate the reviewer's comments and insights, which have been instrumental in improving the quality of our study.

Reviewer #3 (Remarks to the Author):

The responses from the authors and the changes that they have made in the manuscript have addressed many of the issues raised by the reviewers, but I still have significant concerns about missing or unclear methods and overstated claims.

Thank you for the reviewer's thoughtful and positive feedback. We sincerely appreciate the reviewer's in-depth evaluation and for providing valuable insights and comments, all of which we recognize as important and necessary to address in order to improve both the clarity and accuracy of the study's presentation and the language used. Below, we have provided detailed responses to each of the reviewer's concerns and comments and have revised the manuscript accordingly. We hope these revisions have effectively addressed the reviewer's concerns, improved the clarity of the methods, and ensured that any overstated claims have been appropriately refined.

Major issues:

1. The abstract says "In particular, the degree of gyration is shown to correlate strongly with the density, growth rate, and mechanical properties of underlying axon bundles." However, there is very little evidence - certainly not to support a claim of "strong correlation". For example, Fig. 5 looks at the effect of growth rate but changes in morphology are not quantified; Figs. 6 and 7 quantify the effect of axonal stiffness and number of fibers on GI and find

statistically significant results (but realistically, likely a very small effect size). What seems to be somewhat more supported by the data is that the cortical folding morphology affects the growth of axons (in their simulations; not necessarily in vivo).

We thank the reviewer for highlighting this critical issue. We fully agree that, in some instances, the presentation of the results required revision to enhance accuracy and eliminate any overstated claims. In response, we have revised the abstract to address the reviewer's constructive feedback. Additionally, we conducted a thorough review of the manuscript and made necessary changes to ensure that any overstated claims were corrected.

We also aimed to convey more clearly, as the reviewer pointed out, that while the effect of fiber properties on gyrification index may not be significant when considered in absolute terms, even though statistically significant differences exist between cases with varying fiber stiffness and fiber volume fractions. Importantly, our results demonstrate that the density of fibers reaching the cortical plate prior to folding plays a significant role in determining the specific locations where gyri and sulci form.

In the first revision, we explicitly informed readers that the results presented are derived from models, which may not perfectly replicate real brain conditions. However, in this revision, we have further emphasized the modeling nature of the study and the derived conclusions by using terms such as "The model predicts", "The models predict", or "model predictions".

In the following, we have listed the specific changes made to the manuscript:

- Abstract:

"In particular, the connectivity patterning in the folding-induced stress field exhibits strong dependence on the density, growth rate, and mechanical properties of the navigating axon bundles."

- Results:

"Remarkably, with an increase in the stiffness ratio, the GI exhibited a statistically significant decrease, while its absolute value changed only slightly."

"Increased stiffness in the white matter region leads to a reduction in cortical gyrification and a lower GI (Fig. 7(e)), although this effect is statistically significant only with a substantial change in the fiber volume fraction."

2. Relatedly, at the bottom of page 28 the section is titled "Effect of stochastic nature of axon growth on cortical folding and fiber organization" but the related figure, Fig. 9, does not quantify the changes in cortical folding (e.g. gyrification index, wavelength, or sulcal depth).

We thank the reviewer for this valuable and constructive comment. We apologize for not recognizing this issue in the previous versions of the manuscript. The titles of the subsections were initially established during the early stages of writing, and we did not update them to accurately reflect the observations and changes made in those sections.

In the revised manuscript, we have carefully revised the subsection titles to better align with the findings presented in each part of the study. Furthermore, we thoroughly reviewed the entire manuscript to identify and correct any similar inconsistencies. Below is the updated list of subsection titles:

Results

- Axonal growth in a stress field
- Effect of axon growth rate on fiber organization
- Effect of axon mechanical properties on fold morphology and fiber organization
- Effect of cortex stiffness on fiber organization
- Effect of stochastic nature of axon growth on fiber organization
- Findings from imaging data for model evaluation

Discussion

- Axonal fibers concentrate more in gyri than sulci
 - Axonal growth in a stress field
 - Effect of axons biophysical properties on fold placement and fiber organization
 - Effect of cortex stiffness on fiber organization
 - Limitations and potential improvements
3. Several methodological details are still missing.
- a. While the response to reviewers explained the use of splines to visualize axonal directions, the only mention of splines in the manuscript is to point out where they aren't used (p22).

We thank the reviewer for pointing out this point of confusion. We have revised the manuscript to ensure the use of spline curves is presented in the appropriate context. It is important to clarify that in our study, spline curve fitting was applied solely to improve the visualization of fiber paths by creating smoother lines, rather than displaying them as connected chains of elements. This was done for illustrative purposes only and did not influence the extraction of any results. All values presented in the study are derived directly from the finite element model without modification. To further clarify, we have included a comparison image showing the model with and without spline fitting. We added a statement to the Methods section to explicitly address this.

“We applied splines to create smooth visual representations of fiber paths, rather than showing them as connected chains of elements. The spline fitting was used primarily to enhance the visualization in the figures and did not affect the extraction of quantitative data. All values presented in the study were derived directly from the FE models without any alterations. The original and dynamic representations of the FE models are available in the supplementary movies.”

- b. The calculation of GI is said to require fitting of a "smoothed, curved line that minimally encloses the cortical surface". More details would be helpful. I also assume the dotted line in Fig. 6c is a hand drawing and not the actual convex hull, because it shows some small areas of concavity.

We thank the reviewer for this constructive comment. The convex hull of the cortical boundary was calculated using MATLAB, which employs a standard algorithm to enclose all points with a minimal convex shape. This convex hull serves as the reference outer boundary (Pial) for calculating the GI. The term 'smoothed, curved line' refers to this convex line that wraps around the cortical boundary, eliminating any inward curves or concavities to create the smallest convex shape that completely enclose the cortex.

Regarding the line shown in Fig. 6c, we agree with the reviewer that the dotted line was intended to illustrate the concept of the convex hull but may not strictly represent the actual convex hull used in the calculations. We used this line to visualize the overall contour, but for precise calculations, the convex hull was determined by MATLAB, ensuring no areas of concavity. We have replaced the dotted line with exact convex hull used in our calculations for better clarity in the following figure. In addition, we have added the following sentences to the Methods section to provide further details on the concept of the gyrification index and its calculation.

Methods: "We used the gyrification index (GI) as a metric to quantify the degree of cortical folding and assess the influence of model parameters on folding patterns. The GI is defined as the ratio of the cortical surface area (Pial) to the convex hull surface area. In our 2D models, this simplifies to the ratio of the total length of the gyri line to the length of the convex hull line. The convex hull represents the smallest convex boundary that fully encloses the gyri line. For the models, the convex hull of the cortical boundary was calculated using MATLAB, which applies a standard algorithm to generate the minimal convex shape that encloses all points."

Results: "The GI is defined as the ratio of the total length of the cortical contour, represented by the solid line in Fig. 6(c), to the length of the convex hull, depicted by the dashed line in Fig. 6(c) (refer to the Methods section for further details)."

- c. Many similar simulations include some kind of perturbation or imperfection to trigger buckling. Was one used here?

We thank the reviewer for raising this important point. In finite element buckling studies, it is common practice to introduce imperfections in either geometry or material properties to initiate buckling, particularly when using a static solver^{1,2}. However, in our case, where a dynamic solver is employed, instability is naturally triggered due to the dynamic terms in the formulation. To ensure quasi-static equilibrium, the kinetic energy in the system should remain below 5% of the internal energy³. In our models, beyond the dynamic contributions, there is the initiation and establishment of axon elements with different material properties than the ECM, introducing heterogeneity. This heterogeneity acts as an imperfection, effectively triggering buckling and contributing to fold formation. We added the following sentences to the revised manuscript to address the comment:

“The initiation and establishment of axon elements, with material properties distinct from the ECM, introduce imperfections into the system, facilitating the triggering of buckling.

However, to ensure quasi-static equilibrium in the dynamic solver, the system's kinetic energy should remain below 5% of the internal energy³."

4. I believe Table 1 has a mistake; if the "Else if" statement in step 2 is not true, then the quantity $\mathbf{n}^A_{T_max}$ would be undefined.

We appreciate the reviewer for highlighting this omission. While the developed code is accurate and includes the term (please refer to the code), we unfortunately overlooked it when preparing Table 1. This omission has been rectified in the revised manuscript, as detailed below.

Table 1. Algorithm update scheme for the reorientation of axon bundle	
Internal variable $\mathbf{n}_k^A = [n_{k1}^A, n_{k2}^A, n_{k3}^A]^t$	
1. Set initial values	$\mathbf{n}^A = \mathbf{n}_k^A, \quad \mathbf{T} = J^{e-1} \mathbf{F}^e \cdot \frac{\partial W}{\partial \mathbf{F}^e \mathbf{T}}$
2. Compute principal stresses and directions	$\mathbf{T} = J^{e-1} \mathbf{F}^e \cdot \frac{\partial W}{\partial \mathbf{F}^e \mathbf{T}} = \sum_{i=1}^3 \sigma_i \mathbf{n}_i \otimes \mathbf{n}_i \quad \text{with} \quad \sigma_1 \geq \sigma_2 \geq \sigma_3$
	Let $\mathbf{n}_{max}^T = \mathbf{n}_1^T$
	If $\sigma_1 < 0$ or $\sigma_1 = \sigma_2$ then
	$\mathbf{n}^A = 0$
	Else if $\mathbf{n}^A \cdot \mathbf{n}_1^T < 0$ then
	$\mathbf{n}_{max}^T \leftarrow -\mathbf{n}_1^T$
	Else
	\mathbf{n}_{max}^T remains as \mathbf{n}_1^T
	End if
3. Compute the magnitude and normal vector of rotation velocity	$\omega = \frac{\pi}{2t^*} \ \mathbf{n}^A \times \mathbf{n}_{max}^T\ $
	$\mathbf{n}^\omega = \frac{\mathbf{n}^A \times \mathbf{n}_{max}^T}{\ \mathbf{n}^A \times \mathbf{n}_{max}^T\ }$
4. Update preferred direction vector	$\mathbf{n}_{k+1}^A = \cos(\Delta t \omega) \mathbf{n}^A + \sin(\Delta t \omega) \mathbf{n}^\omega \times \mathbf{n}^A + [1 - \cos(\Delta t \omega)][\mathbf{n}^\omega \cdot \mathbf{n}^A] \mathbf{n}^\omega$
	$\mathbf{n}_k^A = \mathbf{n}_{k+1}^A$

5. In Fig. S1, why does the number of simulations affect the results? I expected to see some narrowing of a range of uncertainty, but in fact uncertainty/standard deviation is not even indicated on the plot.

Thank you to the reviewer for this constructive comment. We have revised Fig. S1(a) to include the standard deviation, which now captures the variability across simulations. In fact, the purpose

of this figure is to determine the minimum number of simulations required to ensure that the results are independent of the number of simulations. In our models, fiber initiation occurs at random locations, so running only one or a few simulations could lead to variability in the results and potentially misleading conclusions. To ensure the results are independent of fiber initiation locations and thus independent of the number of simulations, we generated Fig. S1 to assess how many simulations are needed to achieve stable and repeatable outcomes. As shown in Fig. S1(a), the fiber density stabilizes (within $\pm 1\%$ change in mean) after approximately 10 simulations, and the standard deviation also stabilizes beyond this point.

Fig. S1 (a) The impact of varying simulation counts on fiber density within gyri and sulci regions. Results demonstrate stability in fiber density beyond a threshold of 10 simulations. (b) Relationship between the number of fiber bundles and the resulting fiber density. The number of fibers doesn't significantly impact the fiber density within gyri and sulci. The used parameters in these figures are: $\mu_f/\mu_s = 2$, $\mu_c/\mu_s = 2$, $a = 0.015 \text{ mm Pa}^{-1} \text{ d}^{-1}$, $G^{\text{axn}} = 0.8 \text{ mm d}^{-1}$. Data are represented as mean values \pm SD (Standard Deviation).

Regarding the expected narrowing of uncertainty with an increasing number of simulations, it is important to clarify that while standard deviation measures the spread of individual data points around the mean, it does not directly represent the uncertainty of the mean itself. To capture how uncertainty decreases as more simulations are conducted, the standard error of the mean (*SEM*) would be the appropriate metric. *SEM* is calculated as:

$$SEM = \frac{SD}{\sqrt{n}}$$

where *SD* is the standard deviation and *n* is the number of simulations. This metric clearly illustrates the reduction in uncertainty as the sample size increases. To address the reviewer's concern, we have recreated Fig. S1(a) with *SEM* as the uncertainty measure, and the expected narrowing of the uncertainty range with increasing simulations is now noticeable. However, in the manuscript, we opted to maintain the use of standard deviation for consistency in error representation throughout. We hope this explanation clarifies the choice and presentation of uncertainty in Fig. S1.

6. The inclusion of the timeline in Table S1 is helpful. However, please indicate the approximate weeks that correspond to your simulations.

Thank you to the reviewer for this constructive comment. We have incorporated additional information in the Results section to align our simulations with the developmental timeframe of the axonal fibers. It is important to note that we considered the growth and development of axonal fibers from the deep brain core to the deep subplate region to have occurred prior to T=0.

“At T=0, cortical expansion initiates (corresponding to approximately 16–20 GWs in Table S1), when thalamocortical fiber tracts begin to spread in the deep subplate. At T=1, the simulation ends (aligning with around 27–36 GWs in Table S1), when axons have settled in the cortical plate.”

Minor issues:

7. The scales of the y axes in Fig. 7e and 6c are extremely misleading and should be revised to include 1.

Thank you for pointing out the scaling on the y-axes in Fig.7e and 6c. We acknowledge that the scales may appear misleading as they do not currently include 1. To enhance clarity and ensure accurate representation, we have adjusted the y-axes scales in both figures. This revision will help readers interpret the results more accurately.

8. In Fig. S3, there are interesting green shapes in the middle of the mostly-blue substrate. Can the authors explain these?

Thank you to the reviewer for this interesting observation. The green shapes in the mostly blue substrate represent regions with slightly higher stress than the surrounding blue areas. Some of these regions contain axons, which have a higher stiffness compared to the substrate. We used a divergent color scale with 255 color levels, making these subtle variations in stress more visible. If we were to use a simplified color scale, such as the 12-color scheme commonly used in ABAQUS, these differences would appear less distinct, resulting in a more uniform color across the substrate. Please see the comparison image below for reference.

9. Fig. S4 still has the titles "with/no reorientation" that was fixed in the last revision.

We apologize for failing to catch this issue. The title of Fig.S4 has been corrected.

Fig. S4 Dynamic growth of fibers for the growth rate of 1.2 mm d^{-1} in models **with and without stress-dependent reorientation** process. The used parameters in these figures are: $\mu_f/\mu_s = 2$, $\mu_c/\mu_s = 2$, $a = 0.015 \text{ mm Pa}^{-1} \text{ d}^{-1}$, $G^{\text{axn}} = 1.2 \text{ mm d}^{-1}$.

10. On p29, specify what quantity you are discussing the standard deviation of.

Thank you for pointing out this issue. In this section, the standard deviation refers specifically to the variability in the orientation angle of the preferred growth directions of axons (\mathbf{n}_k^A) within our models. This variability allows us to mimic the stochastic nature of axonal growth. We have clarified this in the revised text on p29 to ensure that the reference to standard deviation is specific and clear.

11. Some nomenclature is inconsistent - both preferred and present direction are used on p14, and MPTS is used instead of MTPS on p22.

We apologize for failing to catch these instances. We have replaced the terms “present” and “MPTS” with “preferred” and “MTPS”, respectively. We thoroughly reviewed the manuscript to eliminate any inconsistencies.

Reviewer #3 (Remarks on code availability):

I briefly looked through the code and did not see any significant reasons for concern.

We sincerely appreciate the reviewer's time and feedback in evaluating the developed code.

Reviewer #4 (Remarks to the Author):

I have no further comments.

We thank the reviewer for their positive feedback on the revised manuscript. We genuinely appreciate the reviewer's comments and insights, which have been invaluable in enhancing the quality of our study.

References

1. Nikraves, S., Ryu, D. & Shen, Y.-L. Instabilities of Thin Films on a Compliant Substrate: Direct Numerical Simulations from Surface Wrinkling to Global Buckling. *Sci. Rep.* **10**, 5728 (2020).
2. Wang, X., Wang, S. & Holland, M. A. Axonal tension contributes to consistent fold placement. *Soft Matter* **20**, 3053–3065 (2024).
3. Jafarabadi, F., Wang, S. & Holland, M. A. A Numerical Study on the Influence of Cerebrospinal Fluid Pressure on Brain Folding. *J. Appl. Mech.* **90**, 071006 (2023).

Review Reply Letter

We sincerely thank the reviewer for their detailed and constructive evaluation of the revised manuscript. Below, we provide a point-by-point response to the minor revisions made to the manuscript. In this letter, we have included detailed responses to each specific question raised, with the responses highlighted in **blue** and the corresponding revisions in the manuscript highlighted in **red**. However, new minor changes to the main text are not highlighted, as per the formatting instructions of the journal.

Reviewer #3 (Remarks to the Author):

Thank you to the authors for their revisions and response. I continue to have concerns about the support of the authors' claims.

1. The revised abstract states that "Importantly, our results demonstrate that the density of fibers reaching the cortical plate prior to folding plays a significant role in determining the specific locations where gyri and sulci form." Where is this information shown? Figs. 5 and 8 show simulations where fibers reach the cortical plate before and after folding begins, but don't compare resulting morphologies or fold placement, only fiber density in gyri and sulci.

Thank you to the reviewer for raising this comment. The quoted statement, *"Importantly, our results demonstrate that the density of fibers reaching the cortical plate prior to folding plays a significant role in determining the specific locations where gyri and sulci form,"* is included exclusively in the previous Review Reply Letter in response to the reviewer's comments for more discussion, and does not appear in the abstract or any other part of the revised manuscript.

While the statement is accurate and aligns with our models, which demonstrate that areas with a higher density of fibers reaching the cortical plate predominantly form gyri rather than sulci, we have not elaborated on or quantified this finding in the manuscript, as it lies outside the scope of the current study. To clarify, please refer to the following figure. As the figure illustrates, in the center of the model, a sulcus consistently forms when fibers are absent. However, when a region with dense, growing fibers is introduced beneath the center, the termination point of this dense fiber tract invariably corresponds to the formation of a gyrus after folding. We repeated this model multiple times, and a gyrus consistently formed at the center of the model.

Fig. R1. Comparison of models with and without fibers. a) In the absence of fibers, based on the cortical folding over the substrate and the applied mechanical parameters, a sulcus consistently forms at the center

of the cortex in the half-circle model. b) When a region with dense fibers is positioned beneath the center of the cortex, the growth and termination of these fibers at the cortical plate consistently influence the morphology, leading to the formation of a gyrus. This gyrus formation is reproducible across various randomized models with concentrated fibers at the center.

2. The revised abstract also mentions "the connectivity patterning in the folding-induced stress field". What does this mean? What stress field?

Thank you to the reviewer for this insightful comment. The folding of the cortical plate onto the white matter (or early subplate) generates a stress field within the white matter. This stress landscape, characterized by a distribution of tensile and compressive stresses (as shown in the Fig. 2 of the revised manuscript) influences and steers the growth paths of axonal fibers. The stress field in the white matter region is depicted using stress contours. Since this stress field arises as a result of cortical folding, we have termed it the "folding-induced stress field." To enhance clarity in the revised manuscript, we have explicitly added the term "white matter" to the relevant statement, which now reads:

"In particular, the connectivity patterning resulting from cortical folding exhibits a strong dependence on the growth rate and mechanical properties of the navigating axon bundles."

3. The mention of cortical folding morphology depending on fiber organization was removed from the section heading, but not the caption of Fig. 9.

Thank you to the reviewer for point out this issue. In the revised manuscript, we addressed the issue.